# UniGAN: Reducing Mode Collapse in GANs using a Uniform Generator

**Ziqi Pan, Li Niu,**\* **Liqing Zhang**\*
MoE Key Lab of Artificial Intelligence
Department of Computer Science and Engineering
Shanghai Jiao Tong University, Shanghai, China
`{panziqi_ai, ustcnewly}@sjtu.edu.cn, zhang-lq@cs.sjtu.edu.cn`

## Abstract

Despite the significant progress that has been made in the training of Generative Adversarial Networks (GANs), the mode collapse problem remains a major challenge in training GANs, which refers to a lack of diversity in generative samples. In this paper, we propose a new type of generative diversity named *uniform diversity*, which relates to a newly proposed type of mode collapse named *u-mode collapse* where the generative samples distribute nonuniformly over the data manifold. From a geometric perspective, we show that the uniform diversity is closely related with the *generator uniformity* property, and the maximum uniform diversity is achieved if the generator is uniform. To learn a uniform generator, we propose *UniGAN*, a generative framework with a Normalizing Flow based generator and a simple yet sample efficient generator uniformity regularization, which can be easily adapted to any other generative framework. A new type of diversity metric named *udiv* is also proposed to estimate the uniform diversity given a set of generative samples in practice. Experimental results verify the effectiveness of our UniGAN in learning a uniform generator and improving uniform diversity.

## 1 Introduction

Generative Adversarial Networks [1] (GANs) are a technique for unsupervised learning generative models to capture an unknown data distribution. Given training samples, two modules, a *generator* and a *discriminator*, contest with each other in form of a zero-sum game, where the generator aims to produce realistic samples and the discriminator aims to distinguish generative samples from real data. Though originally proposed as a generative model for unsupervised learning, GANs have also been widely used in a variety of learning scenarios [2, 3, 4] due to not only their ability to learning highly structured probability distributions but also their theoretical implications [1, 5, 6].

GANs training is notoriously difficult and is very sensitive to almost every aspect of its setup due to the dynamic training paradigm [7, 8]. Despite significant progress [9, 10, 11, 12, 13, 14] have been made in the training of GANs, the *mode collapse* [15] problem remains a major challenge in training GANs, which refers to a lack of diversity in generative samples [15]. Existing methods reduce mode collapse by modifying model architectures [3, 16, 15, 17, 18, 19, 20], proposing novel optimization algorithms [21, 22, 23, 24, 25] and regularizations [26, 27, 28, 29], *etc.* Several works also provide empirical explanations or formal analysis for mode collapse, including catastrophic forgetting [30] of the discriminator, sample weighting [31], and binary hypothesis testing [15].

In this paper, we adopt the *manifold hypothesis* [32, 33] to formalize our analysis and develop our framework named *UniGAN* for reducing mode collapse. The manifold hypothesis assumes that high

---

\*Corresponding author

36th Conference on Neural Information Processing Systems (NeurIPS 2022).

dimensional data is concentrated on a lower dimensional manifold. First, this assumption provides us prior knowledge about the unknown ground-truth data distribution $p$. Specifically, from the manifold hypothesis, it is natural to *adopt the uniform distribution over the data manifold $\mathcal{M}$ where training samples come from as the ground-truth data distribution $p$*, since samples on $\mathcal{M}$ are equally accepted as real samples for training GANs. Given that $p$ is the uniform distribution over $\mathcal{M}$, we propose the *uniform diversity* of a generative distribution $q$ with formalized definition that measures how close $q$ is to $p$ by computing the differential entropy of $q$, considering that the uniform distribution gives the minimum entropy. Recent work [34, 29] also deem that a diverse generator should yield a generative distribution with lower entropy. To quantify the extent of mode collapse with respect to the uniform diversity, we further propose the *u-mode collapse*, which is a new type of mode collapse compared to the existing $(\varepsilon, \delta)$-mode collapse [15]. We also present theoretical analysis showing that the $u$-mode collapse is more generalized than the $(\varepsilon, \delta)$-mode collapse and hence is more robust and efficient in capturing mode collapse. Second, to maximize uniform diversity, based on the manifold hypothesis, we derive the *uniformity* property of the generator, providing both formalized definition and intuitive geometric interpretation. We also relate the generator uniformity to the uniform diversity, showing that the maximum uniform diversity is achieved if the generator is sufficiently uniform.

To maximize uniform diversity, we propose a framework named *UniGAN*, which consists of a normal discriminator, a Normalizing Flow [35, 36, 37] (NF) based generator, and a simple yet sample efficient regularization for pursuing a uniform generator. NFs are generative models with specifically designed architectures that are bijective and easy-to-invert. Hence we use an NF-based generator since these favorable properties not only make the generator intrinsically satisfy the theoretical prerequisites for maximizing uniform diversity, but also facilitate us to develop a generator uniformity regularization in a sample efficient manner. Moreover, the regularization is simple and thus can be easily integrated into any models, which makes our UniGAN easy to adapt to any generative frameworks. To realize the computation of uniform diversity in practice, we also propose a new metric named *udiv* estimating the uniform diversity of a given set of generative samples. Experimental results show that UniGAN is effective on improving generator uniformity and uniform diversity. Our contributions are threefold:

- We propose a new type of generative diversity named *uniform diversity* and the associated *u-mode collapse* that measures how close a generative distribution is to the data distribution which we chose to be the uniform distribution over the data manifold based on the manifold hypothesis. Theoretical analysis shows that $u$-mode collapse is a generalization of existing $(\varepsilon, \delta)$-mode collapse and is more robust and efficient in capturing mode collapse.

- We derive the *generator uniformity* property that is closely related to uniform diversity, and provide analysis showing that the maximum uniform diversity is achieved if the generator is sufficiently uniform. A geometric interpretation is also provided.

- We propose *UniGAN*, a generative framework with an NF-based generator and a simple yet sample efficient regularization on generator uniformity, which can be easily adapted to any other generative framework. We also propose a new diversity metric named *udiv* estimating the uniform diversity of a given set of generative samples. Experimental results verify the effectiveness of our UniGAN in improving generator uniformity and uniform diversity.

## 2   Related Work

**Reducing Mode Collapse**   A variety of methods have been proposed to mitigate the mode collapse problem in the training of GANs, including modified architectures [3, 4, 16, 38, 15], optimization algorithms [21, 22, 23, 39], and loss functions [34, 40, 26, 27, 28]. Several methods aim to maintain a bijection between the latent and data space by learning an encoder that projects data back into the latent space [3, 4, 16, 41, 42, 38]. For example, BiGANs/ALI [3, 4] and VEEGANs [41, 42] propose to learn an encoder implicitly by making the discriminator to distinguish not only in data space but jointly in data and latent space, and BicycleGAN [16] explicitly encourages reconstruction of both latent codes and training data by combining VAEGAN [41, 42] and LRGAN [16]. Several methods make hypothesis of weak generators [17, 18, 19] and weak discriminators [43, 44, 22, 9, 15, 20] for mode collapse problem. For example, MAD-GAN [17] employs multiple generators to enforce that different generators capture diverse modes, and PacGAN [15] reduces mode collapse by modifying the discriminator to make decisions based on multiple rather than single samples from the same class, either real or artificially generated. PacGAN also provide analysis on mode collapse by proposing a formal definition of mode collapse, namely $(\varepsilon, \delta)$-mode collapse from the view of binary hypothesis

testing. Similarly, VirtualGAN [45] is proposed to merge multiple samples into one before training the discriminator. Another line of research reduces mode collapse by proposing novel optimization algorithms. Inspired by boosting algorithms, AdaGAN [21] is proposed as an iterative meta-algorithm where many potentially weak generators are greedily aggregated to form a strong composite generator. The *unrolled GAN* [22] aims to reduce mode collapse and stabilize training of GANs by defining the generator's objective with respect to an unrolled optimization of the discriminator. The *progressive growing of GAN* [23] aims to grow both the generator and discriminator progressively: starting from a low resolution, then add new layers that model increasingly fine details as training progresses. The BourGAN [24] replaces Gaussian with a Gaussian mixture for the prior latent distribution, since the authors assume that the incompatibility between the prior latent distribution and the mode structure is a cause of the mode collapse problem, which is also supported by the analysis of [46] from the view of optimal transport. There are also some works propose to reduce mode collapse by proposing novel regularizations. For example, MSGAN [40, 26] maximizes the ratio of the distance between samples with respect to the distance between the corresponding latent codes to reduce mode collapse.

**Isometry Learning**   For ease of practical implementation of our regularization on generator uniformity, we turn to a special case of uniform generators, namely *isometric* generators whose Jacobians are orthonormal (up to a constant) [47, 48, 49] (see Section 3.3), which is related to *isometry learning*. Several methods are proposed to learn isometric generators. Following the definition of an isometry, namely $J^\top J = I$ where $J$ is the Jacobian of the generator and $I$ is an identity matrix, the authors of LGAN [50] propose to constrain $v_i^\top v_j = \delta_{ij}$ where $v_i$ is the approximated $i$-th column vector of $J$ and $\delta_{ij}$ is the Kronecker delta. In StyleGAN2 [51], a regularization on path lengths of the generator is proposed to encourage that a fixed-size step in latent space results in a fixed-magnitude change in the generative image, which implicitly encourages the generator to be an isometry. For the task of manifold learning, an algorithm named *Riemannian relaxation* is proposed to construct an isometric mapping in [47]. However, this algorithm does not deal with high-dimensional input such as images, as mentioned by [49]. In RaDOGAGA [49], for an autoencoder, an isometric decoder is learned by injecting random noise to the latent code before reconstructing the input data. There are also several methods [52] learn isometric mappings aided by traditional dimensionality reduction algorithms [53].

## 3   Methodology

We firstly propose the formal definitions of *uniform diversity* and the associated $u$-*mode collapse* in Section 3.1, then relates the uniform diversity with the *generator uniformity* property in Section 3.2, which motivates us to develop the *UniGAN* framework for reducing mode collapse in Section 3.3.

### 3.1   The Uniform Diversity and $u$-Mode Collapse

Let $\mathcal{P}_{\mathcal{M}}$ be the set of all distributions over a finite domain $\mathcal{M}$. The *uniform diversity* of a distribution $q \in \mathcal{P}_{\mathcal{M}}$ measures how close $q$ is to the uniform distribution over $\mathcal{M}$, which is defined as

**Definition 1** (Uniform diversity). *The uniform diversity of $q \in \mathcal{P}_{\mathcal{M}}$ is given as $\mathcal{U}_q \triangleq \frac{e^{\mathcal{H}_q}}{m_{\mathcal{M}}} \in [0, 1]$, where $m_{\mathcal{M}}$ is the measure of $\mathcal{M}$ and $\mathcal{H}_q$ is the differential entropy of $q$, namely*

$$\mathcal{H}_q \triangleq - \int_{\mathrm{supp}(q')} \log q'(x)\, \mathrm{d}q(x), \tag{1}$$

*where $\mathrm{supp}(q') \triangleq \{x \in \mathcal{M} | q'(x) \neq 0\}$ is the support of $q'$. For $\mathcal{H}_q = -\infty$, we define $\mathcal{U}_q \triangleq 0$.*

Note that given a distribution $p \in \mathcal{P}_{\mathcal{M}}$, we use $p'$ to denote its Probability Density Function (PDF). We employ differential entropy $\mathcal{H}_q$ in Definition 1 since it measures the uniformity of distribution $q$. Specifically, from information theory [54] we know that $\mathcal{H}_q$ could be $-\infty$ which corresponds to the case where $q$ is extremely nonuniform (*e.g.*, $q'$ is a Dirac delta function where all probability mass is concentrated on a single point), and $\mathcal{H}_q$ is upper bounded by $\log m_{\mathcal{M}}$ since $\mathcal{M}$ is finite, where $\mathcal{H}_q = \log m_{\mathcal{M}}$ achieves only if $q$ is the uniform distribution over $\mathcal{M}$ (see supplementary). The more uniform the $q$, the higher the $\mathcal{H}_q$. The uniform diversity $\mathcal{U}_q$ is further defined to normalize the range of $\mathcal{H}_q$ to $[0, 1]$. Based on uniform diversity, the $u$-*mode collapse* is formally defined as

**Definition 2** ($u$-mode collapse). *We say that $q \in \mathcal{P}_{\mathcal{M}}$ exhibits $u$-mode collapse for $0 \leqslant u < 1$ if its uniform diversity $\mathcal{U}_q$ is equal to $u$.*

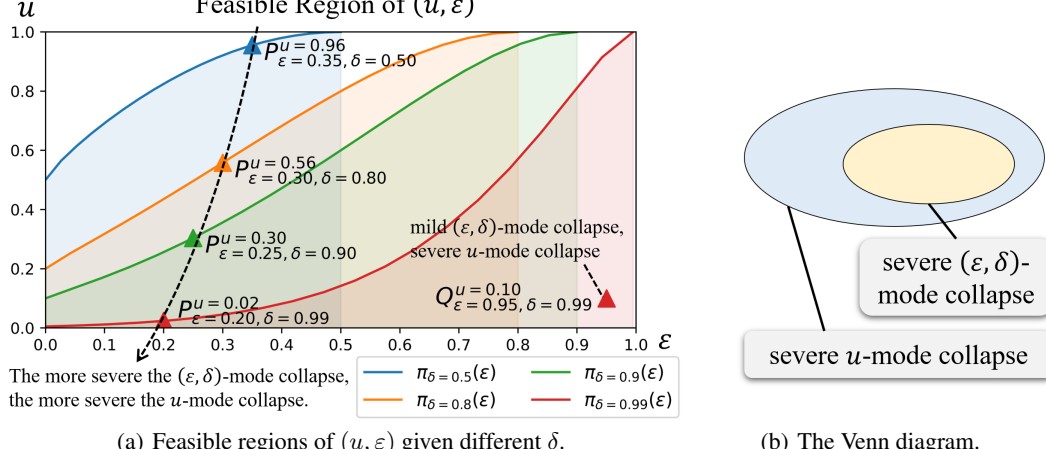

(a) Feasible regions of $(u, \varepsilon)$ given different $\delta$.      (b) The Venn diagram.

Figure 1: In Fig. 1(a), we illustrate the feasible region of $(u, \varepsilon)$ for $\delta \in \{0.5, 0.8, 0.9, 0.99\}$ respectively. Given $\delta$, the corresponding $\pi(\varepsilon, \delta)$ curve with respect to $\varepsilon$ is plotted, and the corresponding feasible region of $(u, \varepsilon)$ is shown as the shaded region under the curve. We also select four pairs of $(\varepsilon, \delta)$ with increasing $\delta$ and decreasing $\varepsilon$, and then plot points $P^u_{\varepsilon, \delta}$ that gives the maximum $u$ for the corresponding $(\varepsilon, \delta)$ to see how $u$-mode collapse changes as $(\varepsilon, \delta)$-mode collapse gets more severe. We also select a point $Q^u_{\varepsilon, \delta}$ to show that $(\varepsilon, \delta)$-mode collapse could be mild while $u$-mode collapse is severe. Different $\delta$ is represented by different color. In Fig. 1(b), a Venn diagram is used to show the relationship between $(\varepsilon, \delta)$-mode collapse and $u$-mode collapse.

The proposed $u$-mode collapse measures how far a distribution $q \in \mathcal{P}_{\mathcal{M}}$ deviates from the uniform distribution over $\mathcal{M}$. Lower $u$ indicates more severe mode collapse, and for $\mathcal{U}_q = 1$, $q$ is sufficiently uniform and does not exhibit $u$-mode collapse.

**Connection to $(\varepsilon, \delta)$-mode collapse**    Recently, another formal definition of mode collapse named $(\varepsilon, \delta)$-*mode collapse* is proposed in [15], which leads to a two-dimensional representation of the mode collapse region of the pair of the data distribution $p$ and a generative distribution $q$. We discuss how it relates to our proposed $u$-mode collapse. Recall that $(\varepsilon, \delta)$-mode collapse is formally defined as

**Definition 3** (($\varepsilon, \delta$)-mode collapse, [15]). *Given data distribution $p \in \mathcal{P}_{\mathcal{M}}$, a generative distribution $q \in \mathcal{P}_{\mathcal{M}}$ exhibits $(\varepsilon, \delta)$-mode collapse for $0 \leqslant \varepsilon < \delta < 1$ if there exists $\mathcal{S} \subset \mathcal{M}$ such that $p(\mathcal{S}) \geqslant \delta$ and $q(\mathcal{S}) \leqslant \varepsilon$.*

Given that the data distribution $p$ is chosen as the uniform distribution over $\mathcal{M}$, we show that $u$-mode collapse is more generalized than $(\varepsilon, \delta)$-mode collapse, and is more robust and efficient in capturing mode collapse. We first present the following Theorem 1 (see supplementary for proof):

**Theorem 1.** *For $q \in \mathcal{P}_{\mathcal{M}}$ that exhibits $(\varepsilon, \delta)$-mode collapse, $0 \leqslant \mathcal{U}_q \leqslant \pi(\varepsilon, \delta)$, where $\pi(\varepsilon, \delta) \triangleq \left(\frac{\delta}{\varepsilon}\right)^{\varepsilon} \left(\frac{1-\delta}{1-\varepsilon}\right)^{(1-\varepsilon)} \in [0, 1)$. That is, a generative distribution that exhibits $(\varepsilon, \delta)$-mode collapse at least exhibits $\pi(\varepsilon, \delta)$-mode collapse.*

Theorem 1 fundamentally specifies the feasible region of three-dimensional representation $(u, \varepsilon, \delta)$, which we illustrate in Fig. 1(a) using a two-dimensional diagram that shows the respective feasible regions of $(u, \varepsilon)$ for different $\delta$. Based on Theorem 1, we then present the following corollaries:

**Corollary 1.** *The following claims can be deduced from Theorem 1:*

    *(i) Let $u^*(\varepsilon, \delta)$ be the maximum uniform diversity that can be achieved for a generative distribution that exhibits $(\varepsilon, \delta)$-mode collapse. Then $u^*(\varepsilon', \delta') < u^*(\varepsilon, \delta)$ holds for $\delta' > \delta$ and $\varepsilon' < \varepsilon$. Moreover, $\forall \xi > 0, \exists \mu, \nu > 0$, such that $u^*(\varepsilon, \delta) < \xi$ if $\varepsilon < \mu$ and $1 - \delta < \nu$.*

    *(ii) Let $q \in \mathcal{P}_{\mathcal{M}}$ be a generative distribution that exhibits $u$-mode collapse for any $u \in [0, 1)$, then for any $\delta \in (0, 1)$ and $\xi > 0$, $q$ can exhibit $(\varepsilon, \delta)$-mode collapse for $\delta - \varepsilon < \xi$.*

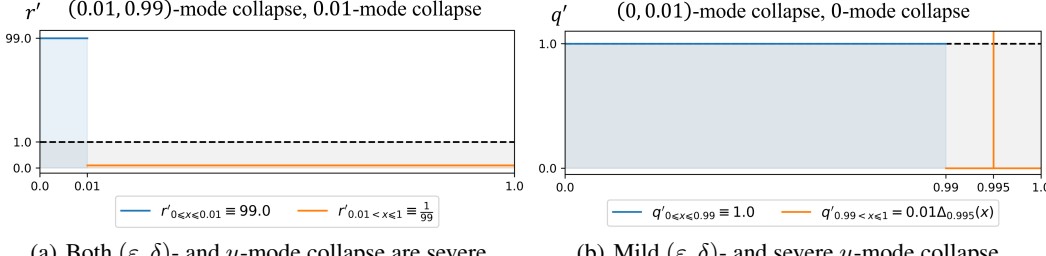

(a) Both $(\varepsilon, \delta)$- and $u$-mode collapse are severe.  (b) Mild $(\varepsilon, \delta)$- and severe $u$-mode collapse.

Figure 2: Illustrative distributions $r, q$ over domain $[0, 1]$. For $r$ and $q$, the corresponding PDFs $r'$ and $q'$ are plotted using colored solid lines, where the PDFs restricted to different sub-domains are represented by different colors for each distribution. The PDF $p'$ of the uniform distribution $p$ over $[0, 1]$ is plotted using black dashed line, where $p' \equiv 1$. In Fig. 2(b), we use $\Delta_{0.995}(x)$ to denote the Dirac delta function at $0.995$. The mode collapse exhibited by each distribution are indicated.

See supplementary for proof. We provide intuitive explanations to aforementioned claims. In terms of $(\varepsilon, \delta)$-mode collapse, larger $\delta$ and smaller $\varepsilon$ indicate more severe mode collapse [15], therefore, we can learn from Claim (i) that as $(\varepsilon, \delta)$-mode collapse becomes more severe, the maximum uniform diversity that can be achieved decreases, *i.e.*, $u$-mode collapse also becomes more severe. Moreover, Claim (i) states that $u$-mode collapse can be sufficiently severe (*i.e.*, $u \to 0$) as $(\varepsilon, \delta)$-mode collapse becomes sufficiently severe (*i.e.*, $\varepsilon \to 0, \delta \to 1$). An illustrative example is given in Fig. 1(a), where $P^u_{\varepsilon,\delta}$ are points in feasible regions of $(u, \varepsilon)$ given $\delta$ such that $P^u_{\varepsilon,\delta}$ gives the maximum $u$ that can be achieved for the given $(\varepsilon, \delta)$. We can intuitively see that as $\delta$ increases and $\varepsilon$ decreases, $u$ decreases for $P^u_{\varepsilon,\delta}$, and $u$ can be sufficiently small as $\delta \to 1$ and $\varepsilon \to 0$. Meanwhile, Claim (ii) indicates that distributions that exhibit severe $u$-mode collapse can exhibit mild $(\varepsilon, \delta)$-mode collapse (*i.e.*, $\varepsilon \approx \delta$), as illustrated by the point $Q^u_{\varepsilon,\delta}$ shown in Fig. 1(a). Hence in summary, the $u$-mode collapse is severe as long as the $(\varepsilon, \delta)$-mode collapse is severe, however, the $(\varepsilon, \delta)$-mode collapse could be mild while the $u$-mode collapse is severe. The relationship between $(\varepsilon, \delta)$-mode collapse and $u$-mode collapse can be depicted using the Venn diagram as shown in Fig. 1(b). Specifically, while severe $(\varepsilon, \delta)$-mode collapse is covered by severe $u$-mode collapse, the $(\varepsilon, \delta)$-mode collapse may not be able to identify severe $u$-mode collapse effectively. Therefore, $u$-mode collapse can be regarded as a generalization of $(\varepsilon, \delta)$-mode collapse, and is more robust and effective in capturing mode collapse.

We also provide distributions $r, q$ in Fig. 2 as illustrative examples of different mode collapses. Both $r$ and $q$ are defined over domain $[0, 1]$, *i.e.*, $r, q \in \mathcal{P}_{[0,1]}$. We take $r$ shown in Fig. 2(a) as an example that exhibits severe $(\varepsilon, \delta)$-mode collapse and severe $u$-mode collapse. By denoting $\mathcal{S} \triangleq [0.1, 1]$, we can observe that $r(\mathcal{S}) = 0.01$ meanwhile $p(\mathcal{S}) = 0.99$, namely $r$ exhibits $(\varepsilon, \delta)$-mode collapse for $\varepsilon = 0.01$ and $\delta = 0.99$, which corresponds to a severe $(\varepsilon, \delta)$-mode collapse since $\varepsilon \approx 0$ and $\delta \approx 1$. Regarding $u$-mode collapse, we can compute that $u = 0.01$, which corresponds to a severe $u$-mode collapse since $u \approx 0$. In Fig. 2(b), by denoting $\mathcal{S} \triangleq [0.99, 0.995) \cup (0.995, 1]$, we can observe that $q(\mathcal{S}) = 0$ meanwhile $p(\mathcal{S}) = 0.01$, hence $q$ exhibits $(\varepsilon, \delta)$-mode collapse for $\varepsilon = 0$ and $\delta = 0.01$. Regarding $u$-mode collapse, due to the infinity of Dirac delta function $\Delta_{0.995}(x)$ at $x = 0.995$, we can compute that $u = 0$, since $\mathcal{H}_q = -\infty$. Therefore, $q$ exhibits severe $u$-mode collapse but mild $(\varepsilon, \delta)$-mode collapse, since $\varepsilon \approx \delta \approx 0$ and $u = 0$.

## 3.2 The Generator Uniformity Property

We now relate the uniform diversity to the property of the generator which we refer to as *uniformity*. Let $\mathcal{Z} \subset \mathbb{R}^K$ be the latent space and $g : \mathcal{Z} \to \mathbb{R}^D$ be the generator with $K < D$. From the manifold hypothesis [32, 33], the image of $g$, $\mathcal{M} \triangleq g(\mathcal{Z}) = \{g(z) | z \in \mathcal{Z}\}$, is assumed to be a $K$-dimensional manifold residing in the ambient space $\mathbb{R}^D$, and $g$ pushes forward the prior distribution $p_z \in \mathcal{P}_\mathcal{Z}$ to a generative distribution $q \in \mathcal{P}_\mathcal{M}$ that is denoted by $q \triangleq g_\# p_z$. According to the definition of PDF, for $x = g(z)$ we have that

$$q'(x) = \frac{p'_z(z) \, \mathrm{d}V_\mathcal{Z}|_z}{\mathrm{d}V_\mathcal{M}|_x}, \tag{2}$$

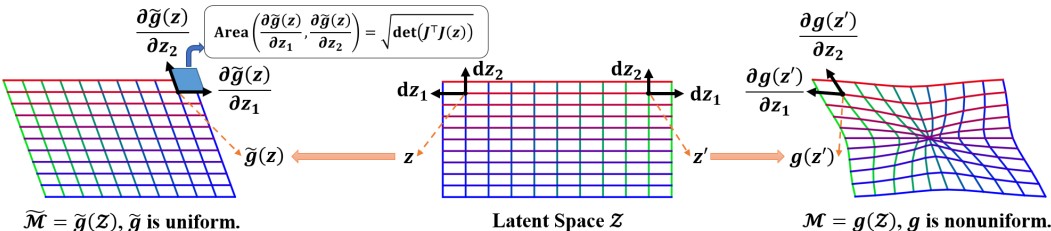

Figure 3: Intuitive illustrations of the generator uniformity. We set up two generators $g : \mathcal{Z} \to \mathcal{M}$ and $\widetilde{g} : \mathcal{Z} \to \widetilde{\mathcal{M}}$ with different uniformity, where both the latent space $\mathcal{Z}$ and the manifolds $\mathcal{M}, \widetilde{\mathcal{M}}$ are 2-dimensional. The coordinate lines of $\mathcal{Z}, \mathcal{M}, \widetilde{\mathcal{M}}$ are plotted, where the correspondence between $\mathcal{Z}$ and $\mathcal{M}$ (resp., $\widetilde{\mathcal{M}}$) for $g$ (resp., $\widetilde{g}$) is indicated by the colors of coordinate lines.

where $\mathrm{d}V_{\mathcal{Z}}|_z$ and $\mathrm{d}V_{\mathcal{M}}|_x$ are the *volume elements* [55] of $\mathcal{Z}$ at $z$ and $\mathcal{M}$ at $x$, respectively. Intuitively, the probability density is the ratio of the probability mass $p'_z(z)\,\mathrm{d}V_{\mathcal{Z}}|_z$ to the volume $\mathrm{d}V_{\mathcal{M}}|_x$ of the region where that probability mass is concentrated. Assume that $g$ is a local diffeomorphism [55], the volume element $\mathrm{d}V_{\mathcal{M}}|_x$ is given according to the Riemannian geometry [55] as

$$\mathrm{d}V_{\mathcal{M}}|_x = \sqrt{\det G(z)}\,\mathrm{d}V_{\mathcal{Z}}|_z, \tag{3}$$

where $G(z) \triangleq J_g^\top(z) J_g(z) \in \mathbb{R}^{K \times K}$ is the matrix form of the *Riemannian metric tensor* [55], and $J_g(z) \in \mathbb{R}^{D \times K}$ is the Jacobian of $g$ at $z$. Therefore, by plugging Eq. (3) into Eq. (2), we arrive at

$$q'(x) = \frac{p'_z(z)}{\sqrt{\det J_g^\top(z) J_g(z)}}, \quad x = g(z), \tag{4}$$

which is also known as the *change of variables formula* [36, 37, 35].

As aforementioned, the uniform diversity of a generative distribution $q \in \mathcal{P}_{\mathcal{M}}$ is maximized only if $q$ is the uniform distribution over $\mathcal{M}$, which is equivalent to $q'$ being constant over $\mathcal{M}$. For ease of analysis, we assume that $p_z$ is the uniform distribution over $\mathcal{Z}$, *i.e.*, $p'_z$ is constant over $\mathcal{Z}$. In such a case, it is obvious that $q'$ is constant over $\mathcal{M}$ only if $\det J_g^\top J_g$ is constant over $\mathcal{Z}$. On this account, we formalize the definition of the *generator uniformity* property as

**Definition 4** (Generator uniformity). *Given a generator $g : \mathcal{Z} \to \mathcal{M}$ that is a local diffeomorphism between $\mathcal{Z} \subset \mathbb{R}^K$ and $\mathcal{M} \subset \mathbb{R}^D$ with $K < D$, we say that $g$ is uniform, if $\det J_g^\top J_g$ is constant.*

Obviously, the connection between the generator uniformity and a maximized uniform diversity can be formalized as the following Theorem 2 (see supplementary for proof)

**Theorem 2.** *Assume that the generator $g : \mathcal{Z} \to \mathcal{M}$ is a local diffeomorphism between $\mathcal{Z} \subset \mathbb{R}^K$ and $\mathcal{M} \subset \mathbb{R}^D$ with $K < D$, and the prior distribution $p_z \in \mathcal{P}_{\mathcal{Z}}$ is the uniform distribution over $\mathcal{Z}$. The uniform diversity of the generative distribution $q = g_{\#}p_z$ is maximized, if $g$ is uniform.*

We provide intuitive illustrations for a better understanding of the generator uniformity property. In Fig. 3, we set up two generators $g, \widetilde{g}$ with different uniformity. The latent space $\mathcal{Z}$ is a 2-dimensional Euclidean subspace, and we divide $\mathcal{Z}$ into even grid blocks by drawing coordinate lines along axes $z_1$ and $z_2$, respectively. The generator $g$ (resp., $\widetilde{g}$) pushes forward coordinate lines of $\mathcal{Z}$ to those of $\mathcal{M}$ (resp., $\widetilde{\mathcal{M}}$). Since the prior distribution $p_z \in \mathcal{P}_{\mathcal{Z}}$ is the uniform distribution over $\mathcal{Z}$ and the area of each grid block is equal, we know that each grid block of $\mathcal{Z}$ contains the same probability mass. So each grid bock of $\mathcal{M}$ (resp., $\widetilde{\mathcal{M}}$) also contains the same probability mass, since the coordinate lines between $\mathcal{Z}$ and $\mathcal{M}$ (resp., $\widetilde{\mathcal{M}}$) correspond. For $\widetilde{\mathcal{M}}$, because the area of each grid block is equal, the ratio of probability mass to the area, *i.e.*, the probability density, for each grid block is equal too, *i.e.*, the generative distribution is the uniform distribution over $\widetilde{\mathcal{M}}$. However, the probability density for each grid blocks of $\mathcal{M}$ is not equal, since the area of each grid block is not equal. Therefore, given that each grid block contains the same probability mass, the equality of areas of grid blocks matters regarding the uniformity of the generative distribution, and the "equality of areas of grid blocks" is essentially an intuitive interpretation of the generator uniformity. We also explain how $\sqrt{\det J^\top J}$

relates to the area of a grid block. As shown in Fig. 3, for a grid block of $\widetilde{\mathcal{M}}$, if we consider $\rho_1 \triangleq \frac{\partial \widetilde{g}}{\partial z_1}$ and $\rho_2 \triangleq \frac{\partial \widetilde{g}}{\partial z_2}$ as two adjacent sides of that grid block, then the computation of

$$\det \left( J^\top J \right) = \begin{vmatrix} \|\rho_1\|^2 & \rho_1 \cdot \rho_2 \\ \rho_1 \cdot \rho_2 & \|\rho_2\|^2 \end{vmatrix} = \left( \|\rho_1\| \, \|\rho_2\| \sin\theta \right)^2 = \left( \mathrm{Area}\left( \rho_1, \rho_2 \right) \right)^2 \tag{5}$$

implies that $\sqrt{\det \left( J^\top J \right)}$ actually gives the area of the parallelogram spanned by $\rho_1$ and $\rho_2$, namely the area of the grid block, where $\theta$ is the angle between $\rho_1$ and $\rho_2$. Hence if $\det \left( J^\top J \right)$ is constant, the area of each grid block is equal, and the generator is uniform since the "equality of areas of grid blocks" is an intuitive interpretation of the generator uniformity, which coincides Definition 4.

### 3.3 The *UniGAN* Framework

We introduce our UniGAN framework for reducing mode collapse based on aforementioned analysis from a manifold view. Our goal is to maximize the uniform diversity of the generative distribution, which can be done by pursuing a uniform generator, as indicated by Theorem 2. However, to apply Theorem 2, we have to satisfy its prerequisite, *i.e.*, the generator $g$ should be a local diffeomorphism. Moreover, constraining $\det J_g^\top J_g$ to be constant involves not only computing the Jacobian $J_g$, but also computing the determinant, both of which are computationally expensive. Hence it is nontrivial to implement the generator uniformity constraint in practice.

Regarding the above problems, we propose to employ an NF-based generative model. On one hand, NFs are intrinsically bijections, therefore, the diffeomorphism prerequisite of the generator $g$ can be satisfied. On the other hand, NFs are specifically designed architectures with tractable determinant of Jacobian [35, 36, 37], which may ease the practical implementation of the constraint. Unfortunately, the above advantages of NFs are at the cost of maintaining a latent space of as high a dimensionality as the data space, which leads to over-parameterized models and very slow training. To overcome this problem, we propose to use an NF-based generator $g : \mathcal{Z} \subset \mathbb{R}^K \to \mathbb{R}^D$ for arbitrary $K < D$ with a hierarchical architecture as follows

$$g = f_L \circ e_L \circ f_{L-1} \circ e_{L-1} \circ \cdots \circ f_1 \circ e_1, \tag{6}$$

where $f_i : \mathbb{R}^{K_i} \to \mathbb{R}^{K_i}$ is a bijective and invertible NF module (*e.g.*, coupling layers [36, 37, 56]) with tractable determinant of Jacobian, $e_i : \mathbb{R}^{K_{i-1}} \to \mathbb{R}^{K_i}$ boosts the dimensionality of the input feature $x$ from $K_{i-1}$ to $K_i$ by simply padding $K_i - K_{i-1}$ zeros at the end of $x$ [57], and the dimensionalities of hierarchical features satisfy $K = K_0 < K_1 < \cdots < K_L = D$. See supplementary for a detailed introduction. However, a notable issue is that $\det J_g^\top J_g$ is intractable despite tractable determinants of Jacobian for $f_i$. Hence for ease of implementation, we turn to pursuing a special case of uniform generators where the $K$ singular values of $J_g$, $\{\sigma_j\}_{j=1}^K$, are equal and constant over $\mathcal{Z}$, considering the fact that $\det J_g^\top J_g = \sigma_1^2 \sigma_2^2 \cdots \sigma_K^2$. This is further realized by constraining the minimum and the maximum singular values of $J_g$ to be equal and constant over $\mathcal{Z}$. Therefore, the main concern is how to obtain the minimum and the maximum singular values of $J_g$.

We propose the *Linearized Transpose* (LT) technique to estimate the spectral norm of the Jacobian of a given network in a sample efficient manner. Due to space limitation, we use $\mathrm{LT}_g(z)$ to denote the estimated spectral norm of $J_g$ at $z \in \mathcal{Z}$, and the detailed introduction of our proposed LT is provided in supplementary. Since the spectral norm is essentially the maximum singular value, we know that $\mathrm{LT}_g(z)$ is the maximum singular value of $J_g$ at $z$. To obtain the minimum singular value of $J_g$, we propose to utilize the invertible property of $f_i$, which makes it tractable to obtain the pseudo inverse of $g$, $g^\dagger : \mathbb{R}^D \to \mathbb{R}^K$, defined as

$$g^\dagger = e_1^\dagger \circ f_1^{-1} \circ \cdots \circ e_{L-1}^\dagger \circ f_{L-1}^{-1} \circ e_L^\dagger \circ f_L^{-1}, \tag{7}$$

where $f_i^{-1} : \mathbb{R}^{K_i} \to \mathbb{R}^{K_i}$ is the inverse of $f_i$, and $e_i^\dagger : \mathbb{R}^{K_i} \to \mathbb{R}^{K_{i-1}}$ is the pseudo inverse of $e_i$ that reduces the dimensionality of the input feature $x$ from $K_i$ to $K_{i-1}$ by simply dropping $K_i - K_{i-1}$ elements at the end of $x$. Given $g, g^\dagger$, we can prove that (see supplementary)

**Proposition 1.** $\forall z \in \mathcal{Z}, J_{g^\dagger}(g(z)) J_g(z) = I \in \mathbb{R}^{K \times K}$ holds, where $I$ is the identity matrix.

Given the above Prop. 1, by denoting $\sigma^*(J)$ (*resp.*, $\sigma_*(J)$) as the maximum (*resp.*, the minimum) singular value of a given matrix $J$, we intuitively illustrate that $\sigma_*(J_g(z)) = \frac{1}{\sigma^*\left(J_{g^\dagger}(g(z))\right)}$ by using

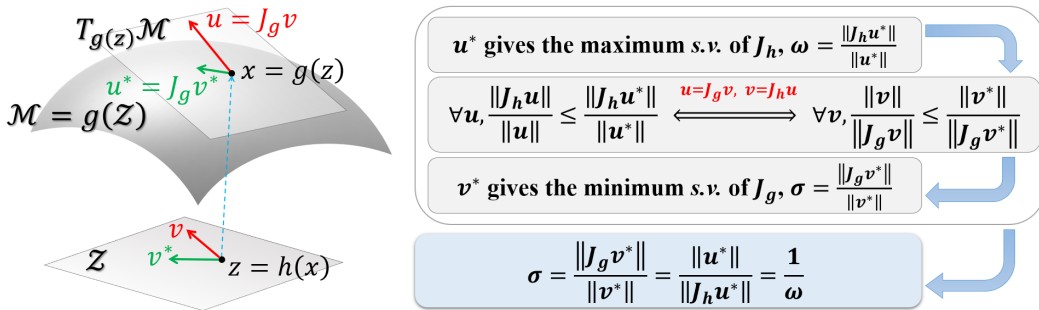

Figure 4: Illustration of the relationship between $\sigma_*(J_g)$ and $\sigma^*(J_h)$, where we use $h \triangleq g^\dagger$. Given that $u^* \in T_{g(z)}\mathcal{M}$ gives the maximum singular value of $J_h$, we have $\sigma_*(J_g) = \frac{1}{\sigma^*(J_h)}$.

Fig. 4. However, the illustration of Fig. 4 is only applicable in the case where the maximum singular value of $J_{g^\dagger}(g(z))$ can be given by a vector $u^* \in T_{g(z)}\mathcal{M} \subset \mathbb{R}^D$ that is on the tangent space of $\mathcal{M}$ at $g(z)$, see supplementary for illustration. To tackle this problem, we observe that by encouraging $\frac{1}{\sigma^*(J_{g^\dagger}(g(z)))} = \sigma^*(J_g(z))$, the maximum singular value of $J_{g^\dagger}(g(z))$ can be given by a vector $u^* \in T_{g(z)}\mathcal{M}$. To see this, assume that $\widetilde{u}^* \in \mathbb{R}^D$ gives the maximum singular value of $J_{g^\dagger}(g(z))$, we have that $\frac{1}{\sigma^*(J_{g^\dagger})} = \frac{\|\widetilde{u}^*\|}{\|J_{g^\dagger}\widetilde{u}^*\|} \leqslant \frac{\|J_g J_{g^\dagger}\widetilde{u}^*\|}{\|J_{g^\dagger}J_g J_{g^\dagger}\widetilde{u}^*\|} = \frac{\|J_g J_{g^\dagger}\widetilde{u}^*\|}{\|J_{g^\dagger}\widetilde{u}^*\|} \leqslant \sigma^*(J_g)$, where the first equality holds if and only if $u^* \triangleq J_g J_{g^\dagger}\widetilde{u}^* \in T_{g(z)}\mathcal{M} \subset \mathbb{R}^D$ gives the maximum singular value of $J_{g^\dagger}$. So in the case of $\frac{1}{\sigma^*(J_{g^\dagger}(g(z)))} = \sigma^*(J_g(z))$, we have $\sigma_*(J_g(z)) = \frac{1}{\sigma^*(J_{g^\dagger}(g(z)))}$ from Fig. 4, and hence $\sigma_*(J_g(z)) = \sigma^*(J_g(z))$, namely the minimum and the maximum singular values of $J_g$ are equal. Therefore, by letting $a$ be the moving average of the maximum and the minimum singular values of $J_g$, the regularization for the generator uniformity implemented in practice is formalized as

$$\mathcal{L}_{\text{gunif}} \triangleq \mathbb{E}_{z \sim p_z}\left\{ \left(\text{LT}_g(z) - a\right)^2 + \left(\frac{1}{\text{LT}_{g^\dagger}(g(z))} - a\right)^2 \right\}, \tag{8}$$

which is a simple yet sample efficient regularization due to the superior sample efficiency of LT (see supplementary). We then combine the above NF-based generator and a normal discriminator into a generative framework augmented using our generator uniformity regularization $\mathcal{L}_{\text{gunif}}$, obtaining our UniGAN whose objective function is as follows,

$$\mathcal{L}_{\text{unigan}} = \mathcal{L}_{\text{gan}} + \lambda_{\text{gunif}}\mathcal{L}_{\text{gunif}}, \tag{9}$$

where $\mathcal{L}_{\text{gan}}$ is the original objective of GANs, and $\lambda_{\text{gunif}}$ is the balancing hyper-parameter. Eq. (9) also implies that $\mathcal{L}_{\text{gunif}}$ can be easily integrated into any models by appending $\mathcal{L}_{\text{gunif}}$ to the original objective function, and our UniGAN can be adapted to any other generative framework by replacing $\mathcal{L}_{\text{gan}}$ with the corresponding objective function.

## 4 Experiments

We show through experimental results that our proposed UniGAN framework is effective in learning a uniform generator and improving uniform diversity of the generative distribution. To demonstrate the adaptability of our UniGAN framework, we adapt UniGAN for a variety of existing models that reduce mode collapse, including GAN [1], BiGAN [3]/ALI [4], MSGAN [40, 26], VEEGAN [38], PacGAN [15], MDGAN [34] and RegGAN [34]. We also show that these models are inferior to ours regarding uniform diversity. For intuitive demonstration, we firstly provide results on a toy synthetic data manifold, then provide results on simple datasets including MNIST [58], FashionMNIST [59] and their colored version [22], and CIFAR10 [60]. We also provide results on natural image datasets including CelebA [61], FFHQ [62], AFHQ [63] and LSUN [64]. To implement the computation of uniform diversity of a given set of generative samples in practice, we also propose a diversity metric named *udiv*. Due to space limitation, the implementation details about our proposed *udiv* metric and experiments can be found in supplementary.

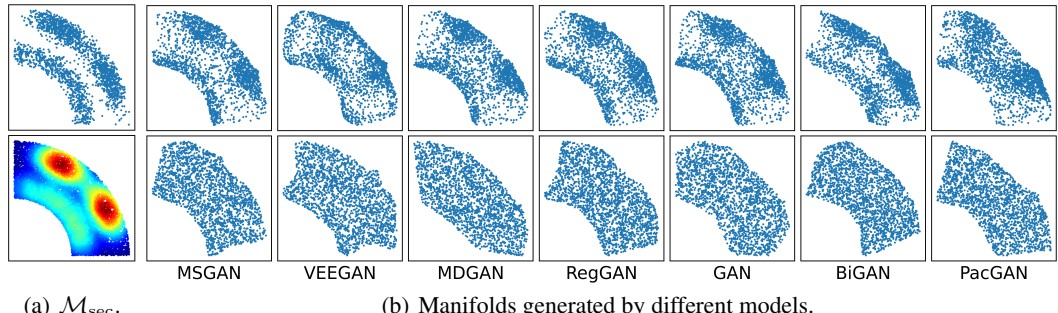

(a) $\mathcal{M}_{\text{sec}}$.  (b) Manifolds generated by different models.

Figure 5: For real samples $\mathcal{M}_{\text{sec}}$ in Fig. 5(a), we visualize training data in the top image, and show the three major modes using heatmap as in the bottom image. We visualize manifolds generated by different models in Fig. 5(b), where the top row visualizes results of baseline models, and the bottom row visualizes results of UniGAN adapted to the corresponding baseline models.

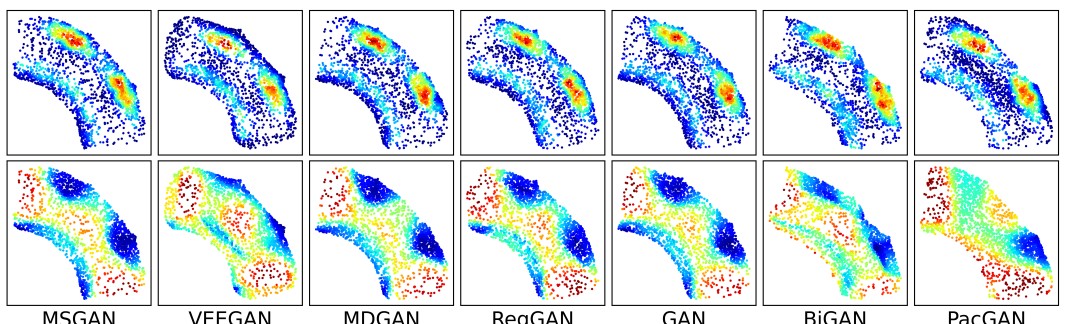

Figure 6: Visualization of probability densities (top row) and $\log \det J^\top J$ (bottom row) of samples generated by baseline methods. The probability densities are estimated by the method we introduced in the *udiv* metric, see supplementary. Magnitudes of values are reflected by using heatmap.

We provide results in this section on synthetic datasets for the sake of intuitive understanding, while leaving more results on other datasets in supplementary due to space limitation. We use a 2D synthetic dataset *sector* which samples points from a sector data manifold $\left\{ (x,y) \,|\, 1 \leqslant \sqrt{x^2 + y^2} \leqslant 2 \right\}$ in the first quadrant. We set up three major modes to simulate unbalanced distribution of modes in real data. Specifically, we use $\mu_r = 1.2, \sigma_r = 0.1$ to obtain $\mathcal{R}_1 \triangleq \left\{ r^{(i)} \sim \mathcal{N}(\mu_r, \sigma_r) \,|\, r^{(i)} \in [1,2] \right\}_{i=1}^{N}$, and use $\mu_\theta = \pi/4, \sigma_\theta = \pi/2$ to obtain $\Theta_1 \triangleq \left\{ \theta^{(i)} \sim \mathcal{N}(\mu_\theta, \sigma_\theta) \,|\, \theta^{(i)} \in [0, \pi/2] \right\}_{i=1}^{N}$, and then perform polar transform, obtaining $\mathcal{M}_1 \triangleq \left\{ \left( x^{(i)}, y^{(i)} \right) \right\}_{i=1}^{N}$ with $x^{(i)} = r^{(i)} \cos \theta^{(i)}$ and $y^{(i)} = r^{(i)} \sin \theta^{(i)}$ as the first mode. Similarly, we use $\mu_r = 1.8, \sigma_r = 0.1$ and $\mu_\theta = 3\pi/20, \sigma_\theta = \pi/20$ to obtain the second mode $\mathcal{M}_2$, and use $\mu_r = 1.8, \sigma_r = 0.1$ and $\mu_\theta = 7\pi/20, \sigma_\theta = \pi/20$ to obtain the third mode $\mathcal{M}_3$. Finally, we obtain $\mathcal{M}_{\text{sec}} \triangleq \cup_{i=1}^{3} \mathcal{M}_i$. See Fig. 5(a).

We train baseline models and the corresponding UniGAN adaptations on the proposed 2D synthetic manifold *sector*. The manifolds generated by different models are intuitively visualized in Fig. 5(b). From Fig. 5(b), we learn that for almost every baseline model, their generative samples lack uniform diversity because we can intuitively observe that they distribute nonuniformly. Meanwhile, samples generated by our UniGAN consistently distribute uniformly, hence our UniGAN outperforms baseline models in terms of uniform diversity, and can better interpolate between training samples on the data manifold compared to baseline models. We also provide quantitative comparison between baselines and our UniGAN in terms of *udiv* and generator uniformity as shown in Tbl. 1. We see that compared to baseline models, our UniGAN achieves a much higher *udiv* score and maintains lower variance of $\log \det J^\top J$, therefore, our UniGAN consistently outperforms baseline models in terms of generator uniformity and uniform diversity of generative samples.

Table 1: Quantitative comparison between different models on the uniform diversity and generator uniformity for the $\mathcal{M}_{\mathrm{sec}}$ manifold. The uniform diversity and generator uniformity is measured by the *udiv* score and the *gunif* score, respectively, where $\uparrow$ (*resp.*, $\downarrow$) indicates that larger (*resp.*, lower) values are better. The *gunif* metric is gives as the *Coefficient of Variation* (CV) of $\log \det J^\top J$ over latent space, which is a standardized measure of dispersion of a distribution. For each metric, the first row and the second row give the results of baseline models and UniGAN adapted to the corresponding baselines, respectively, and we also provide the $p$-value of significant test between results of baselines and the corresponding UniGAN to avoid randomness in the third row. For each baseline model, the better value among the baseline result and the corresponding UniGAN result is shown in bold.

| Metric | MSGAN | VEEGAN | MDGAN | RegGAN | GAN | BiGAN | PacGAN |
|---|---|---|---|---|---|---|---|
| *udiv* $\uparrow$ | 0.46 | 0.82 | 0.52 | 0.75 | 0.50 | 0.80 | 0.78 |
| | **0.91** | **0.93** | **0.93** | **0.91** | **0.90** | **0.93** | **0.93** |
| | $2.07e^{-5}$ | $1.42e^{-10}$ | $8.62e^{-4}$ | $3.59e^{-3}$ | $6.43e^{-3}$ | $1.26e^{-4}$ | $2.90e^{-6}$ |
| *gunif* $\downarrow$ | 0.25 | 0.20 | 0.25 | 0.22 | 0.26 | 0.20 | 0.21 |
| | **0.01** | **0.02** | **0.02** | **0.02** | **0.01** | **0.01** | **0.02** |
| | $1.65e^{-9}$ | $7.69e^{-13}$ | $4.02e^{-9}$ | $1.61e^{-7}$ | $6.91e^{-9}$ | $8.37e^{-9}$ | $6.46e^{-9}$ |

Table 2: Correlations between probability densities and the corresponding $\log \det J^\top J$ values for each baseline model. The first and second rows represent the mean and the $90\%$ confidence interval of correlations over 10 runs with random initialization, respectively.

| Metric | MSGAN | VEEGAN | MDGAN | RegGAN | GAN | BiGAN | PacGAN |
|---|---|---|---|---|---|---|---|
| *corr* | $-0.80$ | $-0.85$ | $-0.81$ | $-0.82$ | $-0.82$ | $-0.85$ | $-0.83$ |
| | $\pm 0.08$ | $\pm 0.02$ | $\pm 0.10$ | $\pm 0.05$ | $\pm 0.09$ | $\pm 0.10$ | $\pm 0.15$ |

We further experimentally demonstrate that the uniform diversity is closely related with the generator uniformity, which justifies our theoretical analysis in Sec. 3.2. Specifically, in Fig. 6, we visualize the correspondences between the values of $\log \det J^\top J$ and the probability densities on the generated manifold for different models trained on the 2D synthetic dataset *sector*. From Fig. 6, we learn that the probability densities of generated samples and the corresponding values of $\log \det J^\top J$ are inversely correlated, *i.e.*, the higher the probability density, the lower the values of $\log \det J^\top J$. Quantitative results provided in Tbl. 2 show that the correlation coefficients *corr* between the probability densities and the values of $\log \det J^\top J$ are lower than $-0.8$ across all baselines, which indicates that they are strongly negatively correlated and coincides with the intuitive illustration of the generator uniformity in Fig. 3 and the analysis on the relationship between the generator uniformity and uniform diversity.

# 5   Conclusion

In this paper, we propose the *uniform diversity*, which is associated with a new type of mode collapse named *u-mode collapse* where generative samples distribute nonuniformly over the data manifold, and is closely related with the *generator uniformity* property. To maximize the uniform diversity, we propose a simple yet effective generative framework named *UniGAN* with an NF-based generator and a sample efficient regularization on the generator uniformity, which can be adapted to any other framework. A new type of diversity metric named *udiv* is also proposed to estimate uniform diversity in practice. Experimental results verify the effectiveness of our UniGAN framework.

## Acknowledgements

The work was supported by the National Science Foundation of China (62076162), and the Shanghai Municipal Science and Technology Major/Key Project, China (2021SHZDZX0102, 20511100300).

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
