# Supplementary for UniGAN: Reducing Mode Collapse in GANs using a Uniform Generator

**Ziqi Pan, Li Niu,*, Liqing Zhang***
MoE Key Lab of Artificial Intelligence
Department of Computer Science and Engineering
Shanghai Jiao Tong University, Shanghai, China
{panziqi_ai, ustcnewly}@sjtu.edu.cn, zhang-lq@cs.sjtu.edu.cn

## 1 Proofs

We prove Theorem 1 (restated as Theorem 1) and its Corollary 1 (restated as Corollary 1), Theorem 2 (restated as Theorem 2) and Proposition 1 (restated as Proposition 1) in main text.

### 1.1 Proof of Theorem 1

Same as main text, let $\mathcal{P}_\mathcal{M}$ be the set of all distributions over a finite manifold $\mathcal{M}$. Given a distribution $p \in \mathcal{P}_\mathcal{M}$, we use $p'$ to denote its PDF. As we mentioned in main text, the differential entropy of $q$, $\mathcal{H}_q$, is maximized (*i.e.*, $\mathcal{H}_q = \log m_\mathcal{M}$) only if $q$ is the uniform distribution over $\mathcal{M}$. This can be seen directly from the following Jensen inequality due to the concavity of the $\log$ function:

$$\mathcal{H}_q = \int_{\mathrm{supp}(q')} \log \frac{1}{q'(x)} \mathrm{d}q(x) \leqslant \log \left( \int_{\mathrm{supp}(q')} \frac{1}{q'(x)} \mathrm{d}q(x) \right) = \log m_{\mathrm{supp}(q')} \leqslant \log m_\mathcal{M}, \tag{1}$$

where $m_{\mathrm{supp}(q')}$ and $m_\mathcal{M}$ is the measure of $\mathrm{supp}(q') \subset \mathcal{M}$ and $\mathcal{M}$, respectively, and the equality holds if and only if $q'$ is constant almost everywhere on $\mathrm{supp}(q')$, and $\mathrm{supp}(q')$ is almost equal to $\mathcal{M}$. We then prove the following Theorem 1.

**Theorem 1.** *For $q \in \mathcal{P}_\mathcal{M}$ that exhibits $(\varepsilon, \delta)$-mode collapse, $0 \leqslant \mathcal{U}_q \leqslant \pi(\varepsilon, \delta)$, where $\pi(\varepsilon, \delta) \triangleq \left(\frac{\delta}{\varepsilon}\right)^\varepsilon \left(\frac{1-\delta}{1-\varepsilon}\right)^{(1-\varepsilon)} \in [0, 1)$. That is, a generative distribution that exhibits $(\varepsilon, \delta)$-mode collapse at least exhibits $\pi(\varepsilon, \delta)$-mode collapse.*

*Proof.* From Lemma 1, we learn that $\mathcal{H}_q$ is maximized if and only if $q'$ is constant almost everywhere on $\mathrm{supp}(q') \cap \mathcal{S}$ and $q'$ is constant almost everywhere on $\mathrm{supp}(q') \cap \mathcal{S}'$. In such a case, let $q'(x) \equiv \xi$ for almost every $x \in \mathrm{supp}(q') \cap \mathcal{S}$ and $q'(x) \equiv \xi'$ for almost every $x \in \mathrm{supp}(q') \cap \mathcal{S}'$, then by plugging $q(\mathcal{S}) = \int_{\mathrm{supp}(q') \cap \mathcal{S}} \mathrm{d}q(x) = \xi m$ and $q(\mathcal{S}') = \int_{\mathrm{supp}(q') \cap \mathcal{S}'} \mathrm{d}q(x) = \xi' m'$ into Eq. (18), we have that

$$\mathcal{H}_q \leqslant q(\mathcal{S}) \log \frac{m}{q(\mathcal{S})} + q(\mathcal{S}') \log \frac{m'}{q(\mathcal{S}')} = \xi m \log \frac{1}{\xi} + \xi' m' \log \frac{1}{\xi'} \triangleq E(\xi, \xi', m, m'), \tag{2}$$

**Construction of a linear programming problem** Since $p(\mathcal{S}) = \int_{\mathrm{supp}(q') \cap \mathcal{S}} \mathrm{d}p(x) = \frac{m}{m_\mathcal{M}} \geqslant \delta$, we have that $\delta m_\mathcal{M} \leqslant m \leqslant m_\mathcal{M}$. We also know that $0 \leqslant m' \leqslant m_\mathcal{M}$ and $m + m' \leqslant m_\mathcal{M}$. Since $q(\mathcal{S}) = \xi m \leqslant \varepsilon$, we have that $\xi \leqslant \frac{\varepsilon}{m}$. Since $q(\mathcal{S}) + q(\mathcal{S}') = 1$, we have that $\xi m + \xi' m' = 1$.

---

*Corresponding author

36th Conference on Neural Information Processing Systems (NeurIPS 2022).

Therefore, we consider the following linear programming problem:

$$\max E\left(\xi, \xi', m, m'\right), \quad \text{s.t.} \tag{3}$$

$$\delta m_{\mathcal{M}} \leqslant m \leqslant m_{\mathcal{M}}, \quad 0 \leqslant m' \leqslant m_{\mathcal{M}}, \quad m + m' \leqslant m_{\mathcal{M}}, \quad 0 \leqslant \xi \leqslant \frac{\varepsilon}{m}, \quad \xi m + \xi' m' = 1. \tag{4}$$

**Solving the linear programming problem** Considering two sets of solutions $(\xi, \xi'_1, m, m'_1)$ and $(\xi, \xi'_2, m, m'_2)$, where $m'_2 \geqslant m'_1$. According to constraint (4), we have that $\xi'_1 m'_1 = \xi'_2 m'_2$, which implies that $\xi'_1 \geqslant \xi'_2$. Therefore, we have that

$$E\left(\xi, \xi'_2, m, m'_2\right) - E\left(\xi, \xi'_1, m, m'_1\right) = \xi'_1 m'_1 \left(\log \frac{1}{\xi'_2} - \log \frac{1}{\xi'_1}\right) \geqslant 0, \tag{5}$$

hence according to constraint (4), the solution $(\xi, \xi', m, m')$ to the linear programming problem (Eq. (3)) satisfies $m + m' = m_{\mathcal{M}}$. On the other hand, according to constraint $\xi m + \xi' m' = 1$, we can rewrite the objective as $F\left(\xi, m, m'\right) \triangleq E\left(\xi, \frac{1-\xi m}{m'}, m, m'\right)$. Since $\left(\log \frac{1}{\xi}\right)' = -\frac{1}{\xi}$,

$$\frac{\partial F}{\partial \xi} = m\left(\log \frac{1}{\xi} - 1\right) - \frac{m}{m'} m'\left(\log \frac{1}{\xi'} - 1\right) = m\left(\log \frac{1}{\xi} - \log \frac{1}{\xi'}\right). \tag{6}$$

Since

$$m'\left(\xi' - \xi\right) = 1 - \left(m + m'\right)\xi = 1 - m_{\mathcal{M}}\xi \geqslant 1 - \frac{\varepsilon m_{\mathcal{M}}}{m} = 1 - \frac{\varepsilon}{p\left(\mathcal{S}\right)} > 0, \tag{7}$$

we have that $\xi' > \xi$, i.e., $\frac{\partial F}{\partial \xi} > 0$. Hence according to constraint (4), the solution $(\xi, \xi', m, m')$ to the linear programming problem (Eq. (3)) satisfies $\xi = \frac{\varepsilon}{m}$. Based on the above analysis, we have the following equivalent linear programming problem to Eq. (3) after simplifying

$$\max E\left(\xi, \xi', m, m'\right), \quad \text{s.t.} \tag{8}$$

$$\delta m_{\mathcal{M}} \leqslant m \leqslant m_{\mathcal{M}}, \quad m + m' = m_{\mathcal{M}}, \quad \xi = \frac{\varepsilon}{m}, \quad \xi m + \xi' m' = 1. \tag{9}$$

Therefore, the above linear programming problem (Eq. (8)) is equivalent to the maximum problem of the following function $G\left(m\right) \triangleq E\left(\frac{\varepsilon}{m}, \frac{1-\varepsilon}{m_{\mathcal{M}}-m}, m, m_{\mathcal{M}} - m\right)$,

$$\max G\left(m\right), \quad \delta m_{\mathcal{M}} \leqslant m \leqslant m_{\mathcal{M}}. \tag{10}$$

For $G'\left(m\right)$, according to the chain rule of derivatives, we have that

$$G'\left(m\right) = \frac{\partial E}{\partial \xi} \frac{\mathrm{d}\xi}{\mathrm{d}m} + \frac{\partial E}{\partial \xi'} \frac{\mathrm{d}\xi'}{\mathrm{d}m} + \frac{\partial E}{\partial m} + \frac{\partial E}{\partial m'} \frac{\mathrm{d}m'}{\mathrm{d}m} \tag{11}$$

$$= m\left(\log \frac{1}{\xi} - 1\right)\left(-\frac{\varepsilon}{m^2}\right) + m'\left(\log \frac{1}{\xi'} - 1\right)\frac{1-\varepsilon}{\left(m_{\mathcal{M}} - m\right)^2} + \xi \log \frac{1}{\xi} - \xi' \log \frac{1}{\xi'} \tag{12}$$

$$= -\xi\left(\log \frac{1}{\xi} - 1\right) + \xi'\left(\log \frac{1}{\xi'} - 1\right) + \xi \log \frac{1}{\xi} - \xi' \log \frac{1}{\xi'} = \xi - \xi' < 0, \tag{13}$$

therefore, we know that $G\left(m\right)$ is maximized at $m = \delta m_{\mathcal{M}}$, and the maximum of $G\left(m\right)$ is given as

$$G\left(\delta m_{\mathcal{M}}\right) = E\left(\frac{\varepsilon}{\delta m_{\mathcal{M}}}, \frac{1-\varepsilon}{\left(1-\delta\right) m_{\mathcal{M}}}, \delta m_{\mathcal{M}}, \left(1-\delta\right) m_{\mathcal{M}}\right) \tag{14}$$

$$= \varepsilon \log \frac{\delta m_{\mathcal{M}}}{\varepsilon} + \left(1-\varepsilon\right) \log \frac{\left(1-\delta\right) m_{\mathcal{M}}}{1-\varepsilon} = \varepsilon \log \frac{\delta}{\varepsilon} + \left(1-\varepsilon\right) \log \frac{1-\delta}{1-\varepsilon} + \log m_{\mathcal{M}} \tag{15}$$

$$= \log\left\{\left(\frac{\delta}{\varepsilon}\right)^{\varepsilon} \left(\frac{1-\delta}{1-\varepsilon}\right)^{\left(1-\varepsilon\right)} m_{\mathcal{M}}\right\}. \tag{16}$$

By denoting $\pi\left(\varepsilon, \delta\right) \triangleq \left(\frac{\delta}{\varepsilon}\right)^{\varepsilon} \left(\frac{1-\delta}{1-\varepsilon}\right)^{\left(1-\varepsilon\right)}$, we have that $\mathcal{H}_q \leqslant \log\left(\pi\left(\varepsilon, \delta\right) m_{\mathcal{M}}\right)$.

**Conclusion** Obviously, $\pi(\varepsilon, \delta) \geqslant 0$, and we can verify that $\pi(\varepsilon, \delta) = 0$ for $\varepsilon = 0$. Since $\frac{\delta}{\varepsilon} \neq \frac{1-\delta}{1-\varepsilon}$, according to Jensen inequality, we have that

$$\log \pi(\varepsilon, \delta) = \varepsilon \log \frac{\delta}{\varepsilon} + (1-\varepsilon) \log \frac{1-\delta}{1-\varepsilon} < \log \left( \varepsilon \frac{\delta}{\varepsilon} + (1-\varepsilon) \frac{1-\delta}{1-\varepsilon} \right) = 0, \quad (17)$$

namely $0 \leqslant \pi(\varepsilon, \delta) < 1$. For $q$ that exhibits $(\varepsilon, \delta)$-mode collapse, as aforementioned we know that $\mathcal{H}_q \in (-\infty, \log(\pi(\varepsilon, \delta) m_{\mathcal{M}})]$, hence $0 \leqslant \mathcal{U}_q \leqslant \pi(\varepsilon, \delta) < 1$, which completes the proof. $\quad\square$

**Lemma 1.** *Given a distribution $q \in \mathcal{P}_{\mathcal{M}}$ that exhibits $(\varepsilon, \delta)$-mode collapse on $\mathcal{S} \subset \mathcal{M}$, namely $p(\mathcal{S}) \geqslant \delta$ and $q(\mathcal{S}) \leqslant \varepsilon$ where $p$ is the ground-truth data distribution, we have that*

$$\mathcal{H}_q \leqslant q(\mathcal{S}) \log \frac{m}{q(\mathcal{S})} + q(\mathcal{S}') \log \frac{m'}{q(\mathcal{S}')}, \quad (18)$$

*where $\mathcal{S}'$ is the complementary set of $\mathcal{S}$ with respect to $\mathcal{M}$, $m \triangleq m_{\mathrm{supp}(q') \cap \mathcal{S}}$ and $m' \triangleq m_{\mathrm{supp}(q') \cap \mathcal{S}'}$, and the equality holds if and only if $q'$ is constant almost everywhere on $\mathrm{supp}(q') \cap \mathcal{S}$ and $q'$ is constant almost everywhere on $\mathrm{supp}(q') \cap \mathcal{S}'$.*

*Proof.* Given that

$$\mathcal{H}_q = \int_{\mathrm{supp}(q')} \log \frac{1}{q'(x)} \mathrm{d}q(x) = \underbrace{\int_{\mathrm{supp}(q') \cap \mathcal{S}} \log \frac{1}{q'(x)} \mathrm{d}q(x)}_{①} + \underbrace{\int_{\mathrm{supp}(q') \cap \mathcal{S}'} \log \frac{1}{q'(x)} \mathrm{d}q(x)}_{②}, \quad (19)$$

we can derive according to the Jensen inequality that

$$① = q(\mathcal{S}) \int_{\mathrm{supp}(q') \cap \mathcal{S}} \log \frac{1}{q'(x)} \mathrm{d} \frac{q(x)}{q(\mathcal{S})} \leqslant q(\mathcal{S}) \log \left( \int_{\mathrm{supp}(q') \cap \mathcal{S}} \frac{1}{q'(x)} \mathrm{d} \frac{q(x)}{q(\mathcal{S})} \right) \quad (20)$$

$$= q(\mathcal{S}) \log \left( \frac{1}{q(\mathcal{S})} \int_{\mathrm{supp}(q') \cap \mathcal{S}} \frac{1}{q'(x)} \mathrm{d}q(x) \right) = q(\mathcal{S}) \log \frac{m}{q(\mathcal{S})}, \quad (21)$$

where the equality holds if and only if $q'$ is constant almost everywhere on $\mathrm{supp}(q') \cap \mathcal{S}$. Similarly, we have that

$$② \leqslant q(\mathcal{S}') \log \frac{m'}{q(\mathcal{S}')}, \quad (22)$$

where the equality holds if and only if $q'$ is constant almost everywhere on $\mathrm{supp}(q') \cap \mathcal{S}'$. Combining Eq. (20)-(22) completes the proof. $\quad\square$

### 1.2 Proof of Corollary 1

We prove Corollary 1 in main text, which is restated as the following Corollary 1:

**Corollary 1.** *The following claims can be deduced from Theorem 1:*

*(i) Let $u^*(\varepsilon, \delta)$ be the maximum uniform diversity that can be achieved for a generative distribution that exhibits $(\varepsilon, \delta)$-mode collapse. Then $u^*(\varepsilon', \delta') < u^*(\varepsilon, \delta)$ holds for $\delta' > \delta$ and $\varepsilon' < \varepsilon$. Moreover, $\forall \xi > 0, \exists \mu, \nu > 0$, such that $u^*(\varepsilon, \delta) < \xi$ if $\varepsilon < \mu$ and $1 - \delta < \nu$.*

*(ii) Let $q \in \mathcal{P}_{\mathcal{M}}$ be a generative distribution that exhibits $u$-mode collapse for any $u \in [0, 1)$, then for any $\delta \in (0, 1)$ and $\xi > 0$, $q$ can exhibit $(\varepsilon, \delta)$-mode collapse for $\delta - \varepsilon < \xi$.*

*Proof.* From Theorem 1, we know that $u^*(\varepsilon, \delta) = \pi(\varepsilon, \delta)$. By denoting

$$F(\varepsilon, \delta) \triangleq \log \pi(\varepsilon, \delta) = \varepsilon \log \frac{\delta}{\varepsilon} + (1-\varepsilon) \log \frac{1-\delta}{1-\varepsilon} \quad (23)$$

and using the fact that $\varepsilon < \delta$ and $1 - \varepsilon > 1 - \delta$, we have that

$$\frac{\partial F}{\partial \varepsilon} = \log \frac{\delta}{\varepsilon} - \log \frac{1-\delta}{1-\varepsilon} > 0, \quad (24)$$

$$\frac{\partial F}{\partial \delta} = \frac{\varepsilon}{\delta} - \frac{1-\varepsilon}{1-\delta} < 0. \quad (25)$$

Therefore, for $\delta' > \delta$ and $\varepsilon' < \varepsilon$, we have that $F(\varepsilon', \delta') < F(\varepsilon, \delta)$, namely $u^*(\varepsilon', \delta') < u^*(\varepsilon, \delta)$. Given $0 < \xi < 1$, we can verify that $\exists \mu = 1, \nu = 1 - \xi$, such that for $\varepsilon < \mu$ and $1 - \delta < \nu$,

$$\pi(\varepsilon, \delta) < \pi(\mu, 1 - \nu) = \pi(1, \xi) = \xi. \tag{26}$$

On the other hand, $\pi(\varepsilon, \delta) < \xi$ always holds for $\xi \geqslant 1$. Hence the proof of Claim (i) is completed. To Claim (ii), we use contradictory. Assume that $\exists u \in [0, 1), \delta \in (0, 1), \xi > 0$, such that $\forall \varepsilon > \delta - \xi, q$ cannot exhibit $(\varepsilon, \delta)$-mode collapse, namely $u > \pi(\varepsilon, \delta)$. However, we can verify that $\pi(\varepsilon, \delta) \to 1$ as $\varepsilon \to \delta$, namely $\exists \varepsilon > \delta - \xi$, such that $\pi(\varepsilon, \delta) > u$, which contradicts the assumption. Therefore, the proof of Claim (ii) is completed. $\qquad \square$

### 1.3 Proof of Theorem 2

We prove Theorem 2 in main text, which is restated as the following Theorem 2:

**Theorem 2.** *Assume that the generator $g : \mathcal{Z} \to \mathcal{M}$ is a local diffeomorphism between $\mathcal{Z} \subset \mathbb{R}^K$ and $\mathcal{M} \subset \mathbb{R}^D$ with $K < D$, and the prior distribution $p_z \in \mathcal{P}_{\mathcal{Z}}$ is the uniform distribution over $\mathcal{Z}$. The uniform diversity of the generative distribution $q = g_{\#} p_z$ is maximized, if $g$ is uniform.*

*Proof.* The uniform diversity, $\mathcal{U}_q = \frac{e^{\mathcal{H}_q}}{m_{\mathcal{M}}}$, of the generative distribution $q$ is maximized, if and only if $\mathcal{H}_q$ is maximized. For $z \in \mathcal{Z}$, let $x = g(z)$, we have that

$$\mathcal{H}_q = -\int_{\text{supp}(q')} \log q'(x) \, dq(x) \tag{27}$$

$$= -\int_{\text{supp}(q')} \frac{p'_z(z)}{\sqrt{\det J_g^\top(z) J_g(z)}} \log \frac{p'_z(z)}{\sqrt{\det J_g^\top(z) J_g(z)}} dV_{\mathcal{M}}|_x \tag{28}$$

$$= -\int_{\mathcal{Z}} \frac{p'_z(z)}{\sqrt{\det J_g^\top(z) J_g(z)}} \log \frac{p'_z(z)}{\sqrt{\det J_g^\top(z) J_g(z)}} \left( \sqrt{\det J_g^\top(z) J_g(z)} dV_{\mathcal{Z}}|_z \right) \tag{29}$$

$$= -\int_{\mathcal{Z}} p'_z(z) \log \frac{p'_z(z)}{\sqrt{\det J_g^\top(z) J_g(z)}} dV_{\mathcal{Z}}|_z \tag{30}$$

$$= -\int_{\mathcal{Z}} p'_z(z) \log p'_z(z) \, dV_{\mathcal{Z}}|_z + \int_{\mathcal{Z}} p'_z(z) \log \sqrt{\det J_g^\top(z) J_g(z)} dV_{\mathcal{Z}}|_z \tag{31}$$

$$= \mathcal{H}_{p_z} + p'_z \int_{\mathcal{Z}} \log \sqrt{\det J_g^\top(z) J_g(z)} dV_{\mathcal{Z}}|_z. \tag{32}$$

Note that $p'_z$ is constant over $\mathcal{Z}$ since $p_z$ is the uniform distribution over $\mathcal{Z}$. From Eq. (32), we learn that maximizing $\mathcal{H}_q$ is equivalent to maximizing $\int_{\mathcal{Z}} \log \sqrt{\det J_g^\top(z) J_g(z)} dV_{\mathcal{Z}}|_z$. On the other hand, given that $\mathcal{Z}$ is finite, we have that

$$\int_{\mathcal{Z}} dV_{\mathcal{Z}}|_z = m_{\mathcal{Z}}, \tag{33}$$

Therefore, according to Jensen inequality, we have that

$$\int_{\mathcal{Z}} \log \sqrt{\det J_g^\top(z) J_g(z)} dV_{\mathcal{Z}}|_z \leqslant m_{\mathcal{Z}} \log \left( \frac{1}{m_{\mathcal{Z}}} \int_{\mathcal{Z}} \sqrt{\det J_g^\top(z) J_g(z)} dV_{\mathcal{Z}}|_z \right) \tag{34}$$

$$= m_{\mathcal{Z}} \log \left( \frac{1}{m_{\mathcal{Z}}} \int_{\text{supp}(q')} dV_{\mathcal{M}}|_x \right) = m_{\mathcal{Z}} \log \frac{m_{\text{supp}(q')}}{m_{\mathcal{Z}}}, \tag{35}$$

where the equality holds if and only if $\det J_g^\top J_g$ is constant over $\mathcal{Z}$, which completes the proof. $\quad \square$

### 1.4 Proof of Proposition 1

We prove Proposition 1 in main text, which is restated as the following Proposition 1. Recall that in main text, we use an NF-based generator $g : \mathcal{Z} \subset \mathbb{R}^K \to \mathbb{R}^D$ for arbitrary $K < D$ with a hierarchical architecture as follows

$$g = f_L \circ e_L \circ f_{L-1} \circ e_{L-1} \circ \cdots \circ f_1 \circ e_1, \tag{36}$$

where $f_i : \mathbb{R}^{K_i} \to \mathbb{R}^{K_i}$ is a bijective and invertible NF module (*e.g.*, coupling layers [1, 2, 3]) with tractable determinant of Jacobian, $e_i : \mathbb{R}^{K_{i-1}} \to \mathbb{R}^{K_i}$ boosts the dimensionality of the input feature $x$ from $K_{i-1}$ to $K_i$ by simply padding $K_i - K_{i-1}$ zeros at the end of $x$ [4], and the dimensionalities of hierarchical features satisfy $K = K_0 < K_1 < \cdots < K_L = D$. We also mentioned that the pseudo inverse of $g$, $g^\dagger : \mathbb{R}^D \to \mathbb{R}^K$, is defined as

$$g^\dagger = e_1^\dagger \circ f_1^{-1} \circ \cdots \circ e_{L-1}^\dagger \circ f_{L-1}^{-1} \circ e_L^\dagger \circ f_L^{-1}, \tag{37}$$

where $f_i^{-1} : \mathbb{R}^{K_i} \to \mathbb{R}^{K_i}$ is the inverse of $f_i$, and $e_i^\dagger : \mathbb{R}^{K_i} \to \mathbb{R}^{K_{i-1}}$ is the pseudo inverse of $e_i$ that reduces the dimensionality of the input feature $x$ from $K_i$ to $K_{i-1}$ by simply dropping $K_i - K_{i-1}$ zeros at the end of $x$. We then prove the following Proposition 1:

**Proposition 1.** $\forall z \in \mathcal{Z}$, $J_{g^\dagger}(g(z)) J_g(z) = I \in \mathbb{R}^{K \times K}$ holds, where $I$ is the identity matrix.

*Proof.* Given $z_0 \in \mathcal{Z}$ and $x_L \in g(\mathcal{Z})$, we denote that

$$\widetilde{z}_l \triangleq e_l(z_{l-1}), \quad z_l \triangleq f_l(\widetilde{z}_l), \qquad\qquad 1 \leqslant l \leqslant L, \tag{38}$$

$$x_{l-1} \triangleq e_l^\dagger(\widetilde{x}_l), \quad \widetilde{x}_l = f_l^{-1}(x_l), \qquad 1 \leqslant l \leqslant L. \tag{39}$$

Therefore, we have that

$$J_g(z_0) = J_{f_L}(\widetilde{z}_L) J_{e_L}(z_{L-1}) J_{f_{L-1}}(\widetilde{z}_{L-1}) J_{e_{L-1}}(z_{L-2}) \cdots J_{f_1}(\widetilde{z}_1) J_{e_1}(z_0), \tag{40}$$

$$J_{g^\dagger}(x_L) = J_{e_1^\dagger}(\widetilde{x}_1) J_{f_1^{-1}}(x_1) \cdots J_{e_{L-1}^\dagger}(\widetilde{x}_{L-1}) J_{f_{L-1}^{-1}}(x_{L-1}) J_{e_L^\dagger}(\widetilde{x}_L) J_{f_L^{-1}}(x_L). \tag{41}$$

Since $e_i : \mathbb{R}^{K_{i-1}} \to \mathbb{R}^{K_i}$ and $e_i^\dagger : \mathbb{R}^{K_i} \to \mathbb{R}^{K_{i-1}}$ are linear transformations, we have that

$$J_{e_i^\dagger} J_{e_i} \equiv I_{K_{i-1} \times K_i} I_{K_i \times K_{i-1}} = I_{K_{i-1} \times K_{i-1}}, \tag{42}$$

where we denote $I_{m \times n}$ as an $m \times n$ identity matrix. On the other hand, for $x_L = g(z_0)$, we have

$$z_L = x_L \implies \widetilde{z}_L = \widetilde{x}_L \implies z_{L-1} = x_{L-1} \implies \cdots \implies z_0 = x_0. \tag{43}$$

Therefore, we have that $x_i = f_i \circ f_i^{-1}(x_i) = f_i(\widetilde{x}_i) = f(\widetilde{z}_i)$, which implies that

$$J_{f_i^{-1}}(x_i) J_{f_i}(\widetilde{z}_i) = I_{K_i \times K_i}. \tag{44}$$

Therefore, by plugging Eq. (42) and Eq. (44) into Eq. (40) and Eq. (41), we have that

$$J_{g^\dagger}(g(z)) J_g(z) \tag{45}$$

$$= J_{e_1^\dagger}(\widetilde{x}_1) J_{f_1^{-1}}(x_1) \cdots J_{e_{L-1}^\dagger}(\widetilde{x}_{L-1}) J_{f_{L-1}^{-1}}(x_{L-1}) J_{e_L^\dagger}(\widetilde{x}_L) J_{f_L^{-1}}(x_L) \tag{46}$$

$$J_{f_L}(\widetilde{z}_L) J_{e_L}(z_{L-1}) J_{f_{L-1}}(\widetilde{z}_{L-1}) J_{e_{L-1}}(z_{L-2}) \cdots J_{f_1}(\widetilde{z}_1) J_{e_1}(z_0) \tag{47}$$

$$= J_{e_1^\dagger}(\widetilde{x}_1) J_{f_1^{-1}}(x_1) \cdots J_{e_{L-1}^\dagger}(\widetilde{x}_{L-1}) J_{f_{L-1}^{-1}}(x_{L-1}) \tag{48}$$

$$J_{f_{L-1}}(\widetilde{z}_{L-1}) J_{e_{L-1}}(z_{L-2}) \cdots J_{f_1}(\widetilde{z}_1) J_{e_1}(z_0) \tag{49}$$

$$= J_{e_1^\dagger}(\widetilde{x}_1) J_{f_1^{-1}}(x_1) \cdots J_{f_1}(\widetilde{z}_1) J_{e_1}(z_0) \tag{50}$$

$$= J_{e_1^\dagger}(\widetilde{x}_1) J_{f_1^{-1}}(x_1) J_{f_1}(\widetilde{z}_1) J_{e_1}(z_0) \tag{51}$$

$$= I_{K \times K}, \tag{52}$$

which completes the proof. $\qquad\square$

For any matrix $J$, we adopt $\sigma^*(J)$ and $\sigma_*(J)$ to denote the maximum and the minimum singular value of $J$, respectively. Given $J_{g^\dagger} J_g = I$, we intuitively illustrate that $\sigma_*(J_g) = \frac{1}{\sigma^*(J_{g^\dagger})}$ in the case of $u^* \in T_{g(z)}\mathcal{M}$ by using Fig. 4 in main text, where $u^*$ gives the maximum singular value of $J_{g^\dagger}$ and $\mathcal{M}$ is the generated manifold of $g$. However, it is notable that $\sigma_*(J_g) = \frac{1}{\sigma^*(J_{g^\dagger})}$ does not universally hold in the case of $J_{g^\dagger} J_g = I$ as we mentioned in main text. For a toy example, consider $J_g \triangleq [a, b]^\top \in \mathbb{R}^{1 \times 2}$ and $J_{g^\dagger} \triangleq [\frac{1}{a}, 0] \in \mathbb{R}^{2 \times 1}$. We can then verify that $J_{g^\dagger} J_g = [1] \in \mathbb{R}^{1 \times 1} = I$. We know that $\mathcal{U}^* \triangleq \left\{ [\alpha, 0]^\top \,|\, \alpha \in \mathbb{R} \right\}$ are all the vectors that give the maximum singular value of $J_{g^\dagger}$ and $T_{g(z)}\mathcal{M} \triangleq \{(\alpha a, \alpha b) \,|\, \alpha \in \mathbb{R}\}$, hence in the case of $b \neq 0$, we know that $\mathcal{U}^* \cap T_{g(z)}\mathcal{M} = \emptyset$, *i.e.*, there does not exist a tangent vector that gives the maximum singular value of $J_{g^\dagger}$. On the other hand, we know that $\frac{1}{\sigma^*(J_{g^\dagger})} = |a| \neq \sqrt{a^2 + b^2} = \sigma_*(J_g)$ in the case of $b \neq 0$. Therefore, this toy example is an illustration that $\frac{1}{\sigma^*(J_{g^\dagger})} \neq \sigma_*(J_g)$ can hold for $J_{g^\dagger}$ such that the maximum singular value can not be given by a vector from $T_{g(z)}\mathcal{M}$.

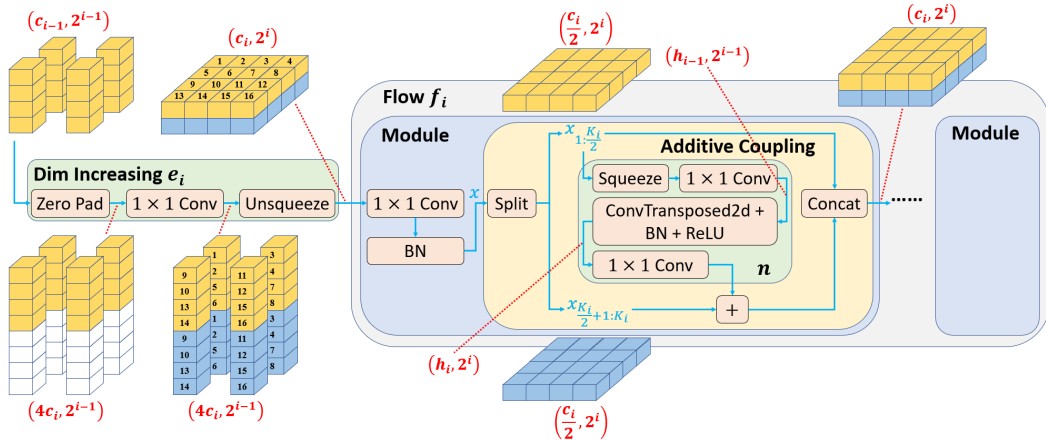

Figure 1: Illustration of the architecture of the $i$-th *block* $f_i \circ e_i : \mathbb{R}^{K_{i-1}} \to \mathbb{R}^{K_i}$, which consists of a *dim increasing* layer $e_i : \mathbb{R}^{K_{i-1}} \to \mathbb{R}^{K_i}$ and a *flow* layer $f_i : \mathbb{R}^{K_i} \to \mathbb{R}^{K_i}$. We use text $(c, s)$ in red to denote a feature of shape $c \times s \times s$, where $c$ and $s$ are the number of channels and spatial size, respectively. Therefore, $K_i = c_i \times 2^i \times 2^i$. To understand the operating mechanism of the *unsqueeze* layer, we mark pixels of a feature by using different numbers, and the correspondence between pixels of the input and output feature is indicated by numbers. The *squeeze* layer is actually an invert of the *unsqueeze* layer. The *split* and *concat* deal with input features along the channel axis.

## 2 Implementation Details

We provide implementation details of our NF-based generator, the generator uniformity regularization and the *udiv* diversity metric that are briefly introduced in main text.

### 2.1 The NF-based Generator

Given the architecture of our generator $g$ as shown in Eq. (36), we refer to $f_i \circ e_i : \mathbb{R}^{K_{i-1}} \to \mathbb{R}^{K_i}$ as the $i$-th *block* layer, which consists of a *dim increasing* layer $e_i$ and a *flow* layer $f_i$. We illustrate the architecture of a *block* layer as shown in Fig. 1. Given a feature of shape $K_{i-1} \triangleq c_{i-1} \times 2^{i-1} \times 2^{i-1}$, the *dim increasing* layer $e_i$ promotes the dimensionality of the input feature and outputs a feature of shape $K_i \triangleq c_i \times 2^i \times 2^i$ by using a *zero pad* layer, a $1 \times 1$ convolution layer, and a *unsqueeze* layer. The *zero pad* layer pads $(4c_i - c_{i-1}) \times 2^{i-1} \times 2^{i-1}$ zeros at the end of the input feature along the channel axis, obtaining an output feature of shape $4c_i \times 2^{i-1} \times 2^{i-1}$ whose subfeatures of different channels are further mixed by the following $1 \times 1$ convolution layer. The final *unsqueeze* layer further increases the spatial size of the feature while decreasing the number of channels without changing the dimensionality of the feature, obtaining the output feature of shape $c_i \times 2^i \times 2^i$. In terms of the *flow* layer $f_i : \mathbb{R}^{c_i \times 2^i \times 2^i} \to \mathbb{R}^{c_i \times 2^i \times 2^i}$, it consists of several *module* layers, where a *module* layer consists of $1 \times 1$ convolutional layer, a Batch Normalization [5] (BN) layer, and an *additive coupling* layer [1, 2, 3]. All these layers do not change the dimensionality of features. In terms of an *additive coupling* layer $r : \mathbb{R}^{K_i} \to \mathbb{R}^{K_i}$, for $y = r(x)$, the computation is given as

$$y_{1:\frac{K_i}{2}} = x_{1:\frac{K_i}{2}}, \tag{53}$$

$$y_{\frac{K_i}{2}+1:K_i} = x_{\frac{K_i}{2}+1:K_i} + n\left(x_{1:\frac{K_i}{2}}\right), \tag{54}$$

where we use $x_{a:b}$ to denote $[x_a, x_{a+1}, \cdots, x_b] \in \mathbb{R}^{b-a+1}$, and $n : \mathbb{R}^{\frac{K_i}{2}} \to \mathbb{R}^{\frac{K_i}{2}}$ (assume that $K_i$ is even) can be any flexible nonlinear neural network. However, a notable issue is that $y_{1:\frac{K_i}{2}}$ remains the same as $x_{1:\frac{K_i}{2}}$, which limits the flexibility of $r$. To overcome this problem, inspired by [3], the $1 \times 1$ convolution layer and the BN layer before $r$ are added. In terms of the nonlinear transformation $n$ inside $r$, given the input feature $x_{1:\frac{K_i}{2}}$ of shape $\frac{c_i}{2} \times 2^i \times 2^i$, we first use a *squeeze* layer to reduce its spatial size, obtaining a feature of shape $2c_i \times 2^{i-1} \times 2^{i-1}$, followed by using a $1 \times 1$ convolution

with a weight matrix of shape $h_{i-1} \times 2c_i$ to obtain a feature of shape $h_{i-1} \times 2^{i-1} \times 2^{i-1}$, where we refer to $h_i$ as the number of channels of hidden features of $i$-th block. We further perform upsampling on this feature by using a layer composed by a transposed convolution layer, a BN layer and a ReLU layer, obtaining a hidden feature of shape $h_i \times 2^i \times 2^i$. Finally, the hidden feature is transformed to the output feature of shape $c_i \times 2^i \times 2^i$ by using a $1 \times 1$ convolution layer with a weight matrix of shape $c_i \times h_i$. To stabilize training and reduce the amount of trainable parameters of our generator, for all $1 \times 1$ convolution layer with a weight matrix $W$ of shape $h \times w$, we first randomly initialize a rotation matrix $R$ of shape $s \times s$, where $s = \max\{h, w\}$, and then chose $W$ to be the first $h$ (*resp.*, $w$) rows (*resp.*, columns) of $R$ in the case of $h \leqslant w$ (*resp.*, $h > w$), and $W$ is fixed during training.

**Conditional generation**    The *manifold hypothesis* [6, 7, 8] assumes that high dimensional data is concentrated on a lower dimensional manifold. This assumption is natural for a set of images where the change between images is continuous, *e.g.*, the change between human face images is continuous. We refer to such a case as *continuous* data mode. However, there also exists sets of images where the change between images can be drastic, *e.g.*, for the MNIST [9] dataset, changes between images of different digits could be discontinuous and drastic. We refer to such a case as *discrete* data mode. Existing work [10, 11] summarize two different types of mode collapse: collapse to a subset of data modes, and collapse to a sub-manifold within the data distribution, which correspond to the discrete data mode and the continuous data mode, respectively. Given continuous data mode, it is reasonable that our proposed generator is applicable, since the generator is a continuous mapping. However, how to extend our generator to the setting of discrete data mode is a notable problem. In order to tackle this problem, we also extend our generator architecture to the conditional generation setting $\widetilde{g}(z; y)$, where $z \in \mathbb{R}^K$ is the latent code, and $y \in \mathbb{N}$ is the category label. Given that $\widetilde{g} : \mathbb{R}^K \times \mathbb{N} \to \mathbb{R}^D$ is the conditional generator, our idea is to make $g_y : \mathbb{R}^K \to \mathbb{R}^D$ a continuous generator that adopts the same architecture as used in the case of continuous data mode, where $g_y(z) \triangleq \widetilde{g}(z; y)$. Though this is similar to a multiple generator setting [12, 13, 14], we share model parameters between generators $g_y$ corresponding to different category $y$. Specifically, let $g : \mathbb{R}^K \to \mathbb{R}^D$ be a continuous generator that uses an architecture as shown in Eq. (36), in addition to the generator, we firstly set up trainable *category embeddings* $C \in \mathbb{R}^{Y \times K}$ where $Y$ is the number of categories, then modify $e_1$ from a layer that increases dimensionality from $K$ to $K_1$ to a layer that increases dimensionality from $2K$ to $K_1$. In such a case, the conditional generator $\widetilde{g} : \mathbb{R}^K \times \mathbb{N} \to \mathbb{R}^D$ is defined as

$$\widetilde{g}(z; y) = g\left(\text{concat}\left(z, C_y\right)\right), \quad z \in \mathcal{Z}, y \in \mathbb{N}, \tag{55}$$

where $C_y$ is the $y$-th row of matrix $C$, *i.e.*, the embedding of the $y$-th category, and $\text{concat}(z, C_y) \in \mathbb{R}^{2K}$ is a vector of length $2K$ obtained by concatenating the latent code $z \in \mathbb{R}^K$ and $C_y \in \mathbb{R}^K$. Such a modification does not increase the amount of model parameters. During our training in practice, for datasets with known number of categories, *e.g.*, MNIST that contains ten different categories of digits, we can set $Y$ to the ground-truth number of category. For datasets without category labels, we can use labels automatically derived from clustering in the discriminator's feature space as proposed in [15]. Moreover, to ensure that different generators capture modes of different category, we employ a MAD-GAN [12] discriminator distinguishes the real and fake samples along with identifying the generator that generates the given fake sample. See our code provided in supplementary material.

## 2.2   The Generator Uniformity Regularization

Recall that the generator uniformity regularization is given in main text as

$$\mathcal{L}_{\text{gunif}} \triangleq \mathbb{E}_{z \sim p_z}\left\{\left(\text{LT}_g(z) - a\right)^2 + \left(\frac{1}{\text{LT}_{g^\dagger}(g(z))} - a\right)^2\right\}, \tag{56}$$

where $\text{LT}_g(z)$ gives the estimated spectral norm of $J_g$ at $z$ by using the *Linearized Transpose* (LT) technique, and $a$ is the moving average of the minimum and the maximum singular values of $J_g$.

### 2.2.1   The *Linearized Transpose* Technique

We firstly present the detailed introduction to our proposed *Linearized Transpose* (LT) technique that estimates the spectral norm of the Jacobian $J$ of a given mapping $g$ in a sample efficient manner. We fundamentally use the fast approximation approach proposed in [16, 17, 18] in our LT process. Given

---

**Algorithm 1** *Linearized Transpose*

---

**Require:** A mapping $g : \mathbb{R}^K \to \mathbb{R}^D$, an input $z \in \mathbb{R}^K$, power of iteration $M$
1: **procedure** $\text{LT}_g(z; M)$
2:      Obtain $x = g(z)$, meanwhile construct $\widehat{g}_z$
3:      Randomly sample $\varepsilon \in \mathbb{R}^D$, then obtain $z' = \widehat{g}_z(\varepsilon) \in \mathbb{R}^K$
4:      Randomly sample $u, u' \in \mathbb{R}^D$ and $v \in \mathbb{R}^K$
5:      **for** $m = 1$ **to** $M$ **do**
6:          Update $v \leftarrow \mathcal{P}_{\text{grad}}\left(z, \frac{u^\top x}{\|u\|_2}\right)$
7:          Update $u' \leftarrow u, u \leftarrow \mathcal{P}_{\text{grad}}\left(\varepsilon, \frac{v^\top z'}{\|v\|_2}\right)$
8:      **end for**
9:      **return** estimated spectral norm $u^\top u'$
10: **end procedure**

---

a matrix $J \in \mathbb{R}^{D \times K}$, the fast approximation approach first randomly sample a vector $u \in \mathbb{R}^D$, then repeat the following update procedure several times

$$v \leftarrow \frac{J^\top u}{\|J^\top u\|_2}, \quad u \leftarrow \frac{Jv}{\|Jv\|_2}. \tag{57}$$

Finally, the spectral norm $\sigma^*$ of $J$ can be effectively approximated as $\sigma^* \approx u^\top Jv$. However, in our case the problem is that the matrix $J$ is implicit and is inefficient to estimate, since it is the Jacobian of a nonlinear mapping $g$. Fortunately, we observe that an explicit $J$ may be not necessary, since it is sufficient to perform fast approximation if we can estimate $J^\top u \in \mathbb{R}^K$ and $Jv \in \mathbb{R}^D$ directly. We are also aware that the *autograd* $\mathcal{P}_{\text{grad}}$ of deep learning libraries [19, 20] can efficiently perform

$$\mathcal{P}_{\text{grad}}\left(z, u^\top g(z)\right) = \sum_{i=1}^{D} \frac{\partial (u^\top g)_i}{\partial z} = J^\top u \in \mathbb{R}^K \tag{58}$$

and produces accurate estimation. Such a method was also adopted in [21]. Based on Eq. (58), $J^\top u$ is tractable. Similarly, to obtain $Jv$, we observe that a linear mapping $\widehat{g}$ should be constructed where $J_{\widehat{g}} = J^\top$, where $J_{\widehat{g}}$ is the Jacobian of $\widehat{g}$. To see this, by exploiting Eq. (58), we have

$$\mathcal{P}_{\text{grad}}\left(\varepsilon, v^\top \widehat{g}(\varepsilon)\right) = J_{\widehat{g}}^\top v = Jv, \tag{59}$$

where $\varepsilon \in \mathbb{R}^K$ can be an arbitrary vector, since $\widehat{g}$ is linear and hence $J_{\widehat{g}}$ is independent of $\varepsilon$. Hence we provide the formal definition of LT as

**Definition 1** (Linearized transpose). *Given a mapping $g : \mathbb{R}^K \to \mathbb{R}^D$, the linearized transpose of $g$ at $z$, is a linear mapping $\widehat{g}_z : \mathbb{R}^D \to \mathbb{R}^K$, such that*

$$J_{\widehat{g}_z} = J_g^\top(z). \tag{60}$$

Therefore, for generator $g$, as long as we are able to construct $\widehat{g}_z$, we can apply the aforementioned fast approximation approach to estimate the spectral norm of $J_g(z)$ by using Eq. 58 and Eq. 59, and the LT algorithm for obtaining $\text{LT}_g(z)$ is provided in Alg. 1. Similarly, $\text{LT}_{g^\dagger}(g(z))$ can be obtained as long as we are able to construct $\widehat{g^\dagger}_{g(z)}$.

### 2.2.2 Construction of Linearized Transpose

As we mentioned in Section 2.2.1, to obtain $\text{LT}_g(z)$ and $\text{LT}_{g^\dagger}(g(z))$, the main task is to construct $\widehat{g}_z$ and $\widehat{g^\dagger}_{g(z)}$. We firstly provide the following Proposition 2:

**Proposition 2.** *Given a mapping $f = f_L \circ f_{L-1} \circ \cdots \circ f_1$, we have*

$$\widehat{f}_{z_1} = \widehat{f_1}_{z_1} \circ \widehat{f_2}_{z_2} \circ \cdots \widehat{f_L}_{z_L}, \tag{61}$$

*where $z_{l+1} = f_l(z_l), 1 \leqslant l \leqslant L$.*

*Proof.* By checking

$$J_{\widehat{f}_{z_1}} = J_{\widehat{f}_{1_{z_1}}} J_{\widehat{f}_{2_{z_2}}} \cdots J_{\widehat{f}_{L_{z_L}}} = J_{f_1}^\top (z_1) J_{f_2}^\top (z_2) \cdots J_{f_L}^\top (z_L) \tag{62}$$

$$= (J_{f_L} (z_L) \cdots J_{f_2} (z_2) J_{f_1} (z_1))^\top = J_f^\top (z_1), \tag{63}$$

we learn that Eq. (61) gives the LT of $f$ at $z_1$ according to the Definition 1 of LT. $\qquad\square$

We then introduce how to construct $\widehat{g}_z$ and $\widehat{g^\dagger}_{g(z)}$. According to Eq. (36) and Eq. (37), by leveraging Proposition 2, we have

$$\widehat{g} = \widehat{e_1} \circ \widehat{f_1} \circ \cdots \circ \widehat{e_{L-1}} \circ \widehat{f_{L-1}} \circ \widehat{e_L} \circ \widehat{f_L}, \tag{64}$$

$$\widehat{g^\dagger} = \widehat{f_L^{-1}} \circ \widehat{e_L^\dagger} \circ \widehat{f_{L-1}^{-1}} \circ \widehat{e_{L-1}^\dagger} \circ \cdots \circ \widehat{f_1^{-1}} \circ \widehat{e_1^\dagger}, \tag{65}$$

where we omit the dependencies of $\widehat{g}_z$ (*resp.*, $\widehat{g^\dagger}_{g(z)}$) on $z$ (*resp.*, $g(z)$) to avoid redundancy. Hence we only need to construct $\widehat{e_i}$, $\widehat{e_i^\dagger}$, $\widehat{f_i}$ and $\widehat{f_i^{-1}}$.

**LT of $e_i$**   As we introduced in Section 2.1 and illustrated in Fig. 1, we know that the *dim increasing* layer $e_i : \mathbb{R}^{K_{i-1}} \to \mathbb{R}^{K_i}$ is composed as

$$e_i = u \circ w \circ p, \tag{66}$$

where $p : \mathbb{R}^{K_{i-1}} \to \mathbb{R}^{K_i}$, $w : \mathbb{R}^{K_i} \to \mathbb{R}^{K_i}$ and $u : \mathbb{R}^{K_i} \to \mathbb{R}^{K_i}$ are the *zero pad* layer, the $1 \times 1$ convolution layer, and the *unsqueeze* layer of $e_i$, respectively. From Proposition 2 we have

$$\widehat{e_i} = \widehat{p} \circ \widehat{w} \circ \widehat{u}. \tag{67}$$

Since $p$ pads $K_i - K_{i-1}$ zeros at the end of the input feature, we know that its Jacobian $J_p$ is the identity matrix $I_{K_i \times K_{i-1}}$. Hence we can verify that $\widehat{p} = p^\dagger$, where $p^\dagger : \mathbb{R}^{K_i} \to \mathbb{R}^{K_{i-1}}$ is the pseudo inverse of $p$ which drops the last $K_i - K_{i-1}$ elements of the input feature, because $J_{p^\dagger} = I_{K_{i-1} \times K_i}$ and hence $J_{p^\dagger} = J_p^\top$. Regarding $\widehat{w}$, we can easily verify that it is the ConvTransposed layer with the same weight as $w$. Since the *unsqueeze* operation $u$ only rearrange the elements of the input feature, we know that its Jacobian $J_u$ is a permutation matrix $P \in \mathbb{R}^{K_i \times K_i}$. Hence we can verify that $\widehat{u} = u^{-1} : \mathbb{R}^{K_i} \to \mathbb{R}^{K_i}$, where $u^{-1}$ is essentially the *squeeze* layer (*i.e.*, the inverted operation of the *unsqueeze* layer), because $J_{u^{-1}} = P^{-1} = P^\top = J_u^\top$. Summarizing the above gives us $\widehat{e_i}$.

**LT of $e_i^\dagger$**   Based on Eq. (66), we know that the pseudo inverse of $e_i$, $e_i^\dagger : \mathbb{R}^{K_i} \to \mathbb{R}^{K_{i-1}}$, is given as

$$e_i^\dagger = p^\dagger \circ w^{-1} \circ u^{-1}, \tag{68}$$

where $p^\dagger$ is the pseudo inverse of $p$ and $u^{-1}$ is the inverse of $u$ (*i.e.*, a *squeeze* layer) as mentioned above, and $w^{-1} : \mathbb{R}^{K_i} \to \mathbb{R}^{K_i}$ is the inverse of $w$ which can be easily verified as a $1 \times 1$ convolution layer with weight $W^\top$ where $W$ is the weight matrix of $w$ (note that $W$ is a rotation matrix). From Proposition 2 we have

$$\widehat{e_i^\dagger} = \widehat{u^{-1}} \circ \widehat{w^{-1}} \circ \widehat{p^\dagger}, \tag{69}$$

and we can know that $\widehat{u^{-1}} = u$ and $\widehat{p^\dagger} = p$ from what we discussed when we introduced the LT of $e_i$. In terms of $\widehat{w^{-1}}$, since $w^{-1}$ is essentially a $1 \times 1$ convolution layer with weight $W^\top$, we know that $\widehat{w^{-1}}$ is a ConvTranspose layer with weight $W^\top$. Summarizing the above gives us $\widehat{e_i^\dagger}$.

**LT of $f_i$**   As we introduced in Section 2.1 and illustrated in Fig. 1, we know that the *flow* layer $f_i$ consists of several *module* layers. Let $q$ be a *module* layer of $f_i$, then $q$ is composed as

$$q = w \circ b \circ r, \tag{70}$$

where $w : \mathbb{R}^{K_i} \to \mathbb{R}^{K_i}$, $b : \mathbb{R}^{K_i} \to \mathbb{R}^{K_i}$ and $r : \mathbb{R}^{K_i} \to \mathbb{R}^{K_i}$ are a $1 \times 1$ convolution layer, a BN layer, and an *additive coupling* layer, respectively. Therefore, in order to construct $\widehat{f_i}$, we only need to construct $\widehat{w}$, $\widehat{b}$ and $\widehat{r}$, respectively. As we mentioned above, $\widehat{w} : \mathbb{R}^{K_i} \to \mathbb{R}^{K_i}$ is a ConvTransposed

layer with the same weight as $w$. In terms of the BN layer $b$, since we set $b$ to the evaluation mode without tracking running statistics [5] when it comes to constructing LT, for $y = b(x)$, we have

$$y = \gamma \frac{x - \text{running\_mean}}{\sqrt{\text{running\_var}}} + \beta, \tag{71}$$

where $\gamma, \beta$ are the learnable parameters of the BN layer $b$, and $\text{running\_mean}, \text{running\_var}$ are the running statistics. Therefore, we know that the Jacobian $J_b \in \mathbb{R}^{K_i \times K_i}$ of $b$ is diagonal, and hence $\widehat{b}$ is $b$ itself because $J_{\widehat{b}} = J_b = J_b^{\top}$. In terms of $\widehat{r}$, we can verify that for $x = \widehat{r}(y)$, the computation

$$x_{1:\frac{K_i}{2}} = y_{1:\frac{K_i}{2}} + \widehat{n}\left(y_{\frac{K_i}{2}+1:K_i}\right), \tag{72}$$

$$x_{\frac{K_i}{2}+1:K_i} = y_{\frac{K_i}{2}+1:K_i}, \tag{73}$$

gives that $J_{\widehat{r}} = J_r^{\top}$, because

$$J_{\widehat{r}} = \begin{bmatrix} I_{\frac{K_i}{2} \times \frac{K_i}{2}} & J_{\widehat{n}} \\ 0 & I_{\frac{K_i}{2} \times \frac{K_i}{2}} \end{bmatrix} = \begin{bmatrix} I_{\frac{K_i}{2} \times \frac{K_i}{2}} & 0 \\ J_n & I_{\frac{K_i}{2} \times \frac{K_i}{2}} \end{bmatrix}^{\top} = J_r^{\top}. \tag{74}$$

Therefore, we need to construct $\widehat{n}$. As we introduced in Section 2.1 and illustrated in Fig. 1,

$$n = c_2 \circ \phi \circ b \circ v \circ c_1 \circ u^{-1}, \tag{75}$$

where $u^{-1} : \mathbb{R}^{\frac{K_i}{2}} \to \mathbb{R}^{\frac{K_i}{2}}$ is the *squeeze* layer of $n$, $c_1 : \mathbb{R}^{\frac{K_i}{2}} \to \mathbb{R}^{\widetilde{h}_{i-1}}$ and $c_2 : \mathbb{R}^{\widetilde{h}_i} \to \mathbb{R}^{\frac{K_i}{2}}$ are the first and second $1 \times 1$ convolution layer of $n$ respectively, $v : \mathbb{R}^{\widetilde{h}_{i-1}} \to \mathbb{R}^{\widetilde{h}_i}$ is the ConvTransposed layer of $n$, and $b, \phi : \mathbb{R}^{\widetilde{h}_i} \to \mathbb{R}^{\widetilde{h}_i}$ are the BN layer and the ReLU layer of $n$ respectively, where we use $\widetilde{h}_i \triangleq h_i \times 2^i \times 2^i$ (see Section 2.1 and Fig. 1). Therefore, we have

$$\widehat{n} = \widehat{u^{-1}} \circ \widehat{c_1} \circ \widehat{v} \circ \widehat{b} \circ \widehat{\phi} \circ \widehat{c_2}, \tag{76}$$

where we know that $\widehat{u^{-1}} = u$ (*i.e.*, a *unsqueeze* layer), $\widehat{c_1}$ (*resp.*, $\widehat{c_2}$) is the ConvTransposed layer with the same weight as $c_1$ (*resp.*, $c_2$), and $\widehat{b} = b$. In terms of the LT $\widehat{v}$ of the ConvTransposed layer $v$, we can verify that $\widehat{v}$ is the convolution layer weight the same weight as $v$. In terms of the LT $\widehat{\phi}$ of the ReLU layer, since it is a elementwise activation layer, for $y = \phi(x)$, we know that

$$J_{\phi}(x) = \begin{bmatrix} \phi'(x_1) & & & \\ & \phi'(x_2) & & \\ & & \ddots & \\ & & & \phi'\left(x_{\widetilde{h}_i}\right). \end{bmatrix} \tag{77}$$

Therefore, by denoting $\xi \triangleq \left[\phi'(x_1), \phi'(x_2), \cdots, \phi'\left(x_{\widetilde{h}_i}\right)\right] \in \mathbb{R}^{\widetilde{h}_i}$, we know that

$$\widehat{\phi}(y) = \xi \otimes y, \tag{78}$$

gives the LT of $\phi$, where $\otimes$ is the elementwise multiplication. Summarizing the above gives us $\widehat{f}_i$.

**LT of $f_i^{-1}$** Since $f_i$ consists of several *module* layers, $f_i^{-1}$ is given based on the invert of *module* layers. Let $q : \mathbb{R}^{K_i} \to \mathbb{R}^{K_i}$ be a *module* layer of $f_i$, based on Eq. (70), we know that the invert of $q$, $q^{-1} : \mathbb{R}^{K_i} \to \mathbb{R}^{K_i}$, is given as

$$q^{-1} = r^{-1} \circ b^{-1} \circ w^{-1}, \tag{79}$$

where $w^{-1} : \mathbb{R}^{K_i} \to \mathbb{R}^{K_i}$ is the invert of the $1 \times 1$ convolution layer $w$, $b^{-1} : \mathbb{R}^{K_i} \to \mathbb{R}^{K_i}$ is the invert of the BN layer $b$, and $r^{-1} : \mathbb{R}^{K_i} \to \mathbb{R}^{K_i}$ is the invert of the *additive coupling* layer $r$. As we mentioned above, $w^{-1}$ is a $1 \times 1$ convolution layer with weight $W^{\top}$ where $W$ is the weight matrix of $w$. In terms of $b^{-1}$, according to Eq. (71), for $x = b^{-1}(y)$, we know that

$$x = \frac{(y - \beta)\sqrt{\text{running\_var}}}{\gamma} + \text{running\_mean}. \tag{80}$$

**Algorithm 2** Train UniGAN.

---

**Require:** Training samples $\mathcal{D} = \left\{x^{(i)}\right\}_{i=1}^{N} \subset \mathbb{R}^D$, latent space $\mathcal{Z} \subset \mathbb{R}^K$, generator $g : \mathcal{Z} \to \mathbb{R}^D$, discriminator $d : \mathbb{R}^D \to (0,1)$, model parameters $\Theta_g, \Theta_d$, optimizers $\rho_g, \rho_d$, uniform prior distribution $p_z$ over $\mathcal{Z}$, batch size $B$, hyper-parameter $\lambda_{\text{gunif}}$, frequency of regularization $s$.

1: Initialize steps of training $i \leftarrow 1$
2: Randomly initialize the moving average $a$
3: **while** not converge **do**
    /* Train discriminator $d$. */
4:    Randomly sample $\left\{x^{(i)} \in \mathcal{D}\right\}_{i=1}^{B}$
5:    Randomly sample $\left\{z^{(i)} \sim p_z\right\}_{i=1}^{B}$, obtaining $\left\{\widetilde{x}^{(i)} = g\left(z^{(i)}\right)\right\}_{i=1}^{B}$
6:    Compute $\mathcal{L}_{\text{disc}} = \frac{1}{B}\sum_{i=1}^{B}\log d\left(x^{(i)}\right) + \frac{1}{B}\sum_{i=1}^{B}\log\left(1 - d\left(\widetilde{x}^{(i)}\right)\right)$
7:    Update $\Theta_d \leftarrow \rho_d \nabla_d \mathcal{L}_{\text{disc}}$
    /* Train generator $g$. */
8:    Randomly sample $\left\{z^{(i)} \sim p_z\right\}_{i=1}^{B}$, obtaining $\left\{\widetilde{x}^{(i)} = g\left(z^{(i)}\right)\right\}_{i=1}^{B}$
9:    Compute $\mathcal{L}_{\text{gen}} = \frac{1}{B}\sum_{i=1}^{B}\log d\left(\widetilde{x}^{(i)}\right)$
    /* Apply the generator uniformity regularization. */
10:   **if** $i \bmod s = 0$ **then**
11:     **if** $i/s \bmod 2 = 0$ **then**
12:       Compute the maximum singular values $\left\{\sigma^{(i)} = \text{LT}_g\left(z^{(i)}\right)\right\}_{i=1}^{B}$
13:     **else**
14:       Compute the minimum singular values $\left\{\sigma^{(i)} = \frac{1}{\text{LT}_{g^\dagger}\left(\widetilde{x}^{(i)}\right)}\right\}_{i=1}^{B}$
15:     **end if**
16:     Compute $\mathcal{L}_{\text{gunif}} = \frac{1}{B}\sum_{i=1}^{B}\left(\sigma^{(i)} - a\right)^2$
17:     Update $\mathcal{L}_{\text{gen}} \leftarrow \mathcal{L}_{\text{gen}} + \lambda_{\text{gunif}}\mathcal{L}_{\text{gunif}}$
18:     Update $a \leftarrow 0.99a + 0.01\left(\frac{1}{B}\sum_{i=1}^{B}\sigma^{(i)}\right)$
19:   **end if**
20:   Update $\Theta_g \leftarrow \rho_g \nabla_g \mathcal{L}_{\text{gen}}$
21:   Update $i \leftarrow i + 1$
22: **end while**

---

In terms of $r^{-1}$, for $x = r^{-1}(y)$ we can verify that the computation gives by $r^{-1}$ is as follows,

$$x_{1:\frac{K_i}{2}} = y_{1:\frac{K_i}{2}}, \tag{81}$$

$$x_{\frac{K_i}{2}+1:K_i} = y_{\frac{K_i}{2}+1:K_i} - n\left(y_{1:\frac{K_i}{2}}\right). \tag{82}$$

To construction $\widehat{q^{-1}} = \widehat{w^{-1}} \circ \widehat{b^{-1}} \circ \widehat{r^{-1}}$, we need to construct $\widehat{w^{-1}}$, $\widehat{b^{-1}}$ and $\widehat{r^{-1}}$. In terms of $\widehat{w^{-1}}$, as mentioned above, we know that it is a ConvTranspose layer with weight $W^\top$. In terms of $\widehat{b^{-1}}$, similar to obtaining $\widehat{b}$ from $b$, we know that $\widehat{b^{-1}}$ is $b^{-1}$ itself, since $J_{b^{-1}}$ is diagonal. In terms of $\widehat{r^{-1}}$, we can verify that for $y = \widehat{r^{-1}}(x)$, the computation

$$y_{1:\frac{K_i}{2}} = x_{1:\frac{K_i}{2}} - \widehat{n}\left(x_{\frac{K_i}{2}+1:K_i}\right), \tag{83}$$

$$y_{\frac{K_i}{2}+1:K_i} = x_{\frac{K_i}{2}+1:K_i}, \tag{84}$$

gives that $J_{\widehat{r^{-1}}} = J_{r^{-1}}^\top$, because

$$J_{\widehat{r^{-1}}} = \begin{bmatrix} I_{\frac{K_i}{2}\times\frac{K_i}{2}} & -J_{\widehat{n}} \\ 0 & I_{\frac{K_i}{2}\times\frac{K_i}{2}} \end{bmatrix} = \begin{bmatrix} I_{\frac{K_i}{2}\times\frac{K_i}{2}} & 0 \\ -J_n & I_{\frac{K_i}{2}\times\frac{K_i}{2}} \end{bmatrix}^\top = J_{r^{-1}}^\top. \tag{85}$$

The construction of $\widehat{n}$ is already given as mentioned above. Summarizing the above gives us $\widehat{f_i^{-1}}$.

### 2.2.3 Training Algorithm

We then introduce the whole training Algorithm 2 of our UniGAN framework augmented with the aforementioned generator uniformity regularization $\mathcal{L}_{\mathrm{gunif}}$ as in Eq. (56). The Algorithm 2 shows the training procedure of our UniGAN adapted for a vanilla GAN [22], and the algorithm can be adapted to other variants of GANs by replacing the method for training the generator and the discriminator.

**Sample efficiency**   We show that our proposed $\mathcal{L}_{\mathrm{gunif}}$ is superior in sample efficiency compared to other isometry learning methods. Given a generator $g : \mathcal{Z} \to \mathbb{R}^D$, the LGAN [8] proposes the following regularization

$$\mathcal{L}_{\mathrm{lgan}} = \mathbb{E}_{z \sim p_z} \mathbb{E}_{i \in [K]} \mathbb{E}_{j \in [K]} \left( \left( \frac{g\left(z + \Delta_i\right)}{\varepsilon} \right)^\top \left( \frac{g\left(z + \Delta_j\right)}{\varepsilon} \right) - \delta_{ij} \right)^2 \qquad (86)$$

to encourage an isometric generator, *i.e.*, $J_g^\top J_g = I$, where we use $[K] \triangleq \{1, 2, \cdots, K\}$, $\varepsilon$ is a small positive real number, and $\Delta_i$ is a $K$-dimensional vector whose $i$-th element is $\varepsilon$ and the other elements are zero. Therefore, $\frac{g(z+\Delta_i)}{\varepsilon}$ is the approximated $i$-th column vector of $J_g(z)$, and $\delta_{ij}$ is the Kronecker delta. By utilizing the automatic differentiation function of existing deep learning libraries like PyTorch [23], the StyleGAN2 [21] proposes the following path length regularization

$$\mathcal{L}_{\mathrm{stylegan2}} = \mathbb{E}_{z \sim p_z} \mathbb{E}_{y \sim \mathcal{N}(0,1)} \left( \left\| J_g^\top\left(z\right) y \right\|_2 - a \right)^2, \qquad (87)$$

where $a$ is the moving average of $\left\| J_g^\top\left(z\right) y \right\|_2$. For $\mathcal{L}_{\mathrm{lgan}}$ and $\mathcal{L}_{\mathrm{stylegan2}}$, we can observe that given a $z \sim p_z$, both regularizations lead to further sampling complexity, since $\mathcal{L}_{\mathrm{lgan}}$ requires to sample $i, j$ from $[k]$ while $\mathcal{L}_{\mathrm{stylegan2}}$ requires to sample $y$ from $\mathcal{N}(0, 1)$ over a high dimensional space $\mathbb{R}^D$. So to effectively encourage the generator $g$ to be an isometry, the further sampling procedure should be sufficient for both regularizations. However, as shown in Eq. (56), our regularization $\mathcal{L}_{\mathrm{gunif}}$ does not require any further sampling procedure given a $z \sim p_z$, since the LT technique does not require any sampling procedure given a $z \sim p_z$. Therefore, our $\mathcal{L}_{\mathrm{gunif}}$ is superior in sample efficiency compared to $\mathcal{L}_{\mathrm{lgan}}$ and $\mathcal{L}_{\mathrm{stylegan2}}$. We provide ablation study on sample efficiency in Section 3.2.

**Computational cost and memory usage**   Though our proposed LT is superior in sampling complexity as we mentioned above, a major concern is that how our LT performs in terms of computational cost and the memory usage. From Algorithm 1, we learn that to obtain $\mathrm{LT}_g(z)$, we need to firstly compute $x = g(z)$ to obtain $x$. However, this does not increase the computational burden since we can incorporate this forward computation into the computation that generates fake samples when training the generator in Algorithm 2. In terms of constructing LT, according to what we mentioned in Section 2.2.2, we learn that except for ReLU which requires recording the derivatives with respect to the input (*i.e.*, $\xi$ in Eq. (78)) to construct LT, other LT constructions can be obtained directly without involving any additional process. However, due to the simplicity of ReLU, the computational cost of the ReLU derivatives is negligible, and the derivatives can be obtained while generating fake samples without additional computation. During training, we only record the ReLU derivatives of the current batch training samples, which leads to negligible memory usage. To obtain $\mathrm{LT}_{g^\dagger}(g(z))$, we only require a computation $z = g^\dagger(g(z))$ to map generated samples back to latent codes, no more extra computation is needed. We notice that our LT algorithm involves a hyper-parameter, *i.e.*, *power of iterations* $M$, which can lead to different computational cost with different $M$. We provide ablation study on $M$ in Section 3.2.

**Lazy regularization**   Similar to [21], we also adopt a lazy regularization strategy when we apply the generator uniformity regularization. Specifically, by changing the hyper-parameter *frequency of regularization* $s$, the computational cost of the $\mathcal{L}_{\mathrm{gunif}}$ can be reduced since it can be computed less frequently than the main loss function [21]. We provide ablation study on $s$ in Section 3.2.

### 2.3 The *udiv* Diversity Metric

To realize the computation of uniform diversity in practice, we propose a new diversity metric named *udiv* estimating the uniform diversity of a given set of generative samples $\mathcal{X} = \left\{x^{(i)}\right\}_{i=1}^N$, where $\mathcal{X}$ contains $N$ samples coming from a generative distribution.

**The *sampling bias* problem**   It is known that directly computing the entropy based on $\mathcal{X}$ leads to biased results compared to the ground-truth entropy of $q$ [24]. We firstly use a toy example to show this problem intuitively and introduce our idea to tackle this problem. Let $q \in \mathcal{P}_{[0,1]}$ be a distribution over the interval $[0, 1]$ whose PDF $q'$ is defined as

$$q'(x) = \begin{cases} 0.5, & 0 \leqslant x \leqslant 0.8, \\ 3.0, & 0.8 < x \leqslant 1. \end{cases} \tag{88}$$

since $q([0, 0.8]) = 0.4$ and $q((0.8, 1]) = 0.6$, given $\mathcal{X}$, it is reasonable to assume that $\frac{2N}{5}$ samples come from $[0, 0.8]$ and $\frac{3N}{5}$ samples come from $(0.8, 1]$. Therefore, according to

$$\mathcal{H}_q = -\int_0^1 q'(x) \log q'(x) \, \mathrm{d}x = \lim_{N \to \infty} \sum_{i=1}^N \frac{1}{N} \left( -q'\left(\frac{i}{N}\right) \log q'\left(\frac{i}{N}\right) \right), \tag{89}$$

the approximated entropy based on $\mathcal{X}$ is given as

$$\mathcal{H}_\mathcal{X} = \frac{1}{N} \sum_{i=1}^N \left( -q'\left(x^{(i)}\right) \log q'\left(x^{(i)}\right) \right) \tag{90}$$

$$= \frac{1}{N} \left( \frac{2N}{5} \left(-0.5 \log 0.5\right) + \frac{3N}{5} \left(-3.0 \log 3.0\right) \right) \tag{91}$$

$$= \frac{2}{5} \left(-0.5 \log 0.5\right) + \frac{3}{5} \left(-3.0 \log 3.0\right) \tag{92}$$

$$\approx -1.83 \tag{93}$$

However, the ground-truth entropy $\mathcal{H}_q$ is given as

$$\mathcal{H}_q = \int_0^1 -q'(x) \log q'(x) \, \mathrm{d}x = 0.8 \left(-0.5 \log 0.5\right) + 0.2 \left(-3.0 \log 3.0\right) \approx -0.38. \tag{94}$$

Comparing Eq. (93) to Eq. (94), we learn that the estimated entropy based on $\mathcal{X}$ is seriously biased compared to the ground-truth result. From Eq. (92) and Eq. (94), we observe that the bias is caused by the inaccuracy of the factors of $(-0.5 \log 0.5)$ and $(-3.0 \log 3.0)$, *i.e.*, $\frac{2}{5}$ and $\frac{3}{5}$ respectively, in Eq. (92), because the ground-truth factors in Eq. (94) is 0.8 and 0.2, respectively. We observe that

$$\frac{2}{5}/0.5 : \frac{3}{5}/3.0 = 0.8 : 0.2, \tag{95}$$

therefore, if we compute the following *weighted* entropy based on $\mathcal{X}$, we have

$$\widetilde{\mathcal{H}}_\mathcal{X} = \frac{1}{N} \sum_{i=1}^N \left( -w^{(i)} q'\left(x^{(i)}\right) \log q'\left(x^{(i)}\right) \right) \tag{96}$$

$$= \frac{1}{N} \left( \frac{2N}{5} \times \frac{1}{0.5} \left(-0.5 \log 0.5\right) + \frac{3N}{5} \times \frac{1}{3.0} \left(-3.0 \log 3.0\right) \right) \tag{97}$$

$$= 0.8 \left(-0.5 \log 0.5\right) + 0.2 \left(-3.0 \log 3.0\right), \tag{98}$$

which gives the exact estimation of $\mathcal{H}_q$, where the weight $w^{(i)} = \frac{1}{q'\left(x^{(i)}\right)}$.

**The *weighted estimation* method**   Based on aforementioned analysis, we formalized our *weighted estimation* method for accurately estimating $\mathcal{H}_q$ based on a set of generative samples $\mathcal{X} = \left\{ x^{(i)} \right\}_{i=1}^N$. Let $p_i$ be the estimated probability density of $x^{(i)}$, and $w_i$ be the weight of $x^{(i)}$ for computing entropy, then the weighted estimation of entropy based on $\mathcal{X}$, $\widetilde{\mathcal{H}}_\mathcal{X}$, is computed as

$$\widetilde{\mathcal{H}}_\mathcal{X} = -\sum_{i=1}^N w_i p_i \log p_i, \quad \text{s.t.} \sum_{i=1}^N p_i w_i = 1, \sum_{i=1}^N w_i = N, \tag{99}$$

where the constrain is added to satisfy the property of a PDF, namely by regarding $p_i$ and $w_i$ as the probability density and the volume of the $i$-th region, the integral of probability densities $\sum_{i=1}^N p_i w_i$

---

**Algorithm 3** The *udiv* metric.

---

**Require:** Generative samples $\mathcal{X} \triangleq \left\{x^{(i)}\right\}_{i=1}^{N}$, KDE bandwidth $0 \leqslant h_1 < h_2 < \cdots < h_T$, uniform
   kernel $\psi_h$ where $\psi_h\left(x; x^*\right) = \begin{cases} 1, & \left\|x - x^*\right\| \leqslant h \\ 0, & \text{otherwise.} \end{cases}$

1: **procedure** $\mathrm{ent}(\mathcal{X}, h)$
2:      Initialize probability densities $p_i \leftarrow 0, i = 1, 2, \cdots, N$
3:      **for** $i = 1, 2, \cdots, N$ **do**
4:          Update $p_i \leftarrow \sum_{j=1}^{N} \psi_h\left(x^{(i)}; x^{(j)}\right)$
5:      **end for**
6:      Normalize $\log p_i \leftarrow \log p_i - 2\log N + \mathrm{logsumexp}\left(-\log p_i\right)$ to satisfy Eq. (100)
7:      Compute entropy $e \leftarrow -\frac{1}{N}\sum_{i=1}^{N} \log p_i$ according to Eq. (101)
8:      Normalize entropy $e \leftarrow \frac{e}{\log N}$
9:      **return** $e$
10: **end procedure**
11: Compute normalized entropies $\left\{e_t \leftarrow \mathrm{ent}\left(\mathcal{X}, h_t\right)\right\}_{t=1}^{T}$
12: Obtain the *udiv* score $u = \min\left\{e_t\right\}_{t=1}^{T}$

---

should be equal to 1, and the total volume $\sum_{i=1}^{N} w_i$ is $N$. As we mentioned above, $w_i : w_j = \frac{1}{p_i} : \frac{1}{p_j}$, hence from $\sum_{i=1}^{N} p_i w_i = 1$ we have $w_i = \frac{1}{Np_i}$, and from $\sum_{i=1}^{N} w_i = N$ we have

$$\sum_{i=1}^{N} \frac{1}{p_i} = N^2. \tag{100}$$

Therefore, from Jensen inequality we have

$$\widetilde{\mathcal{H}}_{\mathcal{X}} = -\sum_{i=1}^{N} w_i p_i \log p_i = -\sum_{i=1}^{N} \frac{1}{N} \log p_i = \sum_{i=1}^{N} \frac{1}{N} \log \frac{1}{p_i} \leqslant \log\left(\sum_{i=1}^{N} \frac{1}{N}\frac{1}{p_i}\right) = \log N, \tag{101}$$

where the equality holds if and only if $p_1 = p_2 = \cdots = p_N$, namely the probability distribution is a uniform distribution. Therefore, the entropy normalized to $[0, 1]$ is given as $\mathcal{E}_{\mathcal{X}} \triangleq \frac{\widetilde{\mathcal{H}}_{\mathcal{X}}}{\log N}$.

**The *udiv* metric** Given $\mathcal{X} = \left\{x^{(i)}\right\}_{i=1}^{N}$, we adopt the normalized weighted entropy $\mathcal{E}_{\mathcal{X}}$ as the *udiv* metric score that indicates the uniform diversity of $\mathcal{X}$. To estimate the probability density $p_i$ of $x^{(i)}$ for computing $\mathcal{E}_{\mathcal{X}}$, we propose to use the *Kernel Density Estimation* [25] (KDE) technique, and the procedure for estimating the *udiv* metric score based on KDE is provided in Algorithm 3. Since KDE involves selecting the kernel bandwidth [26], we know that $\mathcal{E}_{\mathcal{X}}$ is dependent on kernel bandwidth $h$. Specifically, for the uniform kernel $\psi_h$ we adopt in Algorithm 3, we can verify that $h = 0$ gives that $\psi_h\left(x; x^*\right)$ is equal to 1 if and only if $x = x^*$, therefore, $p_i = 1$ for all $i$ and hence $\mathcal{E}_{\mathcal{X}}$ is maximized because all probability densities $p_i$ are equal. On the other hand, for sufficiently large $h$, $\psi_h\left(x; x^*\right)$ is always equal to 1 because $\left\|x^{(i)} - x^{(j)}\right\| \leqslant h$ always holds, therefore, $p_i = N$ for all $i$ and hence $\mathcal{E}_{\mathcal{X}}$ is also maximized because all probability densities $p_i$ are also equal. Hence we know that there exists $h$ such that $h$ gives the minimum of $\mathcal{E}_{\mathcal{X}}$, which we adopt as the *udiv* metric score.

## 3 Experimental Results

We provide addition results on the synthetic data manifold provided in main text, and provide results on simple datasets including MNIST [9], FashionMNIST [27] and their colored version [10], and CIFAR10 [28]. We also provide results on natural image datasets including CelebA [29], FFHQ [30], AFHQ [31] and LSUN [32]. All datasets are publicly available. We adapt our UniGAN for a variety of existing models that reduce mode collapse, including GAN [22], MSGAN [33, 34], PacGAN [11], BiGAN [35]/ALI [36], VEEGAN [37], MDGAN [38] and RegGAN [38]. Our experiments show that the generator uniformity is closely related with the uniform diversity, and also show the effectiveness of our proposed UniGAN framework in improving generator uniformity and uniform diversity.

Table 1: Architectural details for the generator and discriminator for the 2D synthetic data manifold *sector*. For the generator, since all features are vectors, the *dim increasing* layer $e_i : \mathbb{R}^{K_{i-1}} \to \mathbb{R}^{K_i}$ does not involve an *unsqueeze* layer but only consists of a *zero pad* layer and a $1 \times 1$ convolution layer, and the neural network $n : \mathbb{R}^{\frac{K_i}{2}} \to \mathbb{R}^{\frac{K_i}{2}}$ inside the *additive coupling* of a *flow* layer $f_i : \mathbb{R}^{K_i} \to \mathbb{R}^{K_i}$ is composed as "FC@$h_i$, ReLU, FC@$\frac{K_i}{2}$", where we use FC@$c$ to denote a fully connected layer with output shape $c$, and $h_i$ is the shape of hidden features of the $i$-th *block* layer $f_i \circ e_i$, see Section 2.1 and Fig. 1. We denote the block layer with the shape of the output feature as $c$, the shape of hidden features as $h$, and the number of *module* layers as $m$ as Block@$c\#h\#m$.

| Generator | | Discriminator | |
|---|---|---|---|
| layers | output shape | layers | output shape |
| Input noise $z \in \mathbb{R}^2$ | | Input sample $x \in \mathbb{R}^2$ | |
| Block@1024#1024#3 | 1024 | FC@1024, ReLU | 1024 |
| | | FC@1024, ReLU | 1024 |
| | | FC@1024, ReLU | 1024 |
| | | FC@1 | 1 |
| total number of parameters: 3.504M | | total number of parameters: 1.054M | |

**Experimental settings**  We introduce shared setting across all datasets for conducting our experiments, while the settings specific to different datasets are introduced in the corresponding sections below. We use Adam [39] optimizer with fixed learning rate $10^{-4}$, and all models are end-to-end trained with PyTorch [19] from scratch. We train $40,000$ steps with batch size of each step being 96 for all models. In terms of $\mathcal{L}_{\mathrm{gunif}}$, we set $\lambda_{\mathrm{gunif}}$ to $1.0$ and *frequency of regularization s* to 4, and the *power of iteration M* involved in LT is set to 2. In terms of evaluating *udiv*, we randomly generate $1,000$ samples, and 100 kernel bandwidths are selected from range $[0, 1]$ with step $0.01$. In terms of evaluating the *gunif* metric indicating the generator uniformity, we first randomly sampling $1,000$ latent codes $\left\{z^{(i)}\right\}_{i=1}^{1000}$, then compute the standard deviation of estimated $\left\{\log \det J^\top J \left(z^{(i)}\right)\right\}_{i=1}^{1000}$, where for each $z^{(i)}$, $\log \det J^\top J \left(z^{(i)}\right)$ is accurately computed by using the automatic differential function of PyTorch. We also show the correlation between the probability densities $\left\{p^{(i)}\right\}_{i=1}^{1000}$ and $\left\{\log \det J^\top J \left(z^{(i)}\right)\right\}_{i=1}^{1000}$, where $p^{(i)}$ is the probability density of $g\left(z^{(i)}\right)$ estimated by the method introduced in Algorithm 3. In order to avoid randomness, experiments are run 10 times with random initialization, and for each experiment, we provide the $p$-value of the significant test or the confidence interval over multiple runs. Note that a $p$-value below $0.05$ indicates significant differences between the baselines results and ours. In terms of the type of computing resources used, all experiments are done using a single NVIDIA RTX-2080Ti GPU.

## 3.1 Synthetic Data Manifold

**Experimental settings**  We provide architectural details of the generator and discriminator used for the 2D synthetic data manifold *sector* as in Tbl. 1. The generator $g$ is composed of a *block* layer $f \circ e$, namely $g = f \circ e : \mathbb{R}^2 \to \mathbb{R}^{1024}$, where $e : \mathbb{R}^2 \to \mathbb{R}^{1024}$ is a *dim increasing* layer that boosts the dimensionality of the input feature from 2 to 1024 to permit a *flow* layer $f : \mathbb{R}^{1024} \to \mathbb{R}^{1024}$ that is a high dimensional nonlinear mapping with more expressive power. Given an output feature $x \in \mathbb{R}^{1024}$ of the generator, we directly drop the final 1022 elements of $x$, leaving $\widetilde{x} = [x_1, x_2] \in \mathbb{R}^2$ as the final 2-dimensional generated sample, where $x_i$ is the $i$-th element of $x$.

## 3.2 Image Datasets

**Experimental settings**  We provide more experimental results on image datasets, including simple datasets such as MNIST [9], FashionMNIST [27] and their colored version [10], and CIFAR10 [28], and natural image datasets such as CelebA [29], FFHQ [30], AFHQ [31] and LSUN [32]. For simple datasets, architectural details for the generator and discriminator are shown in Tbl. 2, and the shape of image is img_nc $\times 32 \times 32$, where img_nc is 1 for grayscale image datasets including MNIST and FashionMNIST, and img_nc is 3 for RGB image datasets including colored MNIST, colored FashionMNIST, and CIFAR10. For natural image datasets, architectural details for the generator and

Table 2: Architectural details for the generator and discriminator for simple datasets. In terms of the generator, we denote the *block* layer with the number of output feature channels as $c$, the number of hidden feature channels as $h$, and the number of *module* layers as $m$ as Block@$c$#$h$#$m$. Note that for the first layer Block@128#128#1, in order to reduce the amount of parameters, the upsampling scale of the *unsqueeze* layer inside the *dim increasing* layer is 4 instead of 2, namely the *unsqueeze* layer reshapes the input feature of shape $2048 \times 1 \times 1$ into the output feature of shape $128 \times 4 \times 4$. In terms of the discriminator, Conv@$c$#$k$#$s$#$p$ denotes a convolution layer with output channels $c$, kernel size $k$, stride $s$, and padding $p$. We use LReLU to denote a leaky ReLU layer with negative slope 0.2. The channels img_nc of the input image is 1 for grayscale images and 3 for RGB images.

| Generator | | Discriminator | |
|---|---|---|---|
| layers | output shape | layers | output shape |
| Input noise $z \in \mathbb{R}^6$ | | Input image $x \in \mathbb{R}^{\text{img\_nc} \times 32 \times 32}$ | |
| Block@128#128#1 | $128 \times 4 \times 4$ | Conv@32#4#2#1, LReLU | $32 \times 16 \times 16$ |
| Block@64#64#1 | $64 \times 8 \times 8$ | Conv@64#4#2#1, BN, LReLU | $64 \times 8 \times 8$ |
| Block@32#32#1 | $32 \times 16 \times 16$ | Conv@128#4#2#1, BN, LReLU | $128 \times 4 \times 4$ |
| Block@8#16#1 | $8 \times 32 \times 32$ | Conv@1#4#1#0 | 1 |
| total number of parameters: 0.280M | | total number of parameters: 0.188M | |

Table 3: Architectural details for the generator and discriminator for natural image datasets. We use a similar generator architecture as in Tbl. 2, and use a StyleGAN2 [21] discriminator.

| Generator | | Discriminator | |
|---|---|---|---|
| layers | output shape | layers | output shape |
| Input noise $z \in \mathbb{R}^{64}$ | | Input image $x \in \mathbb{R}^{3 \times 64 \times 64}$ | |
| Block@512#512#1 | $512 \times 4 \times 4$ | The StyleGAN2 [21] discriminator | 1 |
| Block@256#256#3 | $256 \times 8 \times 8$ | | |
| Block@128#128#3 | $128 \times 16 \times 16$ | | |
| Block@64#64#3 | $64 \times 32 \times 32$ | | |
| Block@16#32#3 | $16 \times 64 \times 64$ | | |
| total number of parameters: 8.889M | | total number of parameters: 22.679M | |

discriminator are shown in Tbl. 3, and the shape of image is $3 \times 64 \times 64$. Similar to the settings for the 2D synthetic data manifold *sector*, we increase the dimensionality of the intermediate features to be higher than the dimensionality of the image to improve the expressive power of the generator, and obtain the generated sample by dropping redundant channels of the output feature of the generator. In terms of the setting of generation, for simple datasets, we extend our UniGAN framework to a setting of categorical generation as we mentioned in Section 2.1, because all these datasets contain images of different classes with known category labels. In terms of natural image datasets, we use a normal generation setting. For datasets including only a single category, *i.e.*, CelebA and FFHQ, we directly use the whole dataset to train the model, and for datasets including multiple categories, *i.e.*, AFHQ and LSUN, we train different models for each category respectively.

### 3.2.1 Quantitative Results

For explicit comparison, we first provide quantitative results on a variety of metrics as in Tbl. 4-16. For simple datasets involving discrete modes, we adopt metrics that measure the model performance regarding covering different modes, including *Inception Score* [38, 40], *Mode Score* [38], *Number of Modes* [38, 10] and *KL divergence* between generated and ground-truth category distribution [10]. We also use metrics that measure the quality and diversity of generated samples, including *IvOM* [10], *Pairwise Distance* [10], *NDB* [41] and *JSD* [41]. For natural image datasets, we measure the quality and diversity of generated samples by using the *FID* [42] and *LPIPS* [43] metrics. For both types of datasets, we evaluate the uniform diversity and generator uniformity by using the proposed *udiv* and *gunif* metric, respectively. In Tbl. 4-16, the comparisons are made between baseline models and our UniGANs, where ↑ (*resp.*, ↓) indicates that larger (*resp.*, lower) values are better. For each metric, the first and the second rows give the results of baselines and the corresponding UniGANs, respectively,

we also provide the $p$-value of significant tests in the third row. For each baseline model, the better value among the baseline results and the corresponding UniGAN result is shown in bold.

From Tbl. 4-16, we learn that our proposed UniGANs achieve similar performances to baselines in terms of existing metrics that measure quality and diversity of generated samples. However, the *udiv* and *gunif* scores indicate that our UniGANs significantly outperform baselines in terms of uniform diversity and generator uniformity, which also implies that the uniform diversity indeed corresponds to a new type of mode collapse that cannot be captured by existing metrics.

In Tbl. 17, we provide quantitative results on CelebA dataset with $128 \times 128$ resolution to verify the effectiveness of our method on high-resolution datasets. From Tbl. 17, we see that the performance of our method is similar to that on the $64 \times 64$ resolution datasets, namely by imposing the proposed generator uniformity regularization, our UniGANs outperform baselines in terms of the udiv and the gunif metrics, which shows the effectiveness of our method on the $128 \times 128$ CelebA dataset.

### 3.2.2 Qualitative Results

We provide qualitative results on each image dataset in Fig. 3-14. For each dataset, we first randomly sample a bunch of latent codes, then select $z_1$ and $z_2$ such that $\log \det J^\top J (z_1)$ and $\log \det J^\top J (z_2)$ give the minimum and maximum $\log \det$ that can be achieved among all latent codes. Then, for $z_1$, we generate samples $x = g(z)$ around $x_1 = g(z_1)$ by randomly sample $z$ from $\mathcal{Z}_1 \subset \mathcal{Z}$, where $g$ is the generator, $\mathcal{Z}$ is the latent space, and $\mathcal{Z}_1$ is the latent subspace around $z_1$ spanned by the 5 latent dimensions with the smallest Jacobian column vector norm. Similarly, we obtain generated samples $x = g(z)$ around $x_2 = g(z_2)$ by randomly sample $z$ from a subspace $\mathcal{Z}_2 \subset \mathcal{Z}$ such that $\mathcal{Z}_2$ is the latent subspace around $z_2$ spanned by the 5 latent dimensions with the largest Jacobian column vector norm. We visualize generated samples in such a manner since it can intuitively show how generator uniformity is related with the uniform diversity, and how generator uniformity reduces mode collapse.

In Fig. 3-14, the visualization of generated samples of baselines are shown in the top row. We found that for baselines, $\log \det J^\top J (z_1) \approx 0$, therefore the model tends to collapse on the subspace $\mathcal{Z}_1$ around $z_1$, which leads to almost identical generated samples around $x_1$. Meanwhile for $z_2$, the value of $\log \det J^\top J (z_2)$ tends to be very large, which leads to drastic change of images when varying $z$ in the subspace $\mathcal{Z}_2$ around $z_2$, and the generative qualities are commonly inferior, especially for the generators trained on natural image datasets, see Fig. 9-14. We also found that estimated probability densities of samples around $x_1$ and $x_2$ commonly tend to be large and small respectively, which is also consistent with our analysis on the relationship between generator uniformity and uniform diversity provided in main text. In terms of qualitative results of our UniGANs provided in the bottom rows, we see that for latent codes varying in both $\mathcal{Z}_1$ and $\mathcal{Z}_2$, the generators result in moderate change of generated images, therefore lead to a more smooth generation compared to baseline models. The qualitative results provided in Fig. 9-14 also demonstrate the generative quality of our UniGANs.

### 3.2.3 Ablation Study

We firstly show that our hierarchical architecture design of our NF-based generator indeed accelerates training and greatly reduces the amount of model parameters. We set up a counterpart of our NF-based generator in the case of $K = D$, *i.e.*, a Glow [3] generator, where $K$ and $D$ are the dimensionality of the latent space and data space, respectively. As we shown in Tbl. 3, for our UniGAN generator in the case of $K = 64$ and $D = 3 \times 64 \times 64$, the total amount of parameters of the generator is $8.889$M. However, for the Glow generator in the case of $K = D = 3 \times 64 \times 64$, the total amount of generator parameters is $61.214$M, which is far larger than ours. We also found that the training time and the memory usage of a Glow generator is about 7 times that of ours. Note that all settings except for the dimensionalities are the same for the Glow generator and our UniGAN generator.

**Sample efficiency** We compare our proposed LT algorithm to existing methods including LGAN [8] and StyleGAN2 [21] regularizations (*i.e.*, $\mathcal{L}_{\text{lgan}}$ and $\mathcal{L}_{\text{stylegan2}}$ in Eq. (86) and Eq. (87)) in terms of the sample efficiency on isometry learning. The results are shown in Fig. 2(a). For $\mathcal{L}_{\text{lgan}}$, we sample $1, 2, 3, 4, 5$ latent dimensions given each latent code $z$ (see Eq. (86)), while for $\mathcal{L}_{\text{stylegan2}}$, we sample $1, 2, 3, 4, 5$ noises given each latent code $z$ (see Eq. (87)). For both regularizations, the higher the sample counts, the better the results. Since our LT does not involve an additional sampling process given a latent code, the result is plotted as a single point. We learn that compared to $\mathcal{L}_{\text{stylegan2}}$, the training time cost of $\mathcal{L}_{\text{lgan}}$ is superior, which is reasonable because $\mathcal{L}_{\text{lgan}}$ does not involve a second

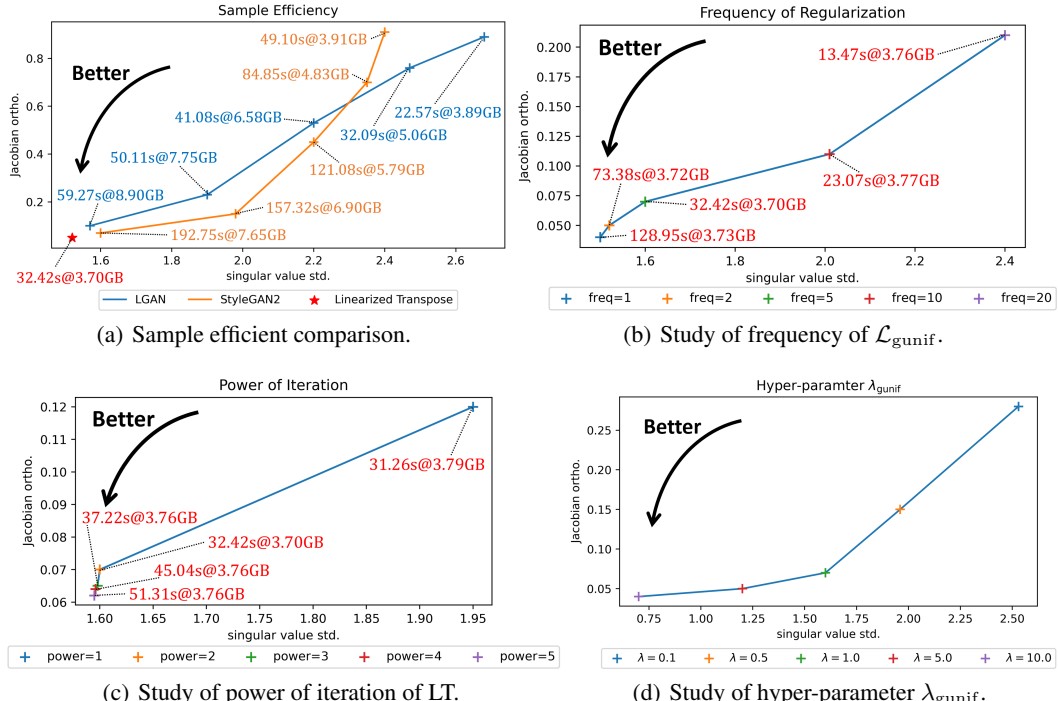

(a) Sample efficient comparison.

(b) Study of frequency of $\mathcal{L}_{\mathrm{gunif}}$.

(c) Study of power of iteration of LT.

(d) Study of hyper-parameter $\lambda_{\mathrm{gunif}}$.

Figure 2: Ablation study results on sample efficiency in isometry learning, frequency of regularization $\mathcal{L}_{\mathrm{gunif}}$, power of iteration of LT, and hyper-parameter $\lambda_{\mathrm{gunif}}$. The effectiveness of the regularization is reflected by the Jacobian orthogonality (*i.e.*, $y$-axis) versus the singular value standard deviation (*i.e.*, $x$-axis). The Jacobian orthogonality is measured by $\mathcal{O} \triangleq \mathbb{E}_{z \sim p_z} \frac{\sum_{i \neq j} \left| \left(J^\top J\right)_{ij}(z)\right|}{(K-1)\sum_{i=1}^{K}\left|(J^\top J)_{ii}(z)\right|}$, namely the ratio of the averaged absolute value of the non-diagonal elements of $J^\top J(z)$ to the averaged absolute value of the diagonal elements of $J^\top J(z)$, hence the lower the $\mathcal{O}$, the more orthogonal the Jacobian. On the other hand, an isometry corresponds to a Jacobian such that all singular values are the same. Hence the lower the $\mathcal{O}$ and the lower the singular value std, the closer the generator to an isometry and the more effective the regularization. For each setting, we also show the *training time per* 100 *batches* $t$ versus the *memory usage* $m$ by marking $t@m$ in the plot.

derivative computation procedure like $\mathcal{L}_{\mathrm{stylegan2}}$. However, because $\mathcal{L}_{\mathrm{stylegan2}}$ can reuse the same forward computation graph for different noises when computing second derivatives using PyTorch, the memory usage of $\mathcal{L}_{\mathrm{stylegan2}}$ is much lower than that of $\mathcal{L}_{\mathrm{lgan}}$. Meanwhile for $\mathcal{L}_{\mathrm{lgan}}$, different latent dimensions $i$ cannot share forward computation process $\frac{g(z+\Delta_i)}{\varepsilon}$ (see Eq. (86)), therefore $\mathcal{L}_{\mathrm{lgan}}$ requires considerable memory footprint as the number of sampled latent dimensions increases. We learn from Fig. 2(a) that our LT achieves the best performance on isometry learning at the minimum cost of memory usage and a lower cost of training time. Note that in our LT Algorithm 1, for iteration index $m$ we only retain the computation graph of PyTorch for computing the second derivative at $m = M$, hence the second derivative computation is not involved for $m < M$, which greatly reduces the cost of training time compared to $\mathcal{L}_{\mathrm{stylegan2}}$ that also involves second derivative computation.

**Frequency of regularization**   We provide ablation study on the *frequency of regularization $s$* for our generator uniformity regularization $\mathcal{L}_{\mathrm{gunif}}$, see lazy regularization in Section 2.2.3. The results are provided in Fig. 2(b). We see that as $s$ decreases (*i.e.*, the regularization becomes more frequent), the result becomes better, which is reasonable. We see that as $s$ decreases, the training time increases meanwhile the memory usage remains the same, and we can better trade-off between the effectiveness of the regularization and training time cost by selecting $s = 5$.

**Power of iteration**    We provide ablation study on the *power of iteration $M$* for our LT algorithm, see Algorithm 1. The results are provided in Fig. 2(c). As mentioned above, the computation graph of PyTorch of different iterations can be reused, therefore, the memory usage remain the same as $M$ increases. It is reasonable that the training time increases as $M$ increase, but we can still trade-off between the effectiveness of the regularization and training time cost by selecting $M = 2$.

**Hyper-parameter $\lambda_{\mathrm{gunif}}$**    We provide ablation study on the hyper-parameter $\lambda_{\mathrm{gunif}}$ for $\mathcal{L}_{\mathrm{gunif}}$ in Fig. 2(d). Changing $\lambda_{\mathrm{gunif}}$ does not lead to changes in training time or memory usage. We learn that compared to a model without involving $\mathcal{L}_{\mathrm{gunif}}$, imposing $\mathcal{L}_{\mathrm{gunif}}$ significantly improves performance on isometry learning, which verifies the effectiveness of our proposed regularization.

## 4    Limitations of Our Work

Our work adopts the *manifold hypothesis* that assumes that high dimensional data is concentrated on a lower dimensional manifold, and we propose to encourage the generator to be a local diffeomorphism between a $K$-dimensional latent space and a manifold residing in a high dimensional data space. To ease practical implementation, we encourage the generator to be an isometry whose Jacobian is an orthonormal matrix up to a constant. In such a case, the dimensionality $K$ should be carefully chosen, because on one hand, a $K$ lower than the ground-truth dimensionality of the data manifold may result in a generator that is unable to recover the manifold well, and on the other hand, a larger $K$ may lead to a generator with many redundant latent dimensions which may affect the effectiveness of isometry learning and increase the computational cost. Hence how to automatically determine the value of $K$ while learning the generator may be a notable research problem for future work.

Table 4: Quantitative comparison on MNIST dataset.

| Metric | MSGAN | VEEGAN | MDGAN | RegGAN | GAN | BiGAN | PacGAN |
|---|---|---|---|---|---|---|---|
| *Inception Score* ↑ | 3.71 | 3.73 | 3.70 | **3.72** | 3.69 | **3.72** | 3.71 |
| | **3.72** | **3.73** | **3.72** | 3.70 | **3.72** | 3.71 | **3.72** |
| | $7.8e^{-2}$ | $2.6e^{-1}$ | $2.2e^{-2}$ | $3.1e^{-1}$ | $3.2e^{-2}$ | $5.7e^{-1}$ | $3.5e^{-1}$ |
| *Mode Score* ↑ | 3.68 | 3.70 | 3.67 | 3.71 | **3.67** | 3.67 | 3.70 |
| | **3.69** | **3.71** | **3.69** | **3.72** | 3.65 | **3.69** | **3.71** |
| | $9.2e^{-1}$ | $2.2e^{-1}$ | $3.2e^{-2}$ | $1.0e^{-1}$ | $6.1e^{-2}$ | $5.2e^{-2}$ | $2.5e^{-1}$ |
| *Number of Modes* ↑ | 10 | 10 | 10 | 10 | 10 | 10 | 10 |
| | 10 | 10 | 10 | 10 | 10 | 10 | 10 |
| | $4.4e^{-1}$ | $2.9e^{-1}$ | $2.9e^{-1}$ | $1.8e^{-1}$ | $5.0e^{-1}$ | $5.9e^{-1}$ | $4.0e^{-1}$ |
| *KL Divergence* ↓ | 0.07 | 0.08 | **0.07** | 0.06 | 0.07 | **0.05** | 0.06 |
| | **0.06** | **0.08** | 0.08 | **0.05** | **0.05** | 0.06 | **0.05** |
| | $7.8e^{-1}$ | $2.6e^{-1}$ | $5.6e^{-1}$ | $1.4e^{-1}$ | $7.4e^{-2}$ | $3.8e^{-1}$ | $2.6e^{-1}$ |
| *IvOM* ↓ | 0.20 | **0.21** | 0.21 | **0.19** | 0.23 | 0.22 | **0.20** |
| | **0.19** | 0.22 | **0.20** | 0.20 | **0.21** | **0.21** | 0.21 |
| | $1.7e^{-1}$ | $8.2e^{-2}$ | $5.7e^{-1}$ | $4.3e^{-1}$ | $3.7e^{-1}$ | $4.8e^{-1}$ | $2.1e^{-1}$ |
| *Pairwise Distance* ↑ | **0.11** | 0.10 | 0.12 | **0.15** | 0.13 | **0.11** | 0.11 |
| | 0.10 | **0.12** | **0.13** | 0.12 | **0.14** | 0.10 | **0.12** |
| | $2.8e^{-1}$ | $1.3e^{-1}$ | $3.1e^{-1}$ | $8.6e^{-2}$ | $3.4e^{-1}$ | $7.3e^{-1}$ | $4.8e^{-1}$ |
| *NDB* ↓ | 20.9 | **20.2** | 22.0 | 20.3 | 21.3 | 20.9 | 20.8 |
| | **19.3** | 20.8 | **19.9** | **19.5** | **20.2** | **19.6** | **20.3** |
| | $8.5e^{-2}$ | $2.5e^{-1}$ | $5.6e^{-2}$ | $2.9e^{-1}$ | $5.6e^{-1}$ | $9.2e^{-2}$ | $2.5e^{-1}$ |
| *JSD* ↓ | 0.13 | 0.14 | 0.13 | **0.12** | **0.13** | 0.11 | **0.12** |
| | **0.12** | **0.11** | **0.12** | 0.14 | 0.14 | **0.10** | 0.15 |
| | $7.0e^{-1}$ | $9.3e^{-2}$ | $4.9e^{-1}$ | $3.6e^{-1}$ | $4.4e^{-1}$ | $1.4e^{-1}$ | $7.1e^{-2}$ |
| *udiv* ↑ | 0.59 | 0.58 | 0.60 | 0.61 | 0.59 | 0.59 | 0.60 |
| | **0.66** | **0.67** | **0.67** | **0.66** | **0.68** | **0.68** | **0.67** |
| | $2.1e^{-5}$ | $5.9e^{-6}$ | $9.7e^{-6}$ | $1.1e^{-5}$ | $8.4e^{-6}$ | $2.1e^{-5}$ | $7.9e^{-5}$ |
| *gunif* ↓ | 0.87 | 0.90 | 0.93 | 0.86 | 0.88 | 0.87 | 0.89 |
| | **0.62** | **0.59** | **0.63** | **0.61** | **0.60** | **0.59** | **0.61** |
| | $8.6e^{-5}$ | $9.9e^{-6}$ | $5.0e^{-5}$ | $3.8e^{-6}$ | $6.5e^{-5}$ | $2.1e^{-5}$ | $8.1e^{-6}$ |

Table 5: Quantitative comparison on colored MNIST dataset.

| Metric | MSGAN | VEEGAN | MDGAN | RegGAN | GAN | BiGAN | PacGAN |
|---|---|---|---|---|---|---|---|
| *Inception Score* ↑ | 3.70 | 3.69 | **3.74** | 3.71 | **3.73** | 3.72 | **3.73** |
| | **3.73** | **3.72** | 3.72 | **3.75** | 3.72 | **3.73** | 3.72 |
| | $5.7e^{-2}$ | $1.3e^{-2}$ | $1.4e^{-1}$ | $2.6e^{-2}$ | $1.0e^{-1}$ | $2.5e^{-1}$ | $4.7e^{-1}$ |
| *Mode Score* ↑ | **3.73** | 3.72 | **3.73** | **3.74** | 3.70 | 3.69 | **3.74** |
| | 3.70 | **3.73** | 3.71 | 3.72 | **3.73** | **3.70** | 3.69 |
| | $1.4e^{-1}$ | $3.0e^{-1}$ | $2.4e^{-2}$ | $3.5e^{-1}$ | $6.6e^{-2}$ | $3.0e^{-1}$ | $1.2e^{-2}$ |
| *Number of Modes* ↑ | 10 | 10 | 10 | 10 | 10 | 10 | 10 |
| | 10 | 10 | 10 | 10 | 10 | 10 | 10 |
| | $9.0e^{-1}$ | $4.1e^{-2}$ | $9.1e^{-1}$ | $8.9e^{-1}$ | $6.2e^{-1}$ | $6.3e^{-1}$ | $4.3e^{-1}$ |
| *KL Divergence* ↓ | **0.12** | 0.11 | 0.09 | **0.11** | 0.07 | **0.10** | **0.11** |
| | 0.13 | **0.10** | **0.07** | 0.12 | **0.06** | 0.12 | 0.12 |
| | $1.1e^{-1}$ | $2.8e^{-1}$ | $3.7e^{-2}$ | $7.4e^{-1}$ | $2.0e^{-1}$ | $4.8e^{-1}$ | $3.1e^{-2}$ |
| *IvOM* ↓ | **0.56** | 0.54 | 0.58 | **0.55** | **0.53** | 0.56 | 0.55 |
| | 0.57 | **0.53** | **0.57** | 0.56 | 0.54 | **0.55** | **0.54** |
| | $4.2e^{-2}$ | $8.1e^{-1}$ | $2.1e^{-1}$ | $7.5e^{-1}$ | $1.9e^{-1}$ | $2.2e^{-1}$ | $8.4e^{-2}$ |
| *Pairwise Distance* ↑ | 0.09 | 0.07 | **0.11** | **0.08** | 0.07 | **0.11** | 0.09 |
| | **0.10** | **0.09** | 0.10 | 0.07 | **0.10** | 0.10 | **0.10** |
| | $8.7e^{-2}$ | $7.4e^{-2}$ | $3.7e^{-1}$ | $7.7e^{-1}$ | $4.5e^{-1}$ | $3.6e^{-1}$ | $4.8e^{-1}$ |
| *NDB* ↓ | 20.4 | **19.3** | 18.4 | 19.6 | 18.1 | **19.4** | 18.5 |
| | **18.2** | 19.5 | **18.3** | **19.0** | **17.9** | 20.0 | **18.1** |
| | $2.1e^{-1}$ | $8.6e^{-2}$ | $4.7e^{-1}$ | $6.5e^{-1}$ | $9.2e^{-2}$ | $7.2e^{-1}$ | $3.7e^{-1}$ |
| *JSD* ↓ | 0.11 | 0.12 | **0.09** | **0.11** | 0.10 | 0.12 | **0.09** |
| | **0.09** | **0.10** | 0.10 | 0.12 | **0.09** | **0.10** | 0.11 |
| | $1.6e^{-1}$ | $1.1e^{-1}$ | $2.9e^{-1}$ | $3.7e^{-1}$ | $3.1e^{-1}$ | $8.5e^{-2}$ | $4.7e^{-1}$ |
| *udiv* ↑ | 0.65 | 0.63 | 0.67 | 0.64 | 0.65 | 0.66 | 0.64 |
| | **0.75** | **0.77** | **0.78** | **0.76** | **0.75** | **0.77** | **0.76** |
| | $4.7e^{-6}$ | $2.2e^{-5}$ | $4.9e^{-6}$ | $9.7e^{-6}$ | $4.3e^{-6}$ | $3.1e^{-6}$ | $2.0e^{-5}$ |
| *gunif* ↓ | 0.93 | 0.87 | 0.89 | 0.91 | 0.92 | 0.90 | 0.89 |
| | **0.67** | **0.65** | **0.66** | **0.65** | **0.67** | **0.65** | **0.65** |
| | $1.1e^{-6}$ | $7.8e^{-5}$ | $9.2e^{-5}$ | $2.8e^{-5}$ | $2.7e^{-6}$ | $9.5e^{-5}$ | $1.6e^{-5}$ |

Table 6: Quantitative comparison on FashionMNIST dataset.

| Metric | MSGAN | VEEGAN | MDGAN | RegGAN | GAN | BiGAN | PacGAN |
|---|---|---|---|---|---|---|---|
| *Inception Score* ↑ | **3.73** | **3.75** | 3.72 | 3.74 | 3.72 | **3.71** | 3.70 |
| | 3.71 | 3.73 | **3.75** | **3.76** | **3.74** | 3.70 | **3.76** |
| | $2.8e^{-1}$ | $5.9e^{-2}$ | $3.7e^{-1}$ | $1.7e^{-1}$ | $6.8e^{-1}$ | $3.1e^{-1}$ | $2.5e^{-2}$ |
| *Mode Score* ↑ | 3.71 | 3.68 | 3.69 | **3.71** | **3.72** | 3.68 | 3.69 |
| | **3.72** | **3.69** | **3.70** | 3.69 | 3.70 | **3.71** | **3.72** |
| | $1.6e^{-1}$ | $4.1e^{-1}$ | $4.3e^{-1}$ | $3.4e^{-1}$ | $2.0e^{-1}$ | $8.5e^{-2}$ | $3.2e^{-1}$ |
| *Number of Modes* ↑ | 10 | 10 | 10 | 10 | 10 | 10 | 10 |
| | 10 | 10 | 10 | 10 | 10 | 10 | 10 |
| | $1.4e^{-1}$ | $8.2e^{-1}$ | $9.7e^{-1}$ | $8.7e^{-1}$ | $4.8e^{-1}$ | $9.4e^{-2}$ | $8.5e^{-1}$ |
| *KL Divergence* ↓ | 0.08 | **0.10** | 0.08 | 0.09 | 0.11 | 0.08 | **0.09** |
| | **0.07** | 0.11 | **0.06** | **0.07** | **0.09** | **0.07** | 0.10 |
| | $3.2e^{-1}$ | $7.9e^{-1}$ | $1.4e^{-1}$ | $9.1e^{-2}$ | $1.5e^{-1}$ | $3.9e^{-1}$ | $8.5e^{-1}$ |
| *IvOM* ↓ | 0.21 | **0.20** | 0.23 | **0.22** | 0.21 | **0.20** | **0.21** |
| | **0.20** | 0.21 | **0.21** | 0.23 | **0.20** | 0.22 | 0.23 |
| | $4.3e^{-1}$ | $9.0e^{-2}$ | $4.2e^{-1}$ | $1.7e^{-1}$ | $2.3e^{-1}$ | $4.4e^{-1}$ | $8.3e^{-2}$ |
| *Pairwise Distance* ↑ | 0.19 | 0.18 | 0.16 | **0.20** | **0.22** | 0.17 | **0.20** |
| | **0.20** | **0.21** | **0.19** | 0.18 | 0.19 | **0.18** | 0.19 |
| | $9.4e^{-2}$ | $8.1e^{-1}$ | $7.8e^{-1}$ | $5.7e^{-1}$ | $3.8e^{-1}$ | $3.3e^{-1}$ | $1.9e^{-1}$ |
| *NDB* ↓ | 23.1 | 22.7 | **21.3** | 22.9 | **21.5** | 22.3 | 21.3 |
| | **21.6** | **21.0** | 21.4 | **20.8** | 22.3 | **21.8** | **20.0** |
| | $2.2e^{-1}$ | $1.4e^{-1}$ | $9.4e^{-2}$ | $1.6e^{-1}$ | $7.4e^{-1}$ | $8.3e^{-1}$ | $1.5e^{-1}$ |
| *JSD* ↓ | **0.17** | 0.17 | 0.16 | 0.17 | **0.16** | 0.18 | 0.17 |
| | 0.19 | **0.15** | **0.14** | **0.14** | 0.17 | **0.16** | **0.15** |
| | $3.6e^{-1}$ | $1.7e^{-1}$ | $9.6e^{-2}$ | $1.0e^{-1}$ | $2.8e^{-1}$ | $8.1e^{-1}$ | $8.4e^{-2}$ |
| *udiv* ↑ | 0.65 | 0.64 | 0.64 | 0.65 | 0.63 | 0.65 | 0.65 |
| | **0.75** | **0.76** | **0.77** | **0.75** | **0.77** | **0.75** | **0.76** |
| | $1.8e^{-6}$ | $5.0e^{-6}$ | $9.8e^{-7}$ | $2.0e^{-5}$ | $4.4e^{-6}$ | $6.2e^{-5}$ | $4.8e^{-5}$ |
| *gunif* ↓ | 1.09 | 1.05 | 1.07 | 1.08 | 1.07 | 1.09 | 1.07 |
| | **0.75** | **0.73** | **0.73** | **0.75** | **0.76** | **0.74** | **0.73** |
| | $1.9e^{-6}$ | $8.2e^{-5}$ | $9.8e^{-6}$ | $1.2e^{-5}$ | $5.1e^{-5}$ | $1.7e^{-5}$ | $2.9e^{-5}$ |

Table 7: Quantitative comparison on colored FashionMNIST dataset.

| Metric | MSGAN | VEEGAN | MDGAN | RegGAN | GAN | BiGAN | PacGAN |
|---|---|---|---|---|---|---|---|
| *Inception Score* ↑ | 3.71 | 3.74 | **3.74** | 3.72 | 3.75 | 3.72 | **3.73** |
| | **3.74** | **3.75** | 3.73 | **3.74** | **3.77** | **3.75** | 3.69 |
| | $3.2e^{-1}$ | $8.6e^{-2}$ | $3.5e^{-1}$ | $1.3e^{-1}$ | $3.2e^{-1}$ | $6.0e^{-1}$ | $7.3e^{-1}$ |
| *Mode Score* ↑ | 3.70 | 3.71 | **3.73** | 3.70 | **3.73** | 3.71 | **3.74** |
| | **3.72** | **3.75** | 3.71 | **3.71** | 3.72 | **3.72** | 3.69 |
| | $2.0e^{-1}$ | $3.3e^{-1}$ | $1.4e^{-1}$ | $3.2e^{-1}$ | $3.8e^{-1}$ | $6.5e^{-2}$ | $1.9e^{-1}$ |
| *Number of Modes* ↑ | 10 | 10 | 10 | 10 | 10 | 10 | 10 |
| | 10 | 10 | 10 | 10 | 10 | 10 | 10 |
| | $7.1e^{-1}$ | $4.2e^{-1}$ | $2.0e^{-1}$ | $3.4e^{-1}$ | $4.4e^{-1}$ | $8.0e^{-1}$ | $8.5e^{-1}$ |
| *KL Divergence* ↓ | **0.10** | **0.07** | 0.09 | **0.07** | **0.08** | 0.12 | 0.10 |
| | 0.11 | 0.08 | **0.08** | 0.08 | 0.12 | **0.09** | **0.08** |
| | $2.8e^{-1}$ | $9.7e^{-2}$ | $3.4e^{-1}$ | $8.6e^{-2}$ | $4.8e^{-1}$ | $7.2e^{-1}$ | $3.7e^{-1}$ |
| *IvOM* ↓ | 1.21 | 1.23 | **1.19** | **1.20** | 1.22 | 1.22 | **1.21** |
| | **1.19** | **1.20** | 1.20 | 1.21 | **1.19** | **1.21** | 1.22 |
| | $3.4e^{-1}$ | $5.4e^{-1}$ | $9.9e^{-2}$ | $4.5e^{-1}$ | $8.5e^{-2}$ | $1.1e^{-1}$ | $3.9e^{-1}$ |
| *Pairwise Distance* ↑ | **0.13** | 0.11 | 0.10 | **0.15** | 0.12 | **0.13** | 0.11 |
| | 0.12 | **0.13** | **0.13** | 0.14 | **0.13** | 0.11 | **0.12** |
| | $2.4e^{-1}$ | $6.3e^{-1}$ | $3.1e^{-1}$ | $2.3e^{-1}$ | $3.7e^{-1}$ | $9.5e^{-2}$ | $5.1e^{-1}$ |
| *NDB* ↓ | 22.0 | 22.7 | 23.9 | **21.7** | **22.9** | **22.1** | 22.3 |
| | **21.2** | **22.1** | **21.3** | 22.2 | 23.1 | 23.0 | **21.9** |
| | $5.1e^{-1}$ | $4.6e^{-1}$ | $3.3e^{-2}$ | $6.8e^{-1}$ | $2.4e^{-1}$ | $2.3e^{-1}$ | $2.1e^{-1}$ |
| *JSD* ↓ | **0.18** | 0.19 | 0.20 | **0.18** | **0.17** | 0.18 | **0.19** |
| | 0.19 | **0.17** | **0.18** | 0.19 | 0.19 | **0.17** | 0.20 |
| | $4.8e^{-1}$ | $4.0e^{-1}$ | $3.1e^{-1}$ | $8.6e^{-2}$ | $9.5e^{-2}$ | $5.7e^{-1}$ | $3.6e^{-1}$ |
| *udiv* ↑ | 0.51 | 0.49 | 0.50 | 0.51 | 0.52 | 0.50 | 0.50 |
| | **0.67** | **0.69** | **0.67** | **0.70** | **0.65** | **0.68** | **0.68** |
| | $1.7e^{-5}$ | $2.9e^{-5}$ | $9.5e^{-6}$ | $1.5e^{-5}$ | $1.1e^{-5}$ | $7.0e^{-5}$ | $2.8e^{-5}$ |
| *gunif* ↓ | 1.24 | 1.28 | 1.19 | 1.23 | 1.22 | 1.18 | 1.20 |
| | **0.81** | **0.80** | **0.79** | **0.78** | **0.81** | **0.80** | **0.80** |
| | $3.6e^{-5}$ | $1.4e^{-5}$ | $1.1e^{-5}$ | $3.9e^{-5}$ | $2.0e^{-5}$ | $2.1e^{-5}$ | $3.9e^{-5}$ |

Table 8: Quantitative comparison on FashionMNIST and partial MNIST [44] dataset.

| Metric | MSGAN | VEEGAN | MDGAN | RegGAN | GAN | BiGAN | PacGAN |
|---|---|---|---|---|---|---|---|
| *Inception Score* ↑ | 3.68 | **3.71** | 3.69 | **3.70** | 3.70 | **3.71** | 3.68 |
| | **3.69** | 3.69 | **3.70** | 3.68 | **3.71** | 3.70 | **3.69** |
| | $8.4e^{-1}$ | $5.5e^{-2}$ | $9.0e^{-2}$ | $7.8e^{-2}$ | $8.6e^{-2}$ | $7.8e^{-2}$ | $6.2e^{-1}$ |
| *Mode Score* ↑ | 3.67 | **3.69** | 3.68 | **3.68** | 3.69 | **3.70** | **3.68** |
| | **3.68** | 3.68 | **3.69** | 3.66 | **3.70** | 3.69 | 3.67 |
| | $4.4e^{-1}$ | $5.6e^{-1}$ | $9.8e^{-2}$ | $7.2e^{-2}$ | $5.4e^{-2}$ | $8.6e^{-2}$ | $6.5e^{-1}$ |
| *Number of Modes* ↑ | 11 | 11 | 11 | 11 | 11 | 11 | 11 |
| | 11 | 11 | 11 | 11 | 11 | 11 | 11 |
| | $8.9e^{-1}$ | $6.5e^{-1}$ | $9.4e^{-1}$ | $8.7e^{-1}$ | $9.7e^{-1}$ | $8.8e^{-1}$ | $7.9e^{-1}$ |
| *KL Divergence* ↓ | **0.12** | 0.15 | 0.11 | 0.16 | 0.13 | 0.11 | **0.12** |
| | 0.13 | **0.12** | **0.10** | **0.15** | **0.12** | **0.10** | 0.14 |
| | $6.5e^{-2}$ | $1.6e^{-2}$ | $5.0e^{-1}$ | $4.2e^{-1}$ | $3.9e^{-1}$ | $1.3e^{-1}$ | $7.8e^{-2}$ |
| *IvOM* ↓ | 1.24 | 1.22 | **1.21** | 1.23 | 1.21 | 1.23 | **1.22** |
| | **1.22** | **1.21** | 1.23 | **1.22** | **1.20** | **1.22** | 1.23 |
| | $7.6e^{-2}$ | $6.2e^{-2}$ | $4.7e^{-2}$ | $7.5e^{-1}$ | $1.4e^{-1}$ | $8.5e^{-2}$ | $2.1e^{-1}$ |
| *Pairwise Distance* ↑ | **0.16** | 0.14 | 0.15 | **0.17** | **0.15** | 0.14 | **0.15** |
| | 0.15 | **0.15** | **0.17** | 0.16 | 0.14 | **0.16** | 0.14 |
| | $9.8e^{-2}$ | $8.6e^{-1}$ | $4.9e^{-2}$ | $8.3e^{-2}$ | $1.5e^{-1}$ | $5.3e^{-2}$ | $8.4e^{-1}$ |
| *NDB* ↓ | **22.1** | 23.5 | 24.3 | **22.4** | 23.7 | 22.9 | **23.4** |
| | 22.8 | **23.2** | **23.1** | 23.3 | **23.5** | **22.4** | 23.6 |
| | $6.4e^{-2}$ | $1.1e^{-1}$ | $5.3e^{-1}$ | $7.8e^{-2}$ | $4.1e^{-1}$ | $8.2e^{-2}$ | $9.1e^{-2}$ |
| *JSD* ↓ | 0.22 | 0.21 | **0.21** | **0.19** | **0.20** | 0.19 | 0.20 |
| | **0.20** | **0.19** | 0.22 | 0.20 | 0.21 | **0.18** | **0.19** |
| | $5.2e^{-2}$ | $1.8e^{-1}$ | $7.3e^{-2}$ | $2.1e^{-1}$ | $4.2e^{-2}$ | $7.5e^{-2}$ | $1.7e^{-1}$ |
| *udiv* ↑ | 0.48 | 0.47 | 0.48 | 0.46 | 0.49 | 0.45 | 0.47 |
| | **0.61** | **0.62** | **0.59** | **0.64** | **0.66** | **0.65** | **0.65** |
| | $6.3e^{-6}$ | $8.8e^{-7}$ | $3.7e^{-6}$ | $8.2e^{-6}$ | $5.2e^{-6}$ | $1.7e^{-6}$ | $7.7e^{-5}$ |
| *gunif* ↓ | 1.28 | 1.31 | 1.27 | 1.30 | 1.29 | 1.27 | 1.26 |
| | **0.84** | **0.85** | **0.80** | **0.81** | **0.83** | **0.82** | **0.83** |
| | $6.2e^{-6}$ | $8.7e^{-7}$ | $4.1e^{-6}$ | $8.1e^{-5}$ | $1.4e^{-7}$ | $6.9e^{-6}$ | $7.5e^{-6}$ |

Table 9: Quantitative comparison on stacked-MNIST [37] dataset.

| Metric | MSGAN | VEEGAN | MDGAN | RegGAN | GAN | BiGAN | PacGAN |
|---|---|---|---|---|---|---|---|
| *Inception Score* ↑ | 2.59 | 2.57 | **2.67** | 2.53 | **2.73** | 2.51 | 2.58 |
| | **2.64** | **2.62** | 2.59 | **2.58** | 2.68 | **2.57** | **2.62** |
| | $2.1e^{-2}$ | $4.7e^{-1}$ | $8.1e^{-2}$ | $9.5e^{-2}$ | $1.7e^{-1}$ | $8.2e^{-2}$ | $2.8e^{-1}$ |
| *Mode Score* ↑ | 2.58 | 2.57 | **2.64** | 2.53 | **2.69** | 2.50 | 2.57 |
| | **2.62** | **2.61** | 2.58 | **2.55** | 2.67 | **2.55** | **2.61** |
| | $6.2e^{-2}$ | $4.1e^{-2}$ | $3.1e^{-1}$ | $9.3e^{-2}$ | $3.8e^{-1}$ | $5.8e^{-2}$ | $3.7e^{-2}$ |
| *Number of Modes* ↑ | 979 | **980** | 980 | 981 | **980** | 988 | **984** |
| | **985** | 979 | **981** | **984** | 979 | **992** | 981 |
| | $6.2e^{-1}$ | $8.7e^{-1}$ | $5.2e^{-1}$ | $7.4e^{-1}$ | $6.3e^{-1}$ | $8.1e^{-1}$ | $9.5e^{-1}$ |
| *KL Divergence* ↓ | **0.15** | 0.17 | **0.16** | 0.18 | **0.15** | 0.19 | 0.18 |
| | 0.16 | **0.16** | 0.18 | **0.17** | 0.16 | **0.17** | **0.17** |
| | $1.3e^{-1}$ | $5.2e^{-2}$ | $7.2e^{-2}$ | $6.2e^{-2}$ | $7.2e^{-2}$ | $8.1e^{-2}$ | $9.1e^{-2}$ |
| *IvOM* ↓ | 1.32 | **1.31** | **1.29** | 1.34 | 1.30 | 1.32 | 1.35 |
| | **1.31** | 1.33 | 1.30 | **1.32** | 1.29 | 1.31 | 1.34 |
| | $8.4e^{-2}$ | $9.7e^{-2}$ | $6.2e^{-2}$ | $1.1e^{-1}$ | $7.2e^{-2}$ | $4.2e^{-1}$ | $7.2e^{-2}$ |
| *Pairwise Distance* ↑ | 0.21 | 0.18 | 0.17 | **0.18** | **0.20** | 0.19 | 0.18 |
| | **0.23** | **0.19** | 0.17 | 0.17 | 0.19 | **0.20** | **0.19** |
| | $5.2e^{-2}$ | $2.3e^{-1}$ | $6.8e^{-2}$ | $9.5e^{-2}$ | $4.2e^{-2}$ | $7.2e^{-2}$ | $1.5e^{-1}$ |
| *NDB* ↓ | **23.1** | 24.2 | 24.8 | **24.1** | **23.6** | 23.9 | 24.5 |
| | 23.4 | **24.1** | **23.9** | 24.3 | 23.8 | **23.4** | **24.0** |
| | $8.5e^{-2}$ | $1.3e^{-1}$ | $5.3e^{-2}$ | $6.5e^{-2}$ | $7.2e^{-2}$ | $2.4e^{-1}$ | $5.8e^{-2}$ |
| *JSD* ↓ | 0.25 | **0.24** | 0.23 | 0.25 | **0.22** | 0.24 | 0.26 |
| | **0.24** | 0.26 | **0.21** | **0.24** | 0.25 | **0.23** | **0.25** |
| | $7.2e^{-2}$ | $9.5e^{-2}$ | $2.6e^{-1}$ | $1.8e^{-1}$ | $8.3e^{-2}$ | $2.6e^{-1}$ | $1.8e^{-1}$ |
| *udiv* ↑ | 0.42 | 0.41 | 0.39 | 0.45 | 0.44 | 0.46 | 0.44 |
| | **0.58** | **0.61** | **0.62** | **0.61** | **0.63** | **0.62** | **0.60** |
| | $6.3e^{-7}$ | $4.1e^{-6}$ | $7.9e^{-7}$ | $5.7e^{-6}$ | $8.1e^{-7}$ | $5.2e^{-7}$ | $1.9e^{-5}$ |
| *gunif* ↓ | 1.34 | 1.39 | 1.35 | 1.41 | 1.38 | 1.29 | 1.34 |
| | **0.81** | **0.80** | **0.83** | **0.79** | **0.82** | **0.81** | **0.80** |
| | $7.6e^{-6}$ | $2.3e^{-5}$ | $1.8e^{-7}$ | $9.2e^{-7}$ | $8.8e^{-7}$ | $9.3e^{-7}$ | $1.5e^{-6}$ |

Table 10: Quantitative comparison on CIFAR10 dataset.

| Metric | MSGAN | VEEGAN | MDGAN | RegGAN | GAN | BiGAN | PacGAN |
|---|---|---|---|---|---|---|---|
| *Inception Score* ↑ | 3.74 | 3.77 | **3.80** | 3.76 | 3.77 | **3.78** | 3.75 |
| | **3.78** | **3.78** | 3.74 | **3.77** | **3.79** | 3.76 | **3.77** |
| | $9.5e^{-2}$ | $6.8e^{-1}$ | $1.7e^{-1}$ | $5.2e^{-1}$ | $1.4e^{-1}$ | $1.1e^{-1}$ | $3.1e^{-1}$ |
| *Mode Score* ↑ | **3.75** | 3.73 | 3.73 | **3.78** | 3.75 | 3.74 | 3.76 |
| | 3.73 | **3.74** | **3.78** | 3.75 | **3.76** | **3.78** | **3.77** |
| | $3.2e^{-1}$ | $6.2e^{-1}$ | $4.9e^{-2}$ | $9.5e^{-2}$ | $2.5e^{-1}$ | $1.5e^{-1}$ | $5.6e^{-1}$ |
| *Number of Modes* ↑ | 10 | 10 | 10 | 10 | 10 | 10 | 10 |
| | 10 | 10 | 10 | 10 | 10 | 10 | 10 |
| | $2.7e^{-1}$ | $9.9e^{-1}$ | $4.5e^{-1}$ | $7.4e^{-2}$ | $6.5e^{-1}$ | $6.1e^{-1}$ | $3.0e^{-1}$ |
| *KL Divergence* ↓ | **0.05** | 0.04 | 0.06 | 0.05 | 0.07 | **0.03** | 0.04 |
| | 0.07 | **0.03** | **0.05** | **0.04** | **0.06** | 0.04 | **0.03** |
| | $8.8e^{-2}$ | $3.3e^{-1}$ | $6.9e^{-1}$ | $1.8e^{-1}$ | $4.9e^{-2}$ | $1.0e^{-1}$ | $8.3e^{-1}$ |
| *IvOM* ↓ | 0.20 | **0.19** | 0.20 | 0.23 | 0.22 | **0.19** | **0.20** |
| | **0.19** | 0.20 | **0.19** | **0.21** | **0.20** | 0.20 | 0.21 |
| | $1.2e^{-1}$ | $5.4e^{-1}$ | $9.8e^{-2}$ | $1.6e^{-1}$ | $4.0e^{-1}$ | $8.5e^{-2}$ | $7.3e^{-1}$ |
| *Pairwise Distance* ↑ | 0.22 | **0.25** | 0.23 | 0.19 | 0.21 | 0.22 | **0.22** |
| | **0.23** | 0.23 | **0.24** | **0.20** | **0.22** | **0.23** | 0.20 |
| | $5.3e^{-1}$ | $1.4e^{-1}$ | $4.5e^{-1}$ | $1.2e^{-1}$ | $9.6e^{-2}$ | $1.2e^{-1}$ | $1.0e^{-1}$ |
| *NDB* ↓ | 22.1 | **23.7** | 25.2 | 22.8 | 23.9 | **22.0** | **22.2** |
| | **21.8** | 24.5 | **22.6** | **21.0** | **21.9** | 23.5 | 23.8 |
| | $4.0e^{-1}$ | $1.6e^{-1}$ | $1.2e^{-1}$ | $4.7e^{-1}$ | $9.1e^{-2}$ | $5.6e^{-1}$ | $1.2e^{-1}$ |
| *JSD* ↓ | **0.12** | 0.17 | 0.13 | **0.14** | 0.13 | 0.15 | **0.13** |
| | 0.14 | **0.12** | **0.11** | 0.15 | **0.12** | **0.14** | 0.16 |
| | $6.2e^{-1}$ | $8.5e^{-2}$ | $6.8e^{-1}$ | $1.3e^{-1}$ | $4.1e^{-1}$ | $3.2e^{-1}$ | $1.9e^{-1}$ |
| *udiv* ↑ | 0.61 | 0.58 | 0.65 | 0.61 | 0.60 | 0.62 | 0.61 |
| | **0.78** | **0.75** | **0.75** | **0.78** | **0.81** | **0.77** | **0.76** |
| | $1.2e^{-5}$ | $6.1e^{-9}$ | $9.6e^{-6}$ | $9.1e^{-6}$ | $1.8e^{-7}$ | $4.7e^{-5}$ | $1.0e^{-8}$ |
| *gunif* ↓ | 12.87 | 14.15 | 14.32 | 15.12 | 14.12 | 13.94 | 15.51 |
| | **7.91** | **8.17** | **9.64** | **8.12** | **9.12** | **8.26** | **9.19** |
| | $3.3e^{-6}$ | $9.8e^{-7}$ | $3.1e^{-6}$ | $4.6e^{-5}$ | $3.1e^{-5}$ | $2.4e^{-6}$ | $7.8e^{-7}$ |

Table 11: Quantitative comparison on CelebA dataset.

| Metric | MSGAN | VEEGAN | MDGAN | RegGAN | GAN | BiGAN | PacGAN |
|---|---|---|---|---|---|---|---|
| *FID* ↓ | 9.21 | **11.24** | **12.15** | **12.92** | 8.53 | 11.82 | **11.22** |
| | **9.18** | 11.87 | 13.04 | 13.12 | **8.22** | **11.00** | 12.43 |
| | $9.6e^{-2}$ | $1.8e^{-1}$ | $2.8e^{-1}$ | $1.3e^{-1}$ | $6.1e^{-2}$ | $8.0e^{-2}$ | $3.6e^{-1}$ |
| *LPIPS* ↑ | **0.27** | 0.23 | 0.24 | 0.24 | 0.22 | 0.22 | **0.25** |
| | 0.26 | **0.24** | **0.25** | **0.25** | **0.25** | **0.23** | 0.24 |
| | $7.7e^{-1}$ | $1.4e^{-1}$ | $1.0e^{-1}$ | $7.5e^{-2}$ | $1.9e^{-1}$ | $9.9e^{-2}$ | $3.1e^{-1}$ |
| *udiv* ↑ | 0.71 | 0.73 | 0.72 | 0.74 | 0.72 | 0.75 | 0.73 |
| | **0.85** | **0.86** | **0.85** | **0.86** | **0.83** | **0.85** | **0.86** |
| | $1.1e^{-6}$ | $4.9e^{-5}$ | $5.2e^{-9}$ | $6.0e^{-5}$ | $1.0e^{-4}$ | $4.1e^{-4}$ | $1.6e^{-7}$ |
| *gunif* ↓ | 28.76 | 24.85 | 27.96 | 30.72 | 28.30 | 35.25 | 30.88 |
| | **21.08** | **20.22** | **20.41** | **24.22** | **21.68** | **21.08** | **23.98** |
| | $2.9e^{-5}$ | $1.5e^{-5}$ | $1.1e^{-5}$ | $9.8e^{-6}$ | $4.8e^{-7}$ | $3.5e^{-9}$ | $1.2e^{-5}$ |

Table 12: Quantitative comparison on FFHQ dataset.

| Metric | MSGAN | VEEGAN | MDGAN | RegGAN | GAN | BiGAN | PacGAN |
|---|---|---|---|---|---|---|---|
| *FID* ↓ | **14.73** | 12.84 | 13.05 | 15.68 | 11.27 | **12.34** | **13.39** |
| | 15.49 | **11.03** | **12.21** | **14.89** | **11.22** | 12.73 | 14.83 |
| | $5.5e^{-1}$ | $8.1e^{-2}$ | $2.4e^{-1}$ | $1.9e^{-1}$ | $5.4e^{-1}$ | $7.3e^{-1}$ | $6.4e^{-1}$ |
| *LPIPS* ↑ | **0.34** | 0.31 | 0.29 | 0.30 | 0.28 | **0.27** | **0.29** |
| | 0.33 | **0.33** | **0.30** | **0.31** | **0.29** | 0.26 | 0.28 |
| | $1.2e^{-1}$ | $2.6e^{-1}$ | $7.2e^{-2}$ | $1.5e^{-1}$ | $9.7e^{-1}$ | $1.7e^{-1}$ | $2.4e^{-1}$ |
| *udiv* ↑ | 0.65 | 0.65 | 0.67 | 0.66 | 0.67 | 0.65 | 0.64 |
| | **0.80** | **0.79** | **0.78** | **0.79** | **0.78** | **0.81** | **0.79** |
| | $1.5e^{-4}$ | $7.6e^{-5}$ | $4.3e^{-3}$ | $1.1e^{-8}$ | $6.2e^{-8}$ | $3.1e^{-7}$ | $5.7e^{-5}$ |
| *gunif* ↓ | 34.78 | 33.04 | 32.19 | 30.88 | 36.06 | 33.30 | 35.69 |
| | **27.11** | **27.10** | **25.90** | **24.91** | **25.87** | **25.17** | **26.51** |
| | $8.0e^{-5}$ | $2.1e^{-3}$ | $6.2e^{-5}$ | $1.2e^{-6}$ | $7.1e^{-5}$ | $1.0e^{-8}$ | $7.1e^{-4}$ |

Table 13: Quantitative comparison on AFHQ Cat dataset.

| Metric | MSGAN | VEEGAN | MDGAN | RegGAN | GAN | BiGAN | PacGAN |
|---|---|---|---|---|---|---|---|
| *FID* ↓ | 26.91 | **24.82** | 25.57 | 23.33 | **23.28** | 23.69 | **25.19** |
| | **25.78** | 26.87 | **25.38** | **21.54** | 23.39 | **21.34** | 26.89 |
| | $7.7e^{-2}$ | $1.6e^{-1}$ | $2.7e^{-1}$ | $6.8e^{-1}$ | $1.6e^{-1}$ | $1.4e^{-1}$ | $6.9e^{-2}$ |
| *LPIPS* ↑ | **0.39** | **0.37** | 0.35 | **0.36** | 0.36 | 0.35 | 0.35 |
| | 0.38 | 0.35 | **0.36** | 0.35 | **0.37** | **0.38** | **0.36** |
| | $5.2e^{-1}$ | $1.7e^{-1}$ | $3.5e^{-1}$ | $1.2e^{-1}$ | $4.7e^{-1}$ | $9.1e^{-2}$ | $5.3e^{-1}$ |
| *udiv* ↑ | 0.67 | 0.64 | 0.70 | 0.68 | 0.65 | 0.66 | 0.69 |
| | **0.79** | **0.81** | **0.81** | **0.79** | **0.80** | **0.80** | **0.78** |
| | $8.4e^{-7}$ | $5.0e^{-5}$ | $3.2e^{-5}$ | $7.5e^{-4}$ | $8.1e^{-7}$ | $7.3e^{-6}$ | $2.6e^{-7}$ |
| *gunif* ↓ | 31.86 | 30.93 | 32.01 | 30.51 | 30.86 | 29.36 | 29.91 |
| | **22.07** | **23.38** | **20.19** | **19.24** | **22.41** | **23.59** | **21.78** |
| | $1.7e^{-6}$ | $3.6e^{-4}$ | $7.1e^{-5}$ | $8.1e^{-7}$ | $1.6e^{-4}$ | $1.4e^{-5}$ | $1.2e^{-6}$ |

Table 14: Quantitative comparison on LSUN Car dataset.

| Metric | MSGAN | VEEGAN | MDGAN | RegGAN | GAN | BiGAN | PacGAN |
|---|---|---|---|---|---|---|---|
| *FID* ↓ | 16.81 | **12.54** | 14.90 | 12.45 | **9.16** | **14.23** | 12.56 |
| | **15.78** | 13.32 | **13.58** | **12.30** | 10.17 | 15.21 | **12.19** |
| | $1.4e^{-1}$ | $9.4e^{-2}$ | $7.4e^{-1}$ | $8.1e^{-2}$ | $5.3e^{-1}$ | $3.6e^{-1}$ | $1.3e^{-1}$ |
| *LPIPS* ↑ | **0.48** | 0.44 | **0.45** | 0.45 | **0.46** | **0.46** | **0.45** |
| | 0.47 | **0.45** | 0.44 | **0.46** | 0.45 | 0.45 | 0.44 |
| | $1.7e^{-1}$ | $8.4e^{-2}$ | $9.7e^{-2}$ | $1.5e^{-1}$ | $9.1e^{-1}$ | $1.4e^{-1}$ | $4.6e^{-1}$ |
| *udiv* ↑ | 0.41 | 0.43 | 0.47 | 0.42 | 0.45 | 0.46 | 0.45 |
| | **0.71** | **0.70** | **0.68** | **0.67** | **0.69** | **0.68** | **0.71** |
| | $3.2e^{-4}$ | $4.4e^{-7}$ | $7.3e^{-4}$ | $1.1e^{-7}$ | $7.4e^{-5}$ | $8.7e^{-4}$ | $1.2e^{-6}$ |
| *gunif* ↓ | 43.48 | 42.77 | 41.82 | 39.32 | 42.77 | 40.21 | 41.32 |
| | **31.59** | **31.68** | **32.55** | **33.67** | **33.71** | **32.46** | **31.88** |
| | $7.6e^{-4}$ | $2.7e^{-8}$ | $2.2e^{-5}$ | $1.1e^{-5}$ | $5.4e^{-8}$ | $3.0e^{-6}$ | $1.4e^{-7}$ |

Table 15: Quantitative comparison on LSUN Bedroom dataset.

| Metric | MSGAN | VEEGAN | MDGAN | RegGAN | GAN | BiGAN | PacGAN |
|---|---|---|---|---|---|---|---|
| *FID* ↓ | **13.87** | 12.66 | 11.63 | **10.93** | 8.20 | 11.04 | 12.53 |
| | 14.21 | **11.97** | **10.67** | 11.66 | 8.96 | **10.87** | **11.87** |
| | $3.7e^{-2}$ | $2.1e^{-1}$ | $1.0e^{-1}$ | $1.9e^{-1}$ | $2.9e^{-1}$ | $1.8e^{-1}$ | $5.3e^{-1}$ |
| *LPIPS* ↑ | **0.49** | 0.47 | **0.46** | 0.45 | 0.46 | 0.46 | 0.45 |
| | 0.48 | **0.48** | 0.45 | **0.46** | **0.47** | **0.47** | **0.46** |
| | $3.4e^{-1}$ | $4.1e^{-1}$ | $3.8e^{-1}$ | $5.6e^{-1}$ | $5.0e^{-1}$ | $7.2e^{-2}$ | $1.1e^{-1}$ |
| *udiv* ↑ | 0.55 | 0.56 | 0.59 | 0.61 | 0.59 | 0.57 | 0.58 |
| | **0.80** | **0.81** | **0.77** | **0.78** | **0.79** | **0.80** | **0.82** |
| | $4.6e^{-5}$ | $8.5e^{-6}$ | $1.4e^{-5}$ | $5.5e^{-7}$ | $2.1e^{-7}$ | $6.0e^{-4}$ | $2.8e^{-5}$ |
| *gunif* ↓ | 35.69 | 36.12 | 34.91 | 35.81 | 33.17 | 36.97 | 33.87 |
| | **23.87** | **21.98** | **25.00** | **24.31** | **22.95** | **26.20** | **24.01** |
| | $5.7e^{-7}$ | $3.5e^{-4}$ | $1.4e^{-7}$ | $7.8e^{-5}$ | $4.3e^{-7}$ | $1.1e^{-6}$ | $3.5e^{-6}$ |

Table 16: Quantitative comparison on LSUN Church dataset.

| Metric | MSGAN | VEEGAN | MDGAN | RegGAN | GAN | BiGAN | PacGAN |
|---|---|---|---|---|---|---|---|
| *FID* ↓ | 15.23 | 13.92 | **13.99** | **12.87** | 10.39 | **13.43** | 12.81 |
| | **14.39** | **13.09** | 14.87 | 15.93 | **9.83** | 14.30 | **11.08** |
| | $1.7e^{-1}$ | $8.4e^{-2}$ | $3.7e^{-1}$ | $6.3e^{-1}$ | $8.0e^{-1}$ | $1.2e^{-1}$ | $8.7e^{-2}$ |
| *LPIPS* ↑ | **0.47** | 0.45 | 0.44 | **0.45** | 0.44 | 0.44 | **0.46** |
| | 0.46 | **0.46** | **0.45** | 0.44 | **0.45** | **0.45** | 0.45 |
| | $1.8e^{-1}$ | $4.7e^{-1}$ | $3.3e^{-1}$ | $4.0e^{-1}$ | $5.2e^{-1}$ | $1.6e^{-1}$ | $8.9e^{-2}$ |
| *udiv* ↑ | 0.63 | 0.66 | 0.64 | 0.63 | 0.65 | 0.65 | 0.66 |
| | **0.79** | **0.81** | **0.77** | **0.80** | **0.78** | **0.79** | **0.81** |
| | $8.4e^{-7}$ | $8.0e^{-4}$ | $6.3e^{-4}$ | $4.8e^{-7}$ | $8.1e^{-4}$ | $3.5e^{-8}$ | $5.6e^{-8}$ |
| *gunif* ↓ | 33.04 | 35.33 | 35.59 | 34.61 | 33.09 | 35.62 | 34.85 |
| | **25.55** | **24.38** | **24.41** | **24.58** | **25.84** | **25.57** | **26.15** |
| | $7.4e^{-8}$ | $1.4e^{-4}$ | $1.7e^{-8}$ | $1.0e^{-7}$ | $6.8e^{-5}$ | $1.5e^{-8}$ | $5.3e^{-6}$ |

Table 17: Quantitative comparison on $128 \times 128$ CelebA dataset.

| Metric | MSGAN | VEEGAN | MDGAN | RegGAN | GAN | BiGAN | PacGAN |
|---|---|---|---|---|---|---|---|
| *FID* ↓ | **17.48** | 18.35 | **17.53** | **18.42** | **16.38** | 16.79 | 16.94 |
| | 18.29 | **18.21** | 17.66 | 20.05 | 17.45 | **16.08** | **16.42** |
| *LPIPS* ↑ | **0.38** | 0.36 | 0.37 | **0.38** | 0.35 | **0.37** | 0.36 |
| | 0.34 | **0.37** | **0.38** | 0.36 | **0.35** | 0.36 | **0.37** |
| *udiv* ↑ | 0.35 | 0.39 | 0.38 | 0.36 | 0.38 | 0.40 | 0.36 |
| | **0.46** | **0.43** | **0.47** | **0.48** | **0.49** | **0.47** | **0.45** |
| *gunif* ↓ | 31.58 | 34.31 | 33.76 | 35.29 | 38.45 | 32.87 | 36.12 |
| | **26.97** | **25.62** | **25.18** | **26.83** | **25.23** | **26.34** | **25.21** |

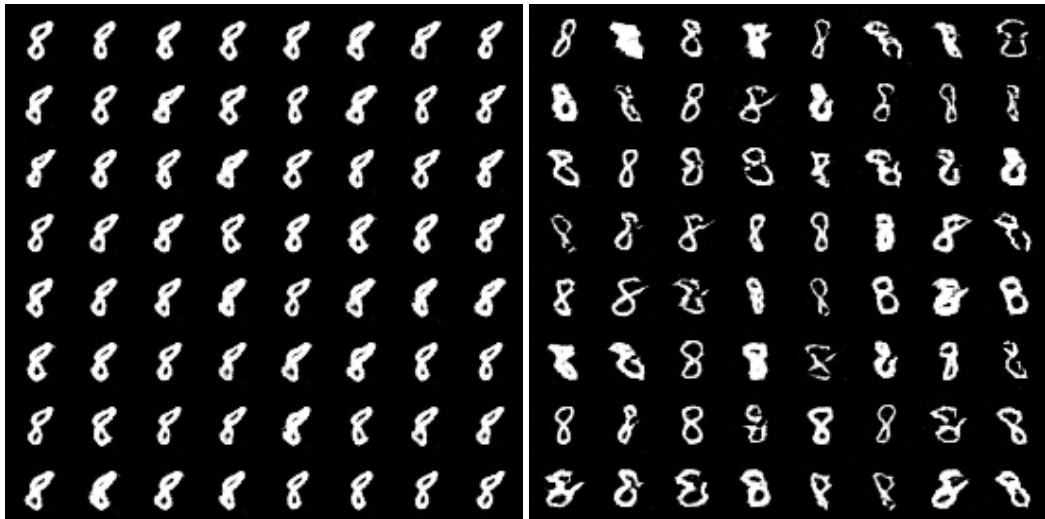

(a) Baseline samples around $x_1$.  (b) Baseline samples around $x_2$.

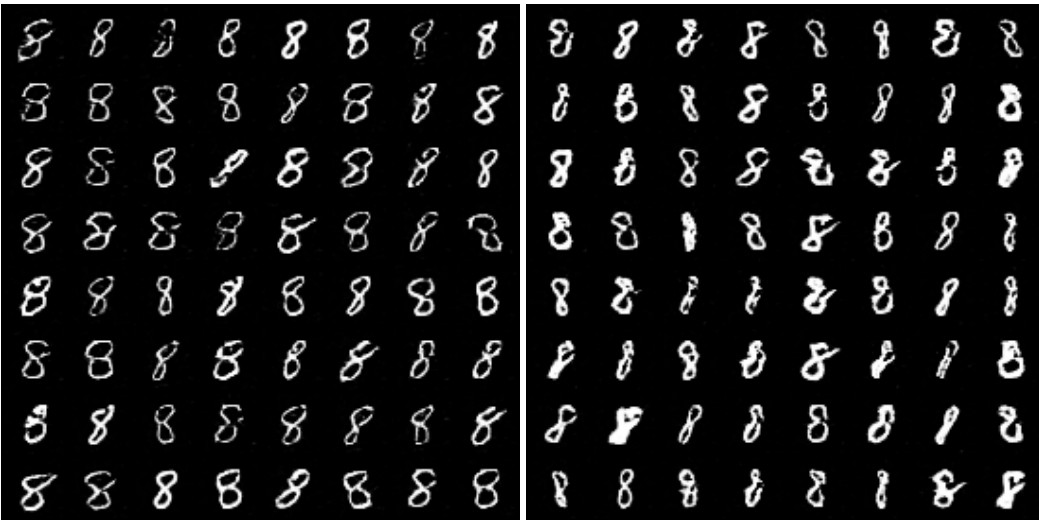

(c) UniGAN samples around $x_1$.  (d) UniGAN samples around $x_2$.

Figure 3: Qualitative results on MNIST dataset.

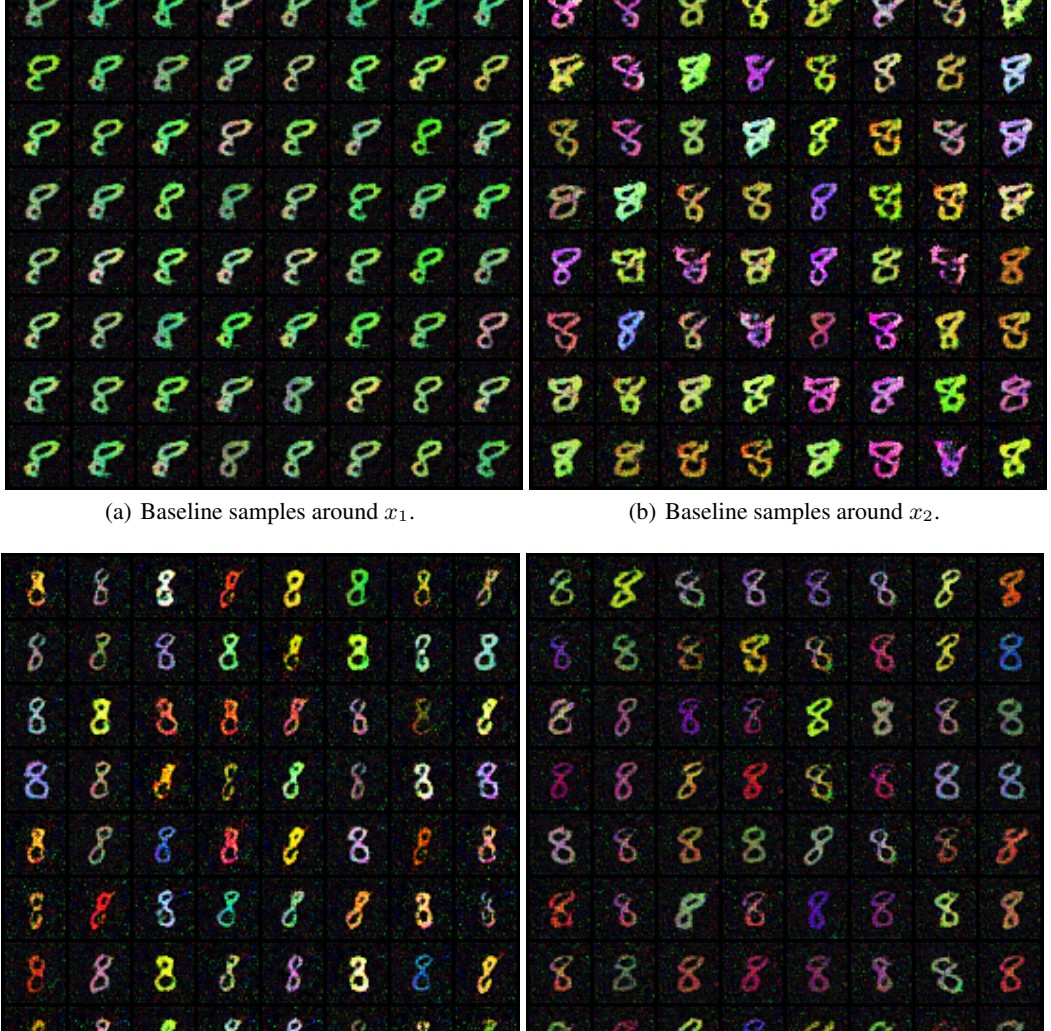

(a) Baseline samples around $x_1$.

(b) Baseline samples around $x_2$.

(c) UniGAN samples around $x_1$.

(d) UniGAN samples around $x_2$.

Figure 4: Qualitative results on colored MNIST dataset.

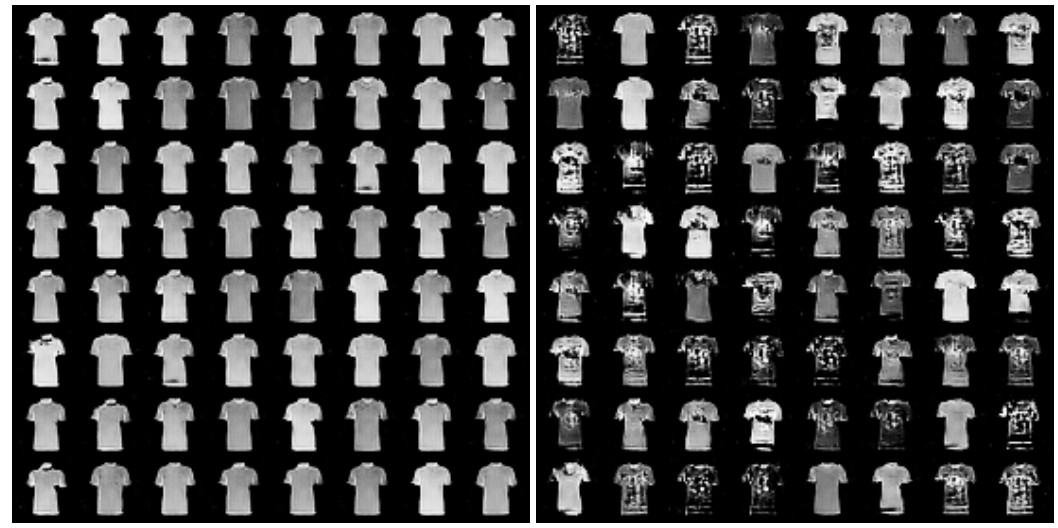

(a) Baseline samples around $x_1$.    (b) Baseline samples around $x_2$.

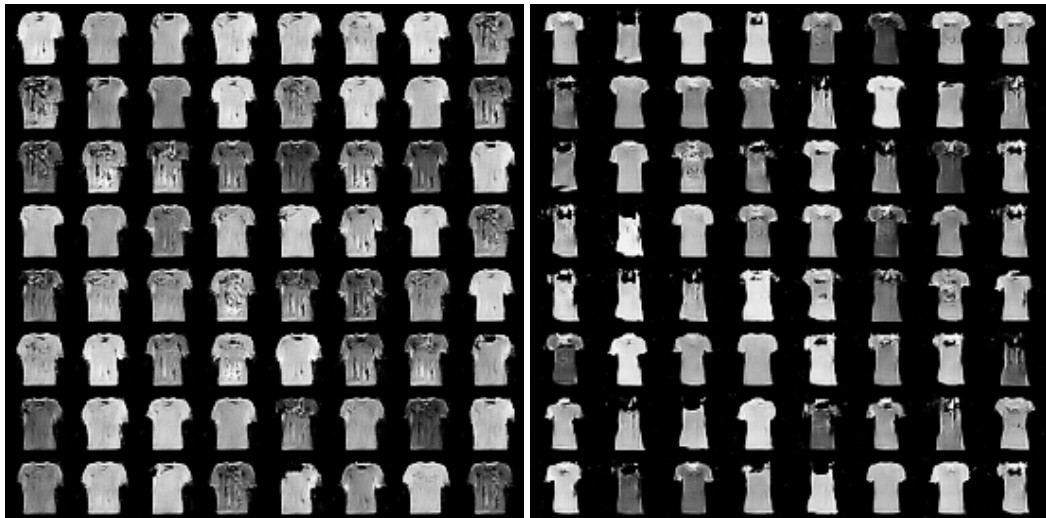

(c) UniGAN samples around $x_1$.    (d) UniGAN samples around $x_2$.

Figure 5: Qualitative results on FashionMNIST dataset.

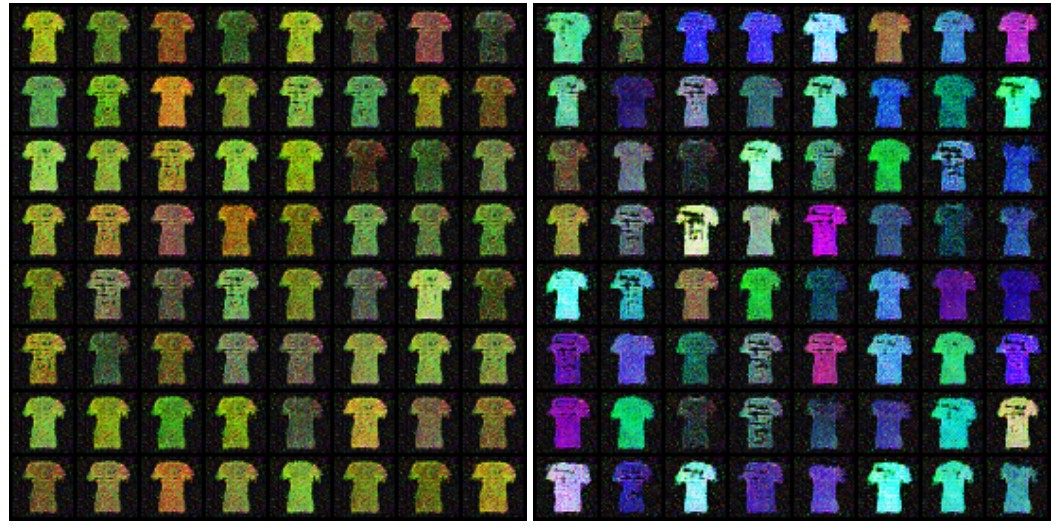

(a) Baseline samples around $x_1$.

(b) Baseline samples around $x_2$.

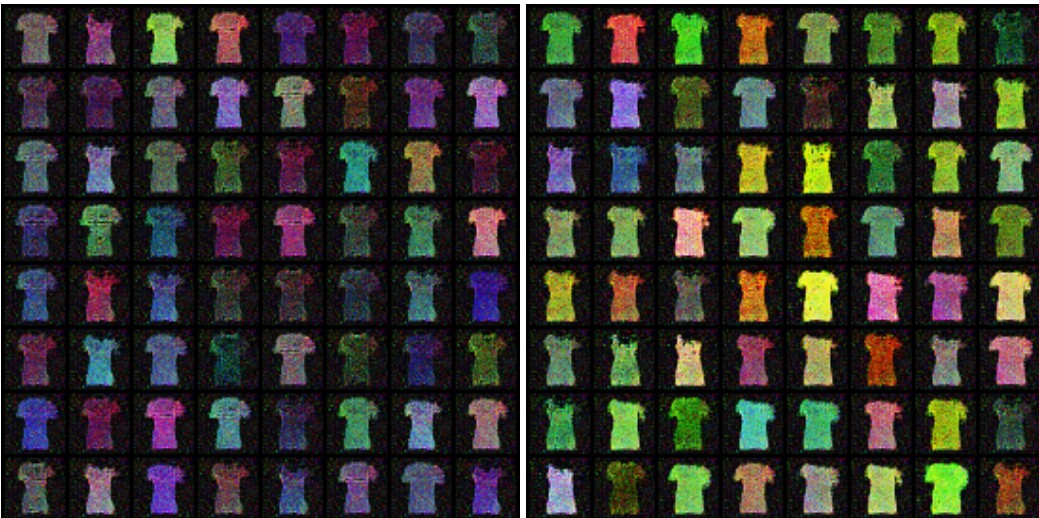

(c) UniGAN samples around $x_1$.

(d) UniGAN samples around $x_2$.

Figure 6: Qualitative results on colored FashionMNIST dataset.

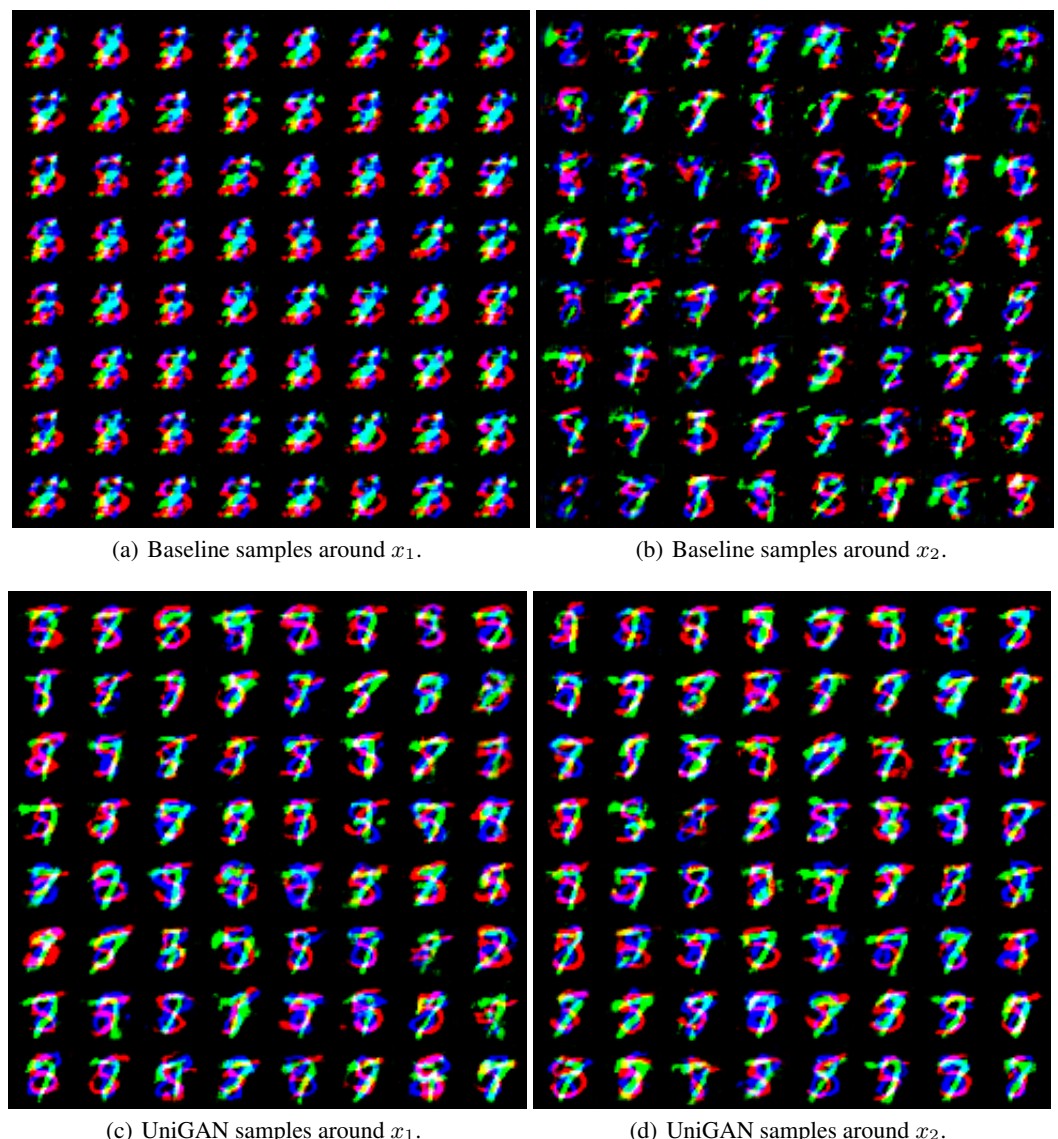

(a) Baseline samples around $x_1$.

(b) Baseline samples around $x_2$.

(c) UniGAN samples around $x_1$.

(d) UniGAN samples around $x_2$.

Figure 7: Qualitative results on stacked-MNIST dataset with channel labels being $5, 7, 8$, respectively.

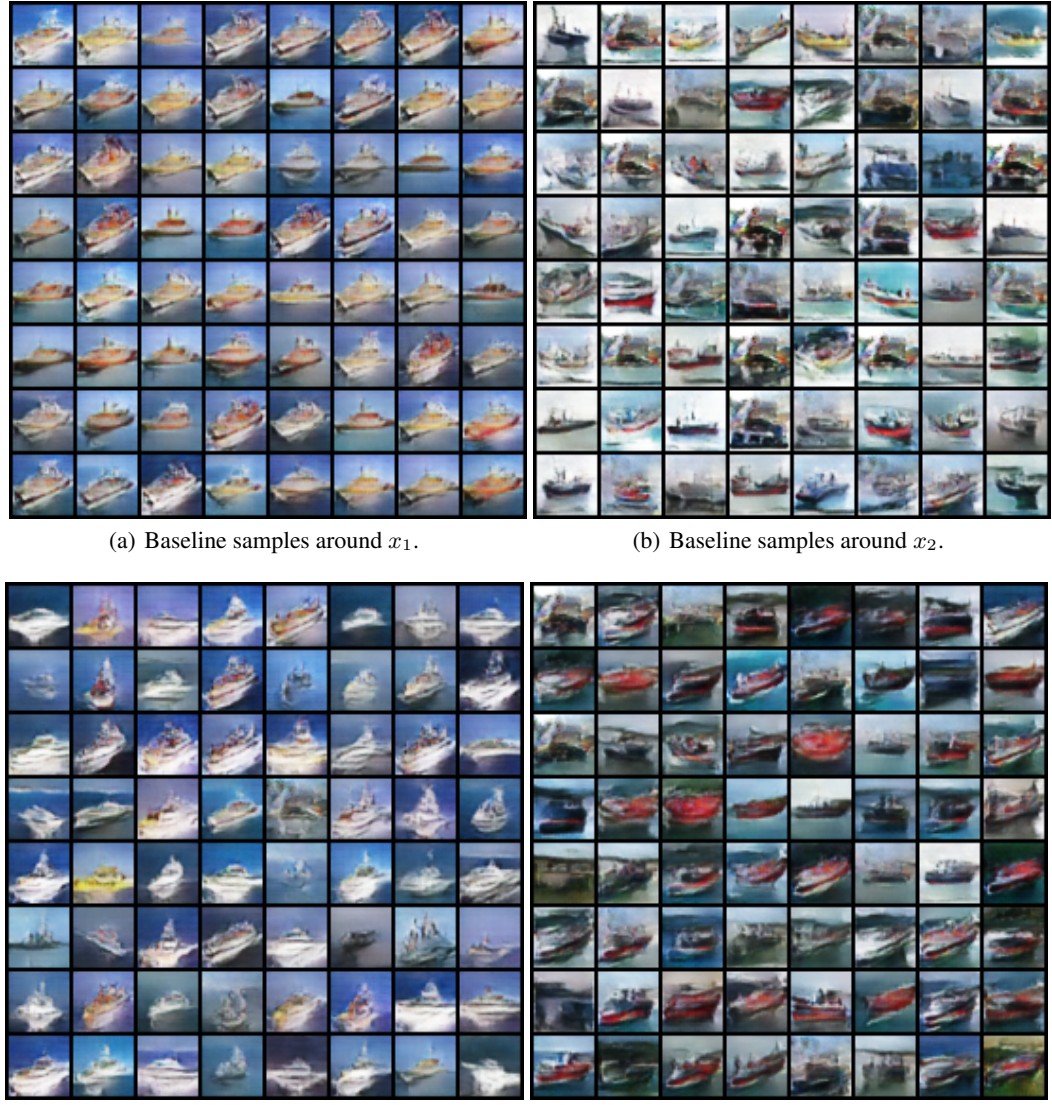

(a) Baseline samples around $x_1$.

(b) Baseline samples around $x_2$.

(c) UniGAN samples around $x_1$.

(d) UniGAN samples around $x_2$.

Figure 8: Qualitative results on CIFAR10 Ship dataset.

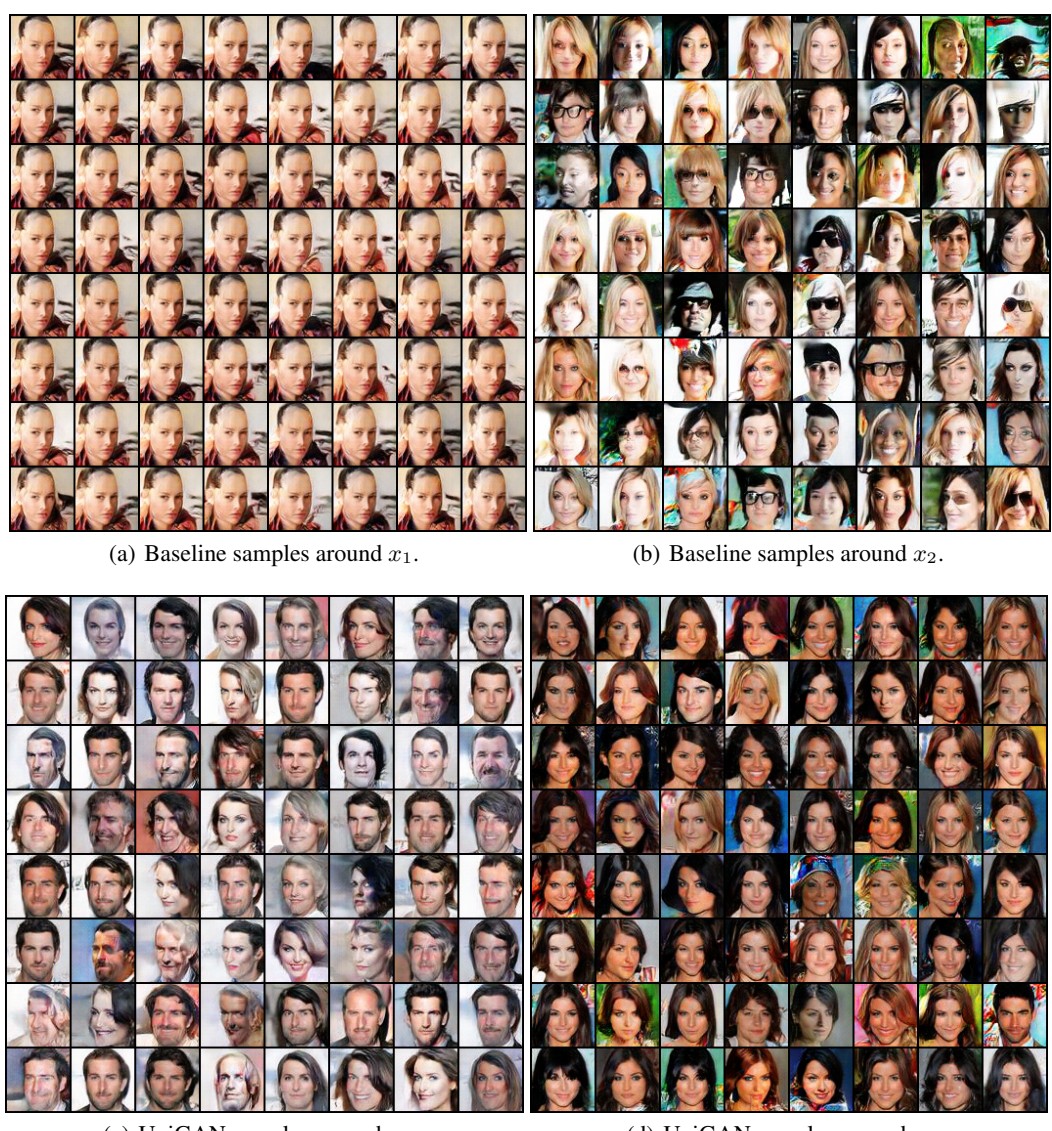

(a) Baseline samples around $x_1$.

(b) Baseline samples around $x_2$.

(c) UniGAN samples around $x_1$.

(d) UniGAN samples around $x_2$.

Figure 9: Qualitative results on CelebA dataset.

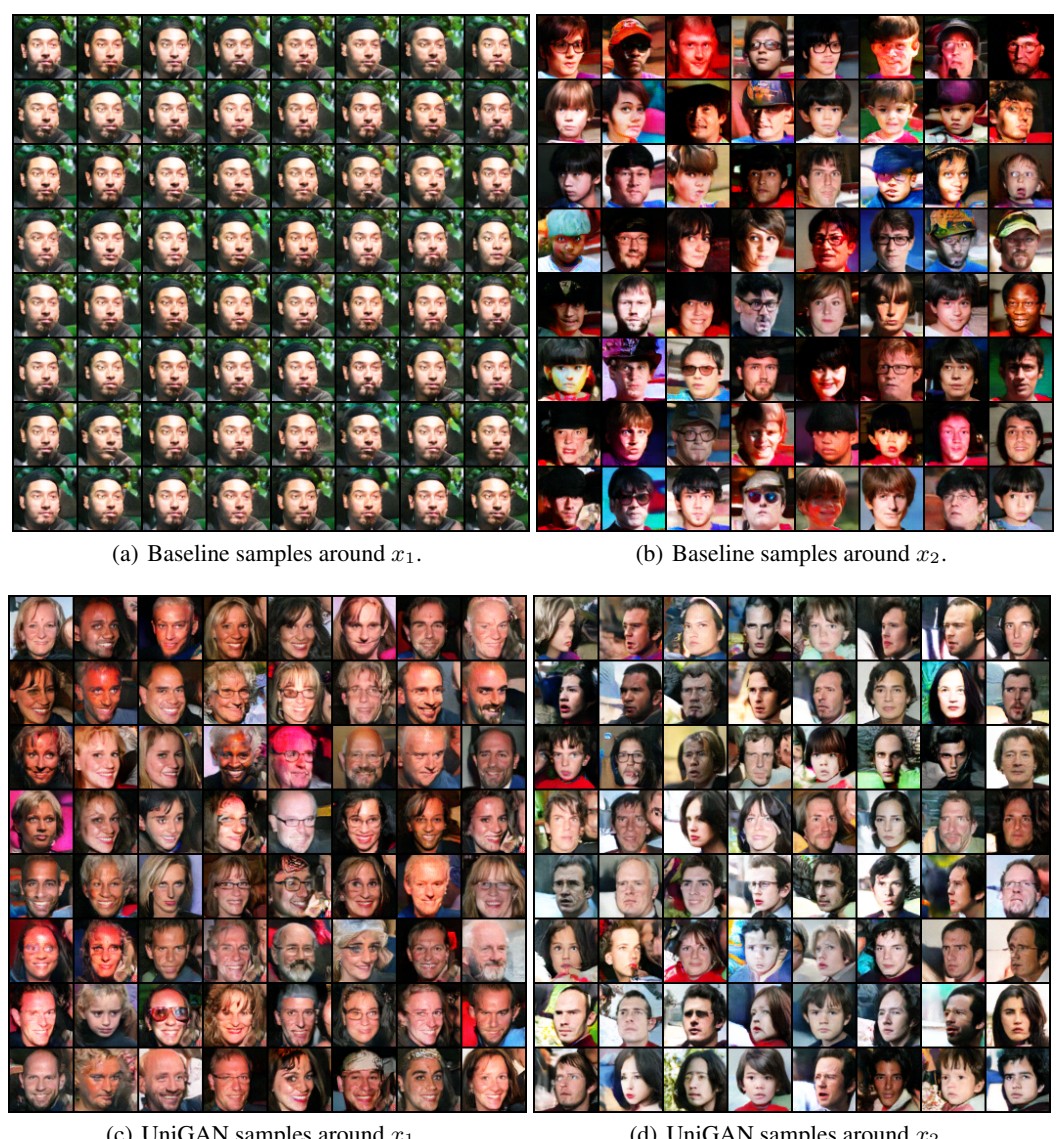

(a) Baseline samples around $x_1$.

(b) Baseline samples around $x_2$.

(c) UniGAN samples around $x_1$.

(d) UniGAN samples around $x_2$.

Figure 10: Qualitative results on FFHQ dataset.

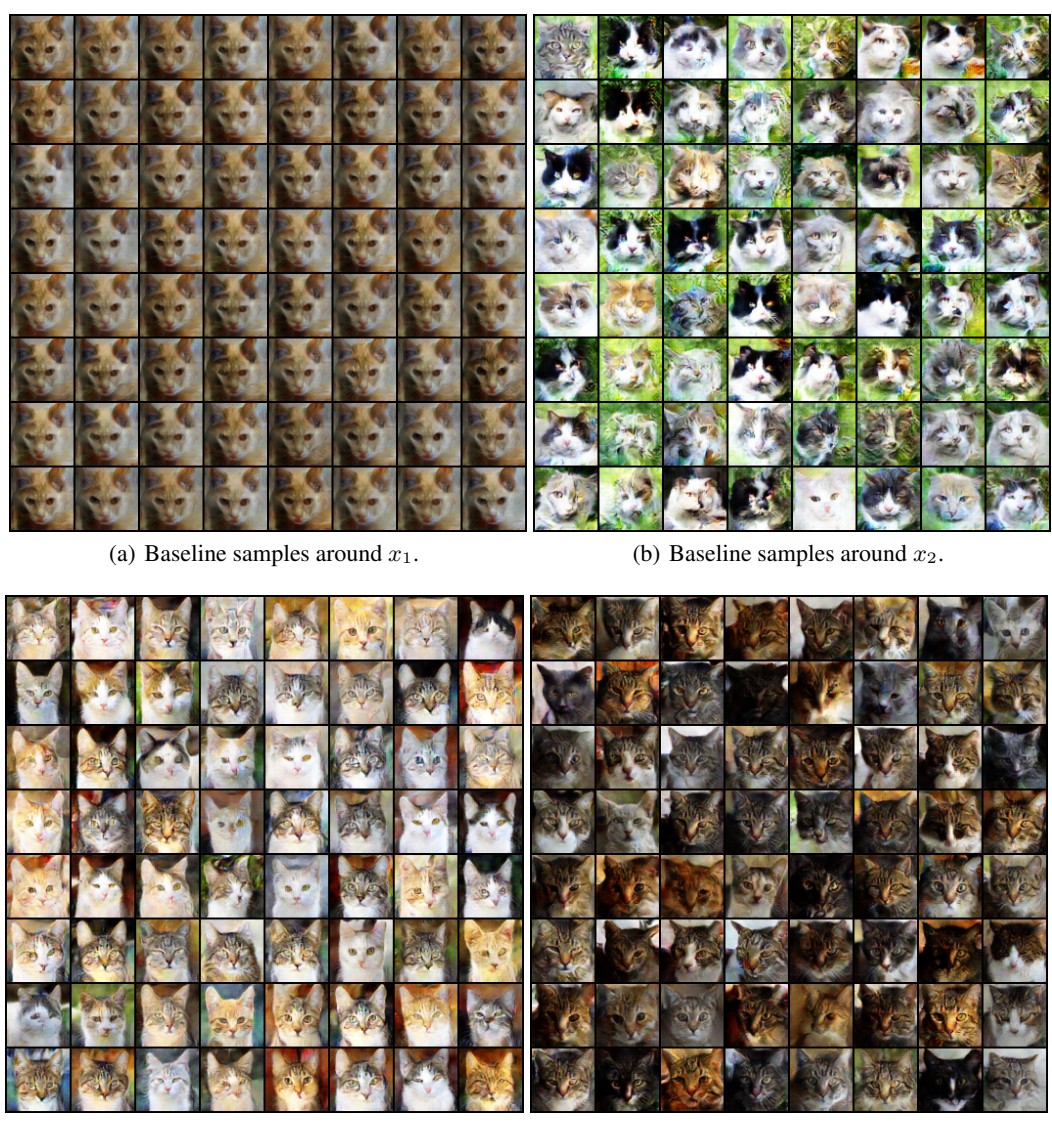

(a) Baseline samples around $x_1$.

(b) Baseline samples around $x_2$.

(c) UniGAN samples around $x_1$.

(d) UniGAN samples around $x_2$.

Figure 11: Qualitative results on AFHQ Cat dataset.

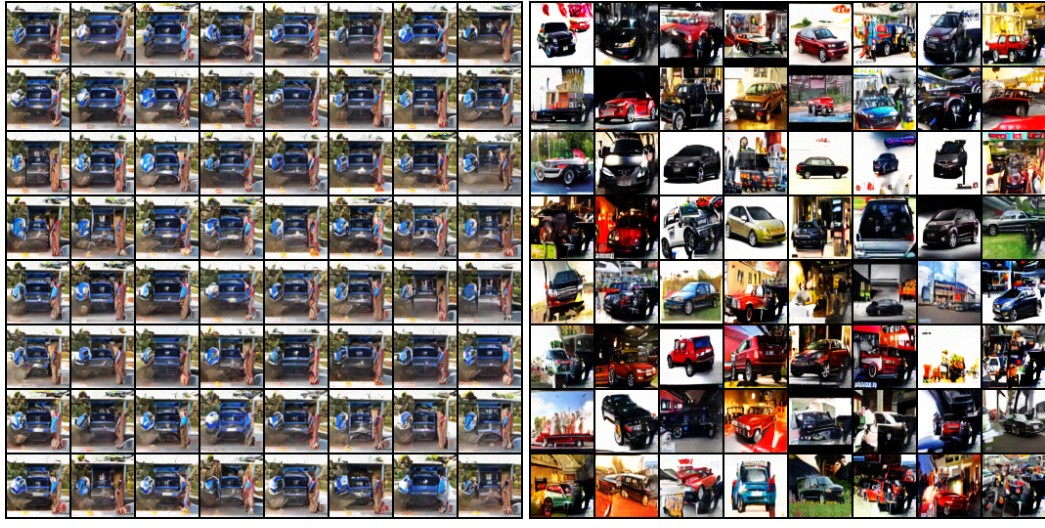

(a) Baseline samples around $x_1$.     (b) Baseline samples around $x_2$.

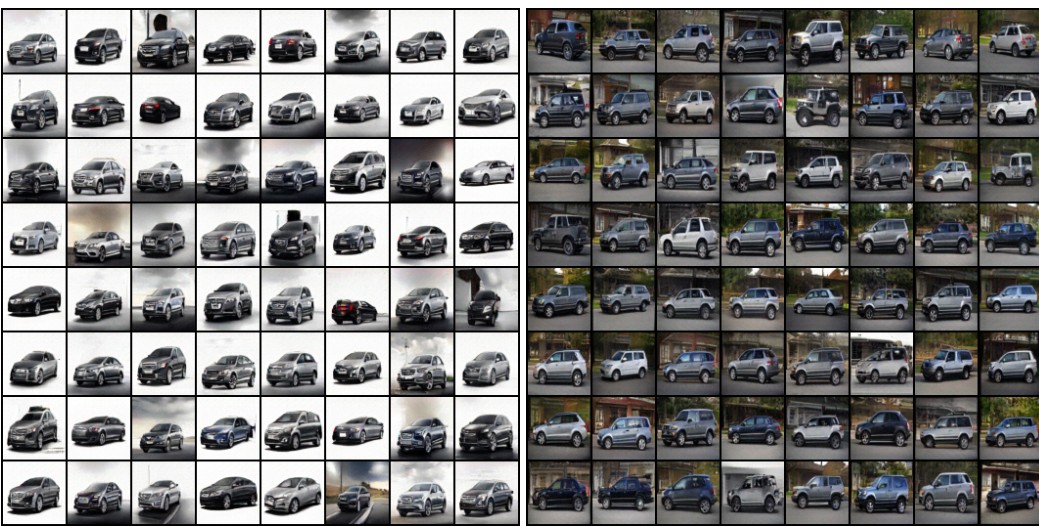

(c) UniGAN samples around $x_1$.     (d) UniGAN samples around $x_2$.

Figure 12: Qualitative results on LSUN Car dataset.

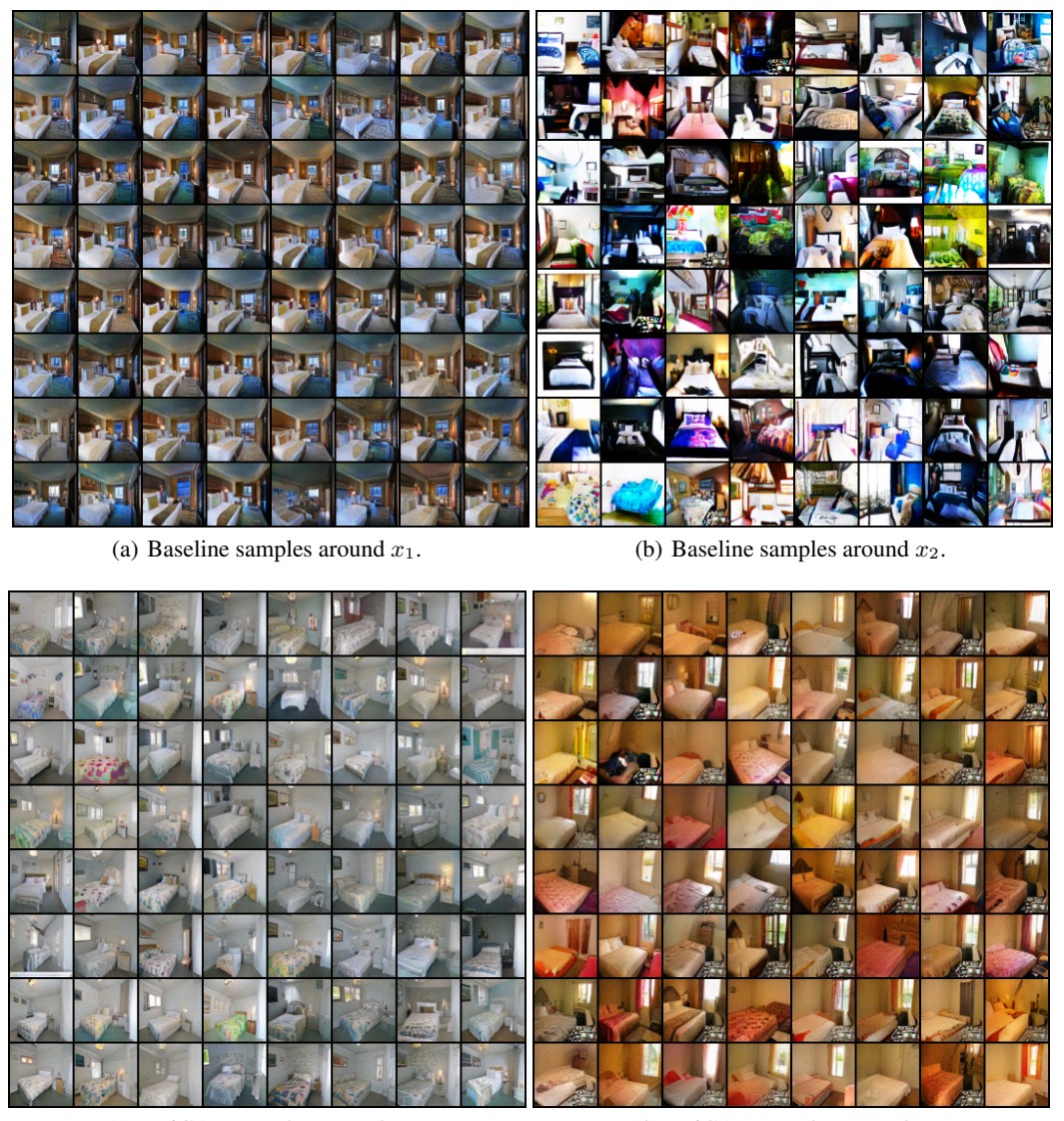

(a) Baseline samples around $x_1$.

(b) Baseline samples around $x_2$.

(c) UniGAN samples around $x_1$.

(d) UniGAN samples around $x_2$.

Figure 13: Qualitative results on LSUN Bedroom dataset.

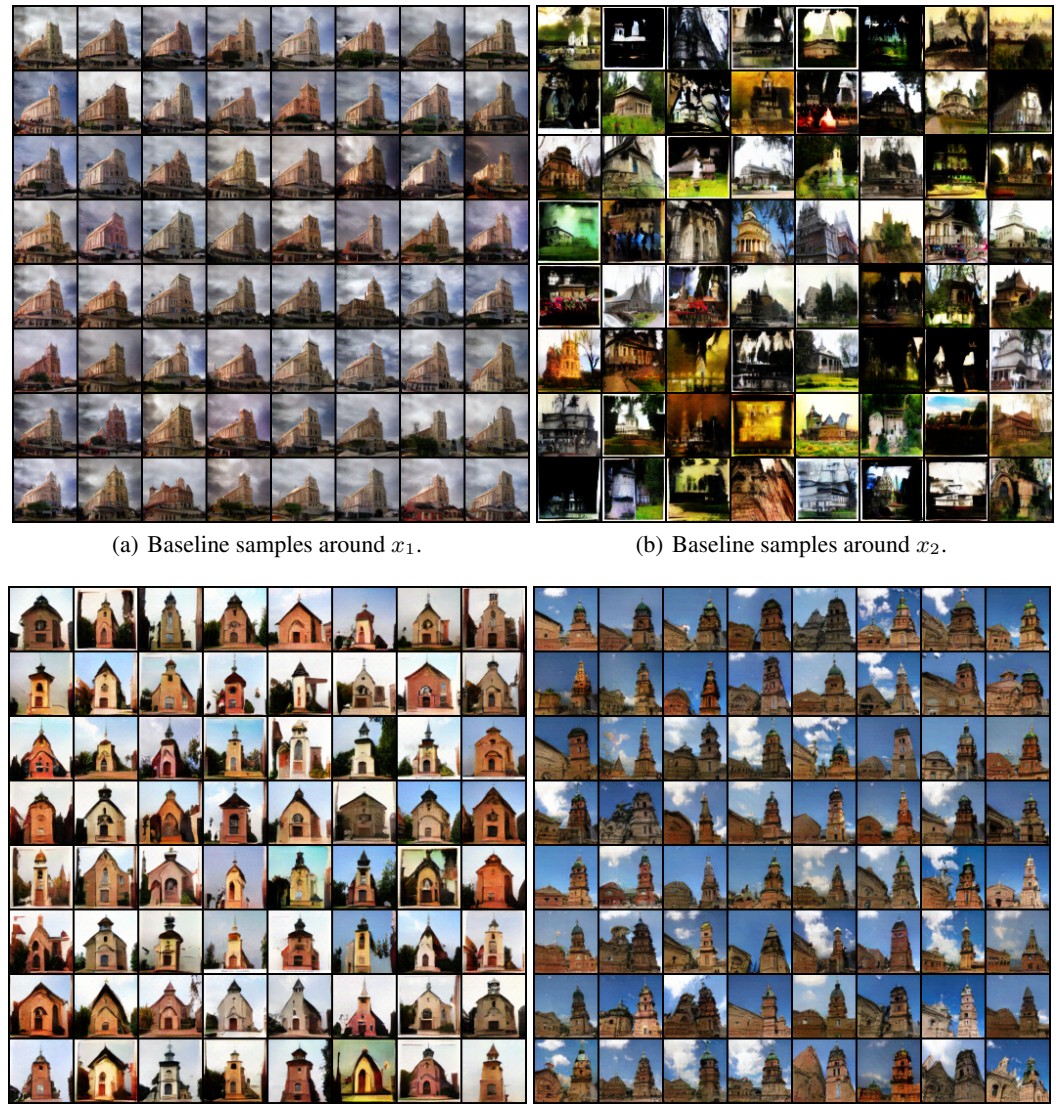

(a) Baseline samples around $x_1$.

(b) Baseline samples around $x_2$.

(c) UniGAN samples around $x_1$.

(d) UniGAN samples around $x_2$.

Figure 14: Qualitative results on LSUN Church dataset.