# OpenReview forum: "UniGAN: Reducing Mode Collapse in GANs using a Uniform Generator"
_NeurIPS.cc/2022/Conference — NeurIPS 2022 Accept_

### Official Review · Reviewer_cP8d · 2022-06-29

**Rating:** 6
**Confidence:** 3
**Soundness:** 3 good
**Presentation:** 3 good
**Contribution:** 3 good

**Summary:**

This paper proposes a new type of generative diversity named uniform diversity, and thus introduce the corresponding $\mu$-mode collapse. And the authors propose a new framework called UniGAN to learn the uniform generator. This framework is formulated with a normalizing flow based generator and a uniformity regularization. Moreover, the authors also propose a new metric called udiv to estimate the uniform diversity.

**Questions:**

Please refer to the weakness.

**Ethics Review Area:**

["I don’t know"]

**Limitations:**

More experimental results, especially the visual analysis can be provided.

**Strengths And Weaknesses:**

Strengths:
This paper proposes a new type of uniform diversity, and analyze the corresponding importance and behavior. And the proposed UniGAN shows great performance under the metric of udiv.

Weakness:
1. The experiments are conducted with small image resolution. And the uniform diversity on the high resolution images is not analyzed. The authors can combine the framework with some up-sampling layers to analyze the uniform diversity with the increase of image resolution.

2. The analysis results of Fig.5 can be made on other datasets to demonstrate the correctness of the conclusion on different datasets, like FFHQ, AFHQ, etc.

3. Will the proposed uniformity regularization suitable for dealing with other types of mode collapses? The relations between other types of mode collapses can be analyzed.

---

> ### Author Response · Authors · 2022-08-01
> **Further evaluations on high-resolution datasets and responses to the other two questions raised**
>
> Thanks for your questions.
>
> Q: Evaluation on high-resolution image datasets?
>
> A: We will conduct experiments on high-resolution image datasets to further evaluate the model. Since experiments on high-resolution image datasets require a lot of time and computational resources, due to limited time before the end of the author response period and limited amounts of GPUs, we cannot immediately provide the experimental results, and please allow us to add the results to the future version of our work.
>
> Q: Analysis similar to Fig. 5 on other datasets like FFHQ, AFHQ, etc?
>
> A: We present qualitative results in supplementary (see Fig. 4-15) to observe the uniformity of the generated distribution. Fig. 5 visually demonstrates the effect of our proposed uniformity regularization on a 2D synthetic dataset. However, for natural image datasets, due to the high dimensionality of images, it is impossible to directly visualize the uniformity of the generated distribution over the entire dataset like the 2D synthetic dataset. Therefore, we randomly select a generated sample and use latent traversal to observe the uniformity of the generated distribution in a neighboring area: if the generated samples of latent traversal vary uniformly, it is considered that the generated distribution in this neighboring area is uniform. Through experiments, when the uniformity regularization is not imposed, we observed that generated samples in some regions collapsed (the samples generated by latent traversal remain the same and did not change at all, see subfigure (a) for each of Fig. 4-15), while in some regions generated samples changed drastically (see subfigure (b) for each of Fig. 4-15), which reflects that the generated distribution is not uniform. However, when the uniformity regularization is imposed, the generated samples of latent traversal in different regions vary more uniformly, which reflects that the generated distribution is more uniform than that obtained without imposing the uniformity regularization.
>
> Q: The relations between different types of mode collapses?
>
> A: As we analyze in Section 3.1 of the main text, $u$-mode collapse is a new mode collapse that focuses on the generated distribution uniformity that cannot be captured by the $\left(\varepsilon,\delta\right)$-mode collapse. Similar to the PPL metric of StyleGAN2, our proposed udiv metric and the traditional LPIPS metric cannot reflect each other, so the $u$-mode collapse and the $\left(\varepsilon,\delta\right)$-mode collapse are two different types of mode collapses, and the proposed uniformity regularization may not improve traditional diversity measured by LPIPS, but the uniform diversity. However, our analysis in Section 3.1 of the main text shows that when $u$ is fixed for the $u$-mode collapse, the $\varepsilon$ is lower bounded for the $\left(\varepsilon,\delta\right)$-mode collapse for a given $\delta$. Moreover, when the generated distribution is sufficiently uniform, the $\left(\varepsilon,\delta\right)$-mode collapse also cease to exist. Therefore, improving uniform diversity can theoretically improve the lower bound on the extent of the $\left(\varepsilon,\delta\right)$-mode collapse.

---

### Official Review · Reviewer_phM4 · 2022-07-10

**Rating:** 7
**Confidence:** 3
**Soundness:** 3 good
**Presentation:** 3 good
**Contribution:** 3 good

**Summary:**

In this paper, the authors propose the maximization of the uniform diversity of the generative distribution as a means for avoiding mode collapse. The main idea relies on the diffeomorphism between input uniform distribution and the samples that can be obtained using a normalizing flow -based generator.  The key insight is due to this property being preserved, one can maximize the uniform diversity.

The paper provides useful theoretical and empirical analysis results that validate their work

**Questions:**

It will be useful if the authors could comment on the weaknesses mentioned above.

**Limitations:**

The negative social impact are not particularly applicable.

The limitations to some extent are addressed in the supplementary material.

**Strengths And Weaknesses:**

Strengths:
1) The use of uniform diversity as a measure to evaluate mode collapse in GANs is interesting. The ideas are also related to the Epsilon-Delta analysis of PacGAN
2) Thorough evalution of their work by considering various existing techniques to avoid mode-collapse and showing that incorporating uniform diversity improves their performance
3) Detailed theoretical analysis of their work with detailed results being provided in the supplementary material.

Weakness:
1) The main observation is that the existing uniform diversity can be achieved if one obtains the diffeomorphism between the distribution and the sample space. However, this exists only in cases such as normalizing flows. In general, normalizing flows have invertible properties but suffer from being inferior in terms of sample generation. It is not clear, how exactly this limitation is addressed in this work.
2) As observed, the normalizing flows are limited in terms of the size of the latent space being exactly the same as the original sample space. The method adopted to solve this by zero-padding appears to be a heuristic. This needs to be better understood.

---

> ### Author Response · Authors · 2022-08-01
> **Responses to the two questions raised**
>
> Thanks for your questions.
>
> Q: How to address the limitation of inferior generative quality of NF-based models?
>
> A: We observe that using a strong discriminator can lead to high-quality generated samples. Specifically, when we use the powerful StyleGAN2 discriminator (see Table 4 of supplementary), the FID scores can be reduced to a very low level (eg, FID<10 on CelebA dataset, and see more quantitative results in Table 12-17 of supplementary), which shows that our NF-based generator can also generate high quality samples when the discriminator is powerful enough.
>
> Q: How to understand the zero-padding manner of the proposed NF-based generator?
>
> A: Each layer of the generator consists of a padding module for padding zeros to boost dimensionality of input features, and a flow module for nonlinear transformation. Corresponding to the traditional convolutional network-based generator, the padding module can be considered as an $\mathrm{Upsample}$ layer and the flow module can be considered as a $\mathrm{Conv}+\mathrm{BN}+\mathrm{ReLU}$ layer.

---

### Official Review · Reviewer_cYPc · 2022-07-11

**Rating:** 7
**Confidence:** 2
**Soundness:** 3 good
**Presentation:** 3 good
**Contribution:** 4 excellent

**Summary:**

The paper proposes a simple yet effective way to mitigate the mode collapse issue of GANs. To this end the authors introduce the generator uniformity property, which is utilized to regularize a flow-based generator. Lastly, a new form of diversity is introduced, labelled uniform diversity.

**Questions:**

I suggest the authors follow my suggestions regarding further evaluation as well as the inclusion of a discussion on ethical considerations.

**Limitations:**

The authors have provided adequate analysis of the limitations of the work. However, I would like to see a discussion on the societal impact of the work.

**Strengths And Weaknesses:**

Strength:

Although I am not particularly experienced in this particular sub-area (i.e., mitigating mode-collapse in GANs), I appreciate the effort the authors put into the technical part of the motivation. The main idea is clear and well motivated. The authors also made sure to thoroughly evaluate their model against a number baselines and metrics, which highlight the efficacy of the method.

Weakness:

Although I don’t think the paper has any strong weakness, I would like to see how the approach performs on other mode-coverage benchmarks, e.g.,
- Fashion-MNIST and partial MNIST of [1]
- Stacked MNIST of  [2]

Lastly, I would also appreciate a discussion on the ethical considerations of this work, especially since it touches the task of generative modelling of faces.

[1]. Rethinking Generative Mode Coverage: A Pointwise Guaranteed Approach

[2] . VEEGAN: Reducing Mode Collapse in GANs using Implicit Variational Learning

---

> ### Author Response · Authors · 2022-08-01
> **The negative societal impact of our work and further evaluations on the mentioned two datasets**
>
> Thanks for your questions.
>
> Q: Negative societal impact of our work?
>
> A: Techniques that generate high-quality fake images (especially human face images) such as DeepFake may be used for malicious purpose, which brings negative societal impacts. Although our method can improve uniformity of generated distributions, the quality of generated images may be more important in practical applications of fake image generation. Since improving generative quality is not the purpose of our work, our work may not pose a challenge to DeepFake detection that prevents malicious use of high-quality fake images, and hence the negative societal impacts may not be particularly applicable for our work.
>
> Q: Further evaluation on the FashionMNIST and partial MNIST dataset as well as the stacked-MNIST dataset?
>
> A: We provide further evaluation on the mentioned two datasets, see Table 9\&10 for quantitative results and Fig. 8 for qualitative results in our revised supplementary. Similar to datasets that provide class labels such as MNIST, FashionMNIST and CIFAR, the mentioned two datasets have multiple discrete modes with each mode corresponding to one class. As we mentioned in Line 147-176 of supplementary, we adopt a conditional generation setting (ie, using $g\left(z;y\right)$ to generate an image, where $g$ is the generator, and $z$ and $y$ are the latent code and the class label, respectively) for datasets that provide class labels, because different classes (modes) correspond to different disjoint submanifolds, and the union of all the disjoint submanifolds cannot be homeomorphic to an continuous Euclidean latent space. Therefore, under the conditional generation setting $g\left(z;y\right)$, ideally, we can cover all the discrete modes by traversing all the class labels $y$ for $g\left(z;y\right)$. In our experiments, for the model trained on each dataset, we first randomly sample 10000 class labels $y^{\left(i\right)}$ and latent codes $z^{\left(i\right)}$, then obtain generated samples $\left\\{x^{\left(i\right)}=g\left(z^{\left(i\right)};y^{\left(i\right)}\right)\right\\}_{i=1}^{10000}$ for evaluation. Our model can cover all 11 modes of the FashionMNIST and partial MNIST dataset and most of the 1000 modes of the stacked-MNIST dataset.

---

### Official Review · Reviewer_YpC8 · 2022-07-16

**Rating:** 4
**Confidence:** 4
**Soundness:** 2 fair
**Presentation:** 2 fair
**Contribution:** 2 fair

**Summary:**

This paper proposes UniGAN a new approach to alleviate mode collapse in GANs. Assuming the manifold hypothesis, the authors motivate training a generator with uniform distribution over the data manifold M. They encourage the uniform distribution by arguing that samples on the manifold M are equally accepted as real samples for training GANs. UniGAN restricts the generator to be Normalizing Flow (NF) based to perform effective and simple regularization in pursuit of a uniform generator. Authors also propose a new measure of performance of GANs and show the effectiveness of their methods on several benchmark datasets experimentally.

**Questions:**

Please look at my above comments.

**Limitations:**

Yes

**Strengths And Weaknesses:**

Pros:

1- the idea of the paper clearly has been explained.

2- The idea is novel.

3-The paper has solid theoretical results.

4- A large set of experiments has been done.

Cons:

One of the main reasons GANs have been preferred to other generative models despite mode collapse is their sample quality. However, the current paper restricts the generator to an NF-based model, which restricts the flexibility of the generator and consequently lowers the quality of samples. This also can be shown by looking at the Inception score of the trained model on CIFAR; UniGAN achieve IS of less than 3.9; however, PacGAN archives IS of more than 6 (please refer to Self-Cond-GAN or PGMGAN) when used with proper architecture.

Also, I have not convinced necessarily a uniform distribution is necessarily better. For an instant, consider toy data of 1D mixture of normals distribution. The support of this distribution is the entire R, and it is not ideal to have a uniform distribution over the entire space. Instead, one may prefer some samples to others.

---

> ### Author Response · Authors · 2022-08-01
> **Responses to the two questions raised**
>
> Thanks for your questions.
>
> Q: Why UniGAN achieves low IS scores on CIFAR dataset?
>
> A: Regarding the difference in IS scores between UniGAN and PGMGAN on the CIFAR dataset, in addition to being likely caused by the different generator architectures of the two models (we use an NF-based generator, while PGMGAN uses a ResBlock-based generator), it is more likely caused by the different discriminator capabilities of the two models. As we show in Table 3 of supplementary, the architecture of the discriminator we used for training on the CIFAR dataset is very simple, it consists of only a few vanilla convolutional layers and the total amount of model parameters is only 0.188M. However, the discriminator of PGMGAN consists of multiple ResBlocks, which is relatively more capable. In addition, it can be seen from supplementary that for the natural image datasets, when we use the powerful StyleGAN2 discriminator (see Table 4 of supplementary), the FID scores that measures the quality of generated samples can be reduced to a very low level (eg, FID<10 on CelebA dataset, and see more quantitative results in Table 12-17 of supplementary), which shows that our NF-based generator can also generate high quality samples when the discriminator is powerful enough.
>
> Q: Is a uniform distribution necessarily better?
>
> A: Regarding the concern that a uniform distribution is not necessarily better, it is indeed not ideal for 1D data with the support being the entire $\mathbb{R}$ to have a uniform distribution over the entire infinite $\mathbb{R}$ space. However, for natural image datasets such as human faces, a uniform distribution over the manifold is reasonable, because all human face images fall on a manifold restricted to a bounded region $\left[0,255\right]^{C\times H\times W}$ rather than extending to the entire infinite $\mathbb{R}^{C\times H\times W}$ space, where $\left[0,255\right]$ is the range of pixel values and $C\times H\times W$ is the dimensionality of the image. Therefore, it is reasonable to adopt a uniform distribution on a finite manifold. In addition, it is subjective to adopt which kind of distribution over the support set. Although one may prefer some samples to others, we adopt the uniform distribution over the manifold because we take into account that every sample on the manifold can be equally accepted as a real image, which should also be acceptable.

---

> > ### Comment · Reviewer_YpC8 · 2022-08-08
> > **Response to Authors and future question**
> >
> > I thank the authors for the response!
> >
> > Regarding the high quality of CelebA dataset having low FID score and justifying lowering the power of the generator to NF-based can still result in high-quality samples if the discriminator is powerful enough:
> >
> > The current paper's best FID on CelebA in the appendix is 9.16; however, StyleGAN2 itself archives the FID score of 5.2 on CelebA. It is important to consider the FID across different datasets with different scales. And I would appreciate it if the authors could help me to understand better.
> >
> > Regarding the usage of uniform generation for image data, based on one manifold hypothesis,
> >  In fact, in GAN literature, papers that design their method to handle disconnected manifolds could achieve better performance, such as: ( Diverse Image Generation via Self-Conditioned GANs) or PGMGAN therefore, I do not see why it is helpful to have that kind of generator in GAN literature, which their main advantage is high-quality samples.

---

> > > ### Author Response · Authors · 2022-08-09
> > > **Response to the questions**
> > >
> > > Thanks for your questions.
> > >
> > > In terms of the FID across different datasets, we provide quantitative results on natural image datasets in Table 12-17 in supplementary. Our NF-based model can achieve the FID scores of 8.22 (CelebA), 11.22 (FFHQ), 9.16 (LSUN Car), 8.20 (LSUN Bedroom), 9.83 (LSUN Church). Though StyleGAN2 can achieve the FID scores <5 on these datasets, the total number of parameters of the StyleGAN2 (18.828M) is much larger than that of our generator (8.882M), hence the comparison between the two models is not very fair. In terms of the FID across different scales, due to limited amounts of GPUs, we will add the experimental results on high-resolution image datasets to the future version of our work.
> > >
> > > In terms of why methods handling disconnected manifolds could achieve better performance, the BourGAN [1] paper can provide some inspiration. Specifically, the authors of BourGAN demonstrate that using a Lipschitz continuous generator to map a Gaussian prior distribution over a connected latent space to the data distribution with disconnected modes may lead to arbitrarily large gradients and hence unavoidably results in unwanted samples (see Fig. 1 of [1]). In our paper, we also adapt our generator uniformity regularization to the setting of multiple disconnected modes, see Line 147-176 in our supplementary.
> > >
> > > [1] BourGAN: Generative Networks with Metric Embeddings. NIPS, 2018.

---

### Meta-Review · Area_Chair_TcgF · 2022-08-30

**Recommendation:** Accept
**Confidence:** Less certain

**Metareview:**

This paper proposes UniGAN to alleviate mode collapse in GANs. They encourage the uniform distribution by arguing that samples on the manifold are equally accepted as real samples for training GANs.

The paper is comprehensive in both theory and experimental results. It receives average rating score 6, leading to an ``Accept'' decision.

To further improve the impact of this paper, I suggest the authors to study it in the context of modern SoTA image generation models in the future. Hopefully, It may help the GAN-based model family [1,2,3] to improve the performance, in the competition with diffusion-model, auto-regressive models.

References:
- [1] Alias-Free Generative Adversarial Networks (StyleGAN3)
- [2] LAFITE: Towards Language-Free Training for Text-to-Image Generation
- [3] ViTGAN: Training GANs with Vision Transformers

**Award:**

No

---

### Decision · Program_Chairs · 2022-09-14

Accept